



# MICRU background map and effective cloud fraction algorithms designed for UV/vis satellite instruments with large viewing angles

Holger Sihler[1], Steffen Beirle[1], Steffen Dörner[1], Marloes Gutenstein-Penning de Vries[1,2],
Christoph Hörmann[1,3], Christian Borger[1], Simon Warnach[1], and Thomas Wagner[1]

[1]Max Planck Institute for Chemistry, Hahn-Meitner-Weg 1, 55128 Mainz, Germany
[2]now at: Deutscher Wetterdienst, Offenbach, Germany
[3]now at: Volume Graphics GmbH, Heidelberg, Germany

**Correspondence:** holger.sihler@mpic.de

**Abstract.** Clouds impact the radiative transfer of the Earth's atmosphere and strongly influence satellite measurements in the UV visible and IR spectral ranges. For satellite measurements of trace gases absorbing in the UV/vis spectral range, particularly clouds ultimately determine the vertical sensitivity profile, mainly by reducing the sensitivity for trace gas columns below the cloud.

The Mainz Iterative Cloud Retrieval Utilities (MICRU) algorithm is specifically designed to reduce the error budget of trace gas retrievals, such as those for nitrogen dioxide ($NO_2$), which strongly depends on the accuracy of small cloud fractions (CF) in particular. The accuracy of MICRU is governed by an empirical parametrisation of the viewing geometry dependent background surface reflectivity taking instrumental and physical effects into account. Instrumental effects are mainly degradation and polarisation effects, physical effects are due to the anisotropy of the surface reflectivity, e. g. shadowing of plants and sun

glitter.

MICRU is applied to main science channel (MSC) and polarisation measuring device (PMD) data collected between April 2007 and June 2013 by the GOME-2A instrument onboard the MetOp-A satellite. CF are retrieved at different spectral bands between 374 and 758 nm. The MICRU results for MSC and PMD at different wavelengths are inter-compared to study CF precision and accuracy, which depend on wavelength and spatial correlation. Furthermore, MICRU results are compared to

FRESCO (Fast Retrieval Scheme for Clouds from the Oxygen A band) and OCRA (Optical Cloud Recognition Algorithm) operational cloud products.

We show that MICRU retrieves small CF with an accuracy of 0.04 or better for the entire 1920 km wide swath with a potential bias between -0.01 and -0.03. CF retrieved at shorter wavelengths are less affected by adverse surface heterogeneities. The comparison to the operational CF algorithms shows that MICRU significantly reduces the dependence on viewing angle,

time, and sun glitter. Systematic effects along coasts are particularly small for MICRU due to its dedicated treatment of land and ocean surfaces.

The MICRU algorithm is designed for spectroscopic instruments ranging from the GOME to TROPOMI/Sentinel-5P, but is also applicable to UV/vis imagers like, for example, AVHRR, MODIS, VIIRS, and Sentinel-2.





## 1 Introduction

Clouds are the most clearly visible component of the atmosphere, both from the ground and from space. Their presence increases the shortwave albedo of the Earth and, hence, reduces the amount of solar radiation absorbed by the Earth. Clouds furthermore alter the radiative transfer (RT) within the atmosphere by effectively shielding the underlying atmosphere and
surface from observation. This paper focuses on the influence of clouds on the measurement sensitivity for atmospheric trace-gases retrieved from satellite observations in the UV and visible (UV/vis) spectral ranges. The largest portion of the Earth's surface is darker than overlying clouds. In this scenario, the measurement sensitivity – that is the air mass factor (AMF) (Noxon et al., 1979; Solomon et al., 1987; McKenzie et al., 1991; Perliski and Solomon, 1993; Sarkissian et al., 1995; Rozanov and Rozanov, 2010; Deutschmann et al., 2011) – is decreased for trace gas abundances below clouds and increased above
(see e. g. Wagner et al., 2003). For trace gases located within clouds, however, the influence of clouds depends on the cloud characteristics and trace-gas profiles. In any case, cloud properties need to be carefully constrained in order to achieve high accuracy in tropospheric trace-gas measurements from satellites (De Smedt et al., 2008; Liu et al., 2011; Barkley et al., 2012; Lorente et al., 2017). Boersma et al. (2004), for example, estimated the uncertainty in the tropospheric air mass factor for GOME measurements of nitrogen dioxide ($NO_2$) due to uncertainties in the cloud fraction between 25 and 30% over large
parts of North America, Europe and South East Asia, where most anthropogenic $NO_2$ emission occur.

Satellite retrievals of trace gases with high abundances in the boundary layer depend on the amount and properties of clouds within one satellite pixel. The effective cloud fraction (CF, $c$) is one measure to quantify cloud contamination. CF is defined by

$$c = \frac{R - R_{\min}}{R_{\max} - R_{\min}} \tag{1}$$

based on the top of atmosphere (TOA) reflectance

$$R = \frac{\pi I}{E_0 \cos \vartheta_0} \tag{2}$$

with TOA radiance $I$, solar irradiance $E_0$, and solar zenith angle $\vartheta_0$. This definition of $c$ applies the independent pixel-approximation (IPA) neglecting the influence of horizontal RT (Martin et al., 2002). $R_{\min}$ and $R_{\max}$ in Eq. (1) denote lower and upper thresholds corresponding to reflectances from cloud free and completely cloudy pixel, respectively. $R_{\min}$ depends
on the reflecting properties of the surface and viewing geometry. Hence, an $R_{\min}$ approximating the time dependent the actual bidirectional reflectance distribution function (BRDF) is required for any geolocation in order to retrieve small CF at high accuracy. In contrary, $R_{\max}$ depends on cloud albedo. The cloud albedo depends on optical density and scattering properties, which can be described by an a-priory cloud model, in addition to the observation geometry. $R_{\max}$ may be empirically determined (e. g Grzegorski et al., 2006; Lutz et al., 2016) or calculated using a radiative transfer model (RTM) (e. g. Wang et al.,
2008). MICRU applies a Lambertian cloud model with a fixed albedo of 0.8 Stammes et al. (2008) rendering $R_{\max}$ depending on observation geometry alone.

Within tropospheric trace gas retrievals, cloudy pixels are usually flagged applying a threshold for $c$ between 10 and 30%, and a high accuracy of $c$ is required for the correction of cloud effects of remaining pixels. Specifically, the accurate determination





of $R_{\min}$ is crucial to determine small CF accurately. $R_{\min}$ depends on the geolocation and time and, therefore, maps of the lower threshold are needed. The Heidelberg iterative cloud retrieval utilities (HICRU), for example, derive background maps based on TOA reflectances using imaging processing techniques (Grzegorski et al., 2006; Grzegorski, 2009). In contrary, Koelemeijer et al. (2003) take Rayleigh scattering into account providing Lambert-equivalent reflectivity (LER) maps. More sophisticated methods apply complex BRDF accounting for the anisotropy of the surface reflectivity (Vasilkov et al., 2017; Lorente et al., 2018).

Algorithms for the retrieval of background maps usually rely on completely cloud-free observations over a certain location. Here, we are interested in CF of spectroscopic measurements. Compared to imager instruments, spectrometers are characterised by lower spatial but much higher spectral resolution. For example, the imager Visible Infrared Imaging Radiometer Suite (VIIRS) features 22 spectral channels at a resolution of approximately 400 m whereas the spectrometer Ozone Monitoring Instrument (OMI) offers 1176 spectral channels at a nadir resolution of 13 km × 24 km (Schueler et al., 2002; Levelt et al., 2006; KNMI, 2019; de Graaf et al., 2016; Sihler et al., 2017). Based on this difference of spatial resolution, it becomes evident that the probability to obtain a completely cloud-free observation over a certain location is much higher for imagers than for spectrometers (Krijger et al., 2007). Therefore, background-reflectance retrieving algorithms applicable to spectrometer data have to deal with much sparser statistics than those tailored for imagers.

Algorithms deriving cloud fractions from spectroscopic measurements feature different approaches for background maps. One of the first algorithms published for the Global Ozone Monitoring Instrument (GOME, Burrows et al., 1999) is Initial Cloud Fitting Algorithm (ICFA) applying the global digital elevation model ETOPO5 (Kuze and Chance, 1994; Tuinder et al., 2004). Seven years later, Koelemeijer et al. (2001) published the fast method for retrieval of cloud parameters using oxygen A-band measurements (FRESCO), whose development continued as FRESCO+ (Fournier et al., 2006; Wang et al., 2008; TEMIS, 2019). Operational algorithms, like FRESCO+, apply background maps from auxiliary instruments in order to provide data directly after launch. The algorithms are then usually updated implementing different background maps as the mission evolves. The background maps may be either supplied from spectrometers or images. FRESCO version 6, for example, applies imager data from the Medium Resolution Imaging Spectrometer (MERIS) resolving albedo gradients, e. g. over coastlines, much better than the spectrometer data (Popp et al., 2011). Kokhanovsky et al. (2009), on the other hand, interprets MERIS data using threshold techniques to derive cloud fractions for Scanning Imaging Absorption Spectrometer for Atmospheric Cartography (SCIAMACHY, Bovensmann et al., 1999). FRESCO version 7 features another approach for the Global Ozone Monitoring Experiment 2 (GOME-2, Callies et al., 2000; Munro et al., 2006, 2016) applying Lambertian Equivalent Reflectance (LER) maps derived from GOME-2 itself (TEMIS, 2019; Tilstra et al., 2017b). Version 8 of FRESCO then applies a LER climatology derived from GOME-2 data taking the viewing geometry into account. In contrast to FRESCO, Optical Cloud Recognition Algorithm (OCRA) applies background maps derived in the RGB color space (Loyola, 1998; Loyola et al., 2007; Lutz et al., 2016). In its third version developed for GOME-2 and the Tropospheric Monitoring Instrument (TROPOMI, Veefkind et al., 2012), OCRA also accounts for degradation, viewing geometry dependence, and sun glint (Lutz et al., 2016). For OMI, the first version of OMCLDO2 (Stammes et al., 2008) uses albedo data from GOME (Koelemeijer et al., 2003) and TOMS (Herman and Celarier, 1997) for calculating effective cloud fractions, whereas OMCLDO2 version 2 (Veefkind et al., 2016) applies a





LER database derived from OMI measurements themselves as published by (Kleipool et al., 2008). It is preferred to apply background maps from the same sensor for cloud retrievals – like in (Grzegorski et al., 2006), (Veefkind et al., 2016), and (Tilstra et al., 2017b) – in order to achieve higher CF accuracy especially at small CF because radiometric properties and their dependence on viewing angles vary between sensors (e. g. Tilstra et al., 2012). This approach is especially suitable for scientific

studies using data processed offline. However, operational CF processors require background obtained from auxiliary sensors because data from the same sensor is then not yet available.

Recent developments also focus on applying geometry dependent background maps in CF algorithms. HICRU empirically derives lower thresholds for each of the three subpixels of GOME separately (Grzegorski et al., 2006). For instruments featuring wider swaths like OMI and GOME-2, the limitations of the LER surface model become more important (Vasilkov et al., 2017;

Lorente et al., 2017, 2018). Lorente et al. (2018) find biases in cloud fractions of up to 50% between backward-scattering and forward-scattering geometries in the GOME-2 FRESCO and 26% in the OMI OMCLDO2 cloud algorithms. Vasilkov et al. (2017) show that applying a geometry dependent LER instead of a regular LER can lead up to a 50% increase of the trace-gas column density over polluted areas. Furthermore, Vasilkov et al. (2018) compared CF derived using geometry dependent LER with those based on regular LER and found CF differences of up to 0.07, especially for small CF. These absolute differences

correspond to relative errors that have a significant impact on trace gas retrievals. Aerosols, however, reduce the effect of the BRDF (Noguchi et al., 2015). These recent studies rely on BRDF information derived from Moderate Resolution Imaging Spectroradiometer (MODIS, Justice et al., 1998) measurements because similar information from spectrometers is yet too sparse to derive all coefficients of, for example, the Ross-Li BRDF model (Wanner et al., 1995).

This paper presents a new method to derive a geometry dependent lower threshold map from spectroscopic measurements.

The Mainz iterative cloud retrieval utilities (MICRU) apply an empirical parametrisation of the geometry dependence in order to overcome the limitation of having sparse data per geolocation. MICRU derives the lower threshold as LER in contrast to its heritage algorithm HICRU, which applies TOA reflectances directly. Hence, first order atmospheric effects are accounted for. Remaining dependencies on the viewing angle, which may be either instrumental artefacts or physical, are modelled by a combination of a second order polynomial and a reduced model for surface effects of land and sun glitter over ocean (Cox and

Munk, 1954a; Harmel and Chami, 2013; Martin et al., 2016). It is noted that the idea of modelling the viewing angle dependence using a second order polynomial is not new. For example, Várnai and Marshak (2007) used a second order polynomial to model the mean optical thickness of inhomogeneous clouds in MODIS measurements.

In this study, we apply MICRU to GOME-2 data exemplarily. Unlike its heritage algorithm, HICRU, MICRU is applicable to almost arbitrary wavelength intervals in the UV/vis wavelength region. MICRU furthermore uses an RT model to reduce

the influence of atmospheric scattering for the retrieval of $R_{\mathrm{min}}$. MICRU CF are evaluated between 382 and 757.5 nm in order to investigate the spectral stability of the algorithm, the influence of the surface on CF accuracy, and the influence of spatial aliasing specific to GOME/GOME-2 instruments (EUMETSAT, 2015).

The MICRU algorithm has been developed as part of the cloud fraction verification activities for the TROPOMI/S-5P (Veefkind et al., 2012) and Sentinel 5 satellite missions. The operational cloud fraction algorithms for these missions are



OCRA (Lutz et al., 2016) and FRESCO (Wang et al., 2008), respectively. A comparison between all three algorithms is performed in Sect. 3, after introducing MICRU in Sect. 2.





## 2 Method

MICRU is designed to be applicable to UV/vis satellite sensors operating on sun-synchronous orbits. In this study, we examine its applicability using GOME-2 data. This section first describes the required input data (sec. 2.1) and the conversion between TOA reflectances $R$ to LER $T$ (Sect. 2.2). Section 2.3 then details the retrieval of $T_{\min}$ maps. The calculation of $R_{\max}$ is described in Sect. 2.4. Section 2.5 specifies the implementation of the MICRU algorithm, followed by a description of data sets MICRU results are compared to (Sect. 2.6).

### 2.1 Input data

#### 2.1.1 GOME-2 data

The primary data used in this work are radiances measured by the GOME-2 instrument onboard MetOp satellites. There are three essentially identical MetOp satellites in total: MetOp-A was launched in 2006 followed by MetOp-B and MetOp-C in 2012 and 2018, respectively. The MetOp satellites fly in a sun-synchronous orbit with an equator crossing-time around 9:30 solar local time (Munro et al., 2016). This study applies data from GOME-2A only as it features the longest uninterrupted time-series at the time of the MICRU algorithm development.

GOME-2 has four spectral main science channels (MSCs) with a spectral resolution between 0.26 and 0.51 nm ranging between 240 and 790 nm. Each MSC band features 1024 spectral channels. This study uses GOME-2A MSC data collected between February 2007 and June 2013. Data before and after this period are discarded in order to avoid interferences from instrument startup and a change of the operational swath width, respectively (EUMETSAT, 2015). Furthermore, GOME-2 has two polarization measurement devices (PMDs) covering a similar spectral range but at a much coarser spectral resolution: PMD-PP and PMD-SP measure the polarised intensity parallel and perpendicular to the slit of the spectrometer, respectively. Lang (2010) defines 15 discrete wavelength intervals for each PMD instrument. On 11 March 2008, the PMD band definitions of GOME-2A were updated to version 3.1 (EUMETSAT, 2015; Munro et al., 2016). Therefore, PMD data obtained before April 2008 are disregarded here in order to achieve a consistent PMD data set. All spectral data is contained in the level 1b (L1b) data provided by EUMETSAT.

GOME-2 is a scanning spectrometer featuring a nominally 1920 km wide swath, which was reduced to 960 km in July 2013 (Munro et al., 2016). One nominal swath consists of 24 MSC or 192 PMD pixels, respectively. At nadir, the nominal MSC pixel size is 40 km × 80 km in along- and across-track direction, respectively. The PMDs feature an eight times higher acquisition frequency leading to a smaller pixel size of 40 km × 10 km. The illumination-observation geometry is defined by the solar zenith angle (SZA) $\theta_0$, the viewing zenith angle (VZA) $\theta$ and the relative azimuth angle (RAA) $\phi$ as sketched in Fig. 1. The angles are defined at the pixel center at the surface. It is noted that the along-track pixel size increases with increasing VZA due to the Earth's curvature. Hence, the pixel shape becomes trapezoidal (de Graaf et al., 2016; Sihler et al., 2017).

In the spectral domain, the MICRU algorithm is applied to 14 MSC and 16 PMD channels in order to assess the influence of systematic differences on the accuracy of $c$. In principle, the radiometric input required by MICRU may be integrated along any spectral interval, but it is beneficial to avoid significant absorption structures in order to minimise the influence of





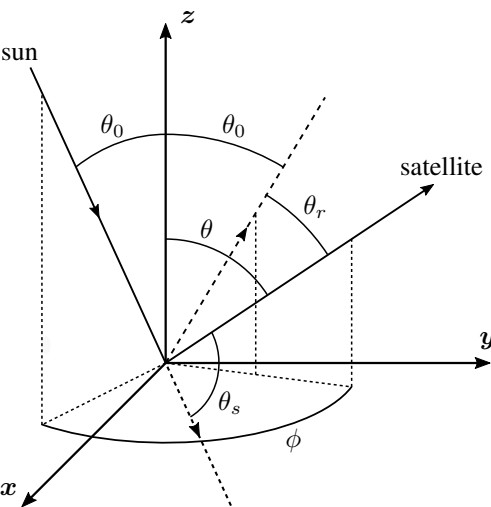

**Figure 1.** Angle definition at the surface: solar zenith angle $\theta_0$, viewing zenith angle $\theta$ and relative azimuth angle $\phi$, scattering angle $\theta_s$, and reflected sun angle $\theta_r$. Zenith is towards $\boldsymbol{z}$.

atmospheric absorptions. A dependence on the accuracy of the spectral calibration, which may not be optimal, is reduced by avoiding narrowband absorption features and Fraunhofer lines. Furthermore, broadband absorption by molecules and aerosols interferes with the inversion of $T$ from measured $R$. Interferences may not be avoided completely in the UV/vis, but MICRU MSC channels are defined minimising interferences from broad- and narrowband spectral features caused by Fraunhofer lines,

inelastic Raman scattering (Grainger and Ring, 1962; Solomon et al., 1987, Ring effect), and molecular absorption by $H_2O$, $O_2$, and $O_4$. The TOA reflectance $R_k$ of MSC channel $k$ is derived from the measured spectrum $R(\lambda)$ by applying

$$R_k = R(\lambda) * K_k(\lambda) \tag{3}$$

where $K_k$ is the convolution kernel of channel $k$. $K_k$ are either Gaussian or boxcar convolution kernels with different widths as listed in Table 1 and depicted in Fig. 2. The MICRU PMD channels as listed in Table 2 are selected from predefined PMD

bands (Lang, 2010).

The convolution kernels for 14 MICRU MSC channels are manually defined in the range between 374 and 758 nm. Hence, CF results are available for a variety of spectral ranges with different atmospheric trace gas absorptions. This is particularly important to improve collocation between CF and trace gas measurement featuring different spatial sensitivities caused by spatial aliasing (EUMETSAT, 2015; Munro et al., 2016). Spatial aliasing is caused by the sequential detector readout in

connection with the movement of the GOME-2 scanning mirror (Sihler et al., 2017). The comparison of MICRU results from MSC channels 2,5, 10, and 11 allows the investigation of spatial aliasing. These channels furthermore allow to compare the effect of spatial with spectral aliasing, which is due to differences in the spectral response of different channels. The horizontal arrows in Fig. 3 indicate MSC channels 2 and 5 matching the spectral sensitivity of the two PMD-PP/SP channel pairs 1/9 and 4/12, respectively. The vertical arrows indicate MICRU channels featuring the same acquisition time, and, hence, minimizing





**Table 1.** Definition of MICRU MSC channels $k$ with spectral convolution kernels $K$ centred at $\overline{\lambda}$ and width $w$. The kernels are either Gaussian (denoted G) or boxcar (denoted b) shaped. According to the spatial aliasing of GOME-2 (Fig. 3), channels 1, 4, 5, and 10 apply the same readout time as corresponding MSC/PMD channels in different bands.

| Channel $k$ | Spectral convolution kernel centre | width | shape | MSC band | RT wavelength | Comment |
|---|---|---|---|---|---|---|
| 1 | 374.96 nm | 1.00 nm | G | 2B | 375.0 nm | same timing as channel 9 |
| 2 | 381.97 nm | 3.57 nm | b | 2B | 382.0 nm | range of PMD-PP band 6 |
| 3 | 388.00 nm | 1.00 nm | G | 2B | 388.0 nm | KNMI GOME-2 LER @ 388 nm |
| 4 | 389.68 nm | 1.00 nm | G | 2B | 389.7 nm | same timing as channel 14 |
| 5 | 424.52 nm | 0.64 nm | b | 3 | 424.5 nm | timing of PMD-PP band 6 |
| 6 | 425.00 nm | 1.00 nm | G | 3 | 425.0 nm | KNMI GOME-2 LER @ 425 nm |
| 7 | 433.40 nm | 1.00 nm | G | 3 | 433.4 nm | same timing as channel 14 |
| 8 | 440.00 nm | 1.00 nm | G | 3 | 440.0 nm | KNMI GOME-2 LER @ 440 nm |
| 9 | 460.00 nm | 1.00 nm | G | 3 | 460.0 nm | range of NO2 retrieval |
| 10 | 516.67 nm | 3.52 nm | b | 3 | 519.0 nm | timing of PMD-PP band 9 |
| 11 | 521.77 nm | 53.98 nm | b | 3 | 521.8 nm | range of PMD-PP band 9 |
| 12 | 670.00 nm | 1.00 nm | G | 4 | 670.0 nm | KNMI GOME-2 LER @ 670 nm |
| 13 | 680.00 nm | 1.00 nm | G | 4 | 680.0 nm | short of red edge and $O_2$-B band |
| 14 | 757.50 nm | 0.75 nm | G | 4 | 757.5 nm | short of $O_2$-A band (FRESCO) |

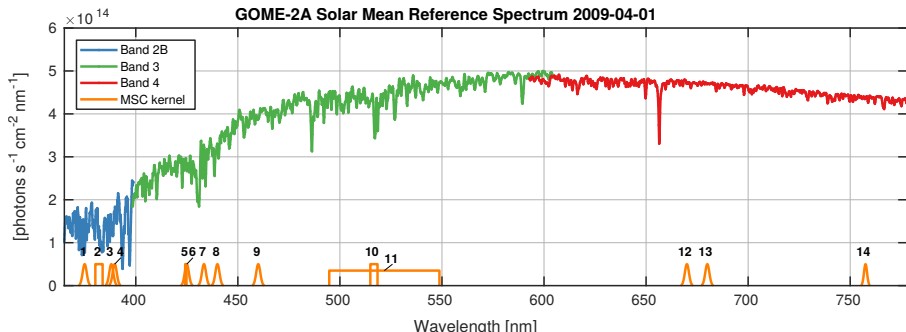

**Figure 2.** Mean solar spectrum of GOME-2A MSC channels 2B, 3, and 4 recorded on 1 April 2009. Spectral convolution kernels $K_k$ from Table 1 are plotted in orange. Note small biases between overlapping bands due to different calibrations.

the spatial aliasing between them: MSC channels 2 and 10 correspond to PMD channels 1/9 and 4/12; MSC data acquired at the same time but in different bands are sampled by MSC channels 1 and 9 in bands 2B and 3, respectively, and MSC channels





**Table 2.** Definition of MICRU PMD channels for PP and SP polarisation, respectively. PMD band definitions are compiled in Sect. 5.1.5 of (EUMETSAT, 2015) or Table 5 by Munro et al. (2016).

| Channel PP | SP | PMD band | approximate w/l range | RT wavelength |
|---|---|---|---|---|
| 1 | 9 | 6 | 380 … 384 nm | 382 nm |
| 2 | 10 | 7 | 400 … 429 nm | 413 nm |
| 3 | 11 | 8 | 434 … 492 nm | 460 nm |
| 4 | 12 | 9 | 495 … 549 nm | 519 nm |
| 5 | 13 | 10 | 553 … 556 nm | 554 nm |
| 6 | 14 | 11 | 568 … 613 nm | 589 nm |
| 7 | 15 | 12 | 618 … 662 nm | 639 nm |
| 8 | 16 | 14 | 794 … 804 nm | 799 nm |

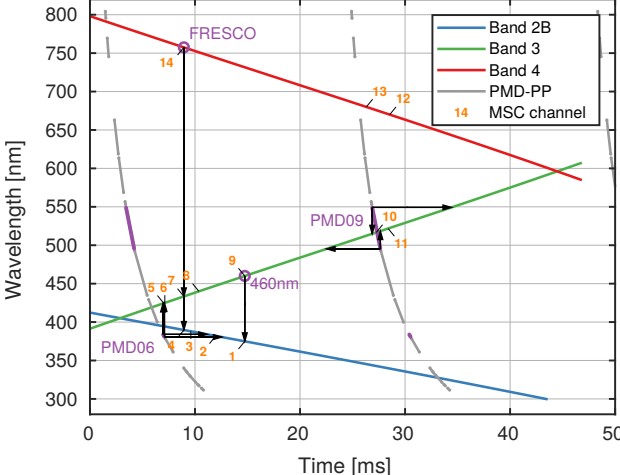

**Figure 3.** Spatial aliasing of GOME-2 for MSC and PMD bands in time-wavelength space. Radiometric and spatial correlation is expected maximal at the respective intersections. Note that the PMD read-out is faster (steep slope of gray lines) and at higher frequency compared to the MSCs. The second PMD readout ($m=1$) begins after 23.4375 ms (EUMETSAT, 2015). Horizontal and vertical black arrows indicate the spectral and temporal mapping between GOME-2 bands respectively.

4, 7, and 14 in bands 2B, 3, and 4, respectively. The correlation of MICRU CF depending on spatial alignment is investigated in Sect. 3.3.3.





**Table 3.** List of external data sets applied in MICRU (see text for details).

| Name | Symbol | Reference(s) | Native data resolution | | Comment |
| --- | --- | --- | --- | --- | --- |
| | | | temporal | spatial | |
| Land sea mask | LSM | Wessel and Smith (1996), NOAA (2018) | — | $\approx 178\,\text{m}$ | GSHHG coast line database, rev. 679 |
| Surface elevation | $h$ | USGS (1996) | — | 30 arcsec | GTOPO30, U.S. Geological Survey |
| Snow concentration | — | Hall and Riggs (2016) | 8 d | 0.05° | MODIS Terra level 3 (MOD10C2) |
| Sea ice | — | Cavalieri et al. (1996) | 1 d | 25 km | compiled passive microwave data |
| Surface wind speed | $W$ | Dee et al. (2011) | 6 h | 1° | ERA Interim (ECMWF) |
| Absorbing aerosol index | AAI | de Graaf et al. (2018) | GOME-2 pixels | | GOME-2A level 2, version 1.01 |

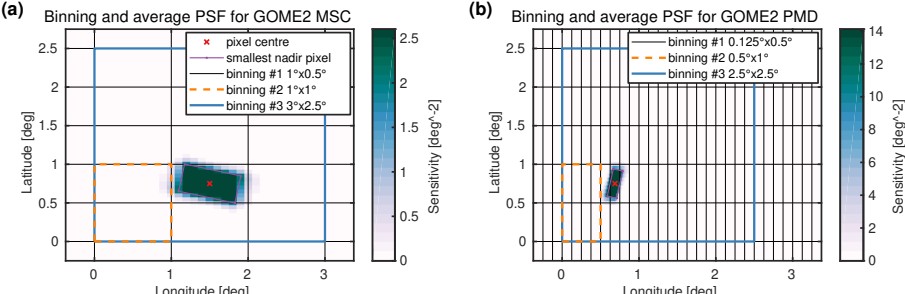

**Figure 4.** Illustration of average MSC (a) and PMD (b) point spread functions (PSF) used as convolution kernels for LSM, elevation, sea-ice and snow-concentration maps. The PSF correspond to a nominal GOME-2 swath width of 1920 km. The native MICRU $T_{\text{min}}$ resolutions for MSC and PMD evaluation (denoted: binning #1) are $0.1° \times 0.05°$ and $0.0125° \times 0.05°$, respectively (cf. Table 6).

### 2.1.2 Auxiliary data for MICRU

The MICRU algorithm requires several external data sets listed in Table 3. Two different strategies of co-locating these data to the GOME-2 observations are applied depending on the spatial resolution. Data provided at spatial resolutions significantly higher than the generic GOME-2 resolution are convolved with the respective average MSC and PMD point spread function

5 (PSF) as depicted in Fig. 4. The average PSFs are derived from all forward scan pixel edges from one orbit of GOME-2A data at latitudes lower than 55°. This approach simplifies the interpolation of auxiliary data on GOME-2 observations because a linear interpolation can be performed based on the GOME-2 pixel center alone while providing still sufficient spatial accuracy. Data provided at coarser resolutions than GOME-2 are linearly interpolated based on the GOME-2 pixel center without prior convolution.

10 One of the main features of MICRU is the separate parametrisation for measurements over land and ocean, respectively. Therefore, an accurate discrimination between land and ocean is crucial for the accuracy of MICRU at coasts. The land sea





mask (LSM) is compiled from revision 679 of the GSHHG coast line database (Wessel and Smith, 1996; NOAA, 2018) whose polygon data is first sampled at $0.1° \times 0.05°$ and $0.0125° \times 0.05°$ for MSC and PMD, respectively, and then convolved with the corresponding PSF (cf. Figure 4). The LSM is processed at eight times higher longitudinal resolution for PMD compared to MSC taking advantage of the smaller PMD pixel size. The LSM is compiled from *intermediate* GSHHG resolution neglecting

polygons smaller in area than one GOME-2 pixel.

The second important input is the surface elevation $h$ required for the inversion of $T$ (Sect. 2.2). Elevations maps are inferred based on GTOPO30 raw data, which are averaged on a $0.05° \times 0.05°$ grid, and convolved with a mean PSF sampled on the same grid resolution. Interpolation to GOME-2 resolution is again performed applying nearest neighbour interpolation.

Snow and sea ice data are queried to flag possible interferences from highly reflecting surfaces during post-processing

(Sect. 2.5.3). Snow data is imported from MODIS Terra measurements with a similar equator-crossing time of 10:30 in descending node similar to GOME-2. Hence, possible effects of different orbital parameters are supposed to be reduced. We used the 8-day composite MOD10C2 product (Hall and Riggs, 2016). Spatio-temporal interpolation uses the nearest neighbour method based on spatially convolved 8-day maps as described above.

Sea ice data is provided by the National Snow & Ice Data Center (NSIDC) and integrates micro-wave measurements from

different sensors (Cavalieri et al., 1996). It has a native resolution of $25\,\text{km} \times 25\,\text{km}$, which is convolved using average PSFs on $0.1°$ resolution before merging to GOME-2. Data voids at the poles are filled with values from the nearest valid latitude. Unfortunately, there is no information on the sea ice concentration close to shores in the applied data set. This limitation leads to interferences at shores at high latitudes because GOME-2 pixels possibly affected by sea ice may not be filtered a-priori.

Information on wind speed for the calculation of contributions from sun glitter is extracted from ERA-Interim data provided

by the European Centre for Medium-Range Weather Forecasts (ECMWF) (Dee et al., 2011). ECMWF 10 m-wind fields are used to parametrise sun glitter. As proposed by Ebuchi and Kizu (2002), the ECMWF wind fields are divided by a factor of 0.918 to approximate the wind speed at 41 ft ($\approx 12.5\,\text{m}$) above the surface, which is required as input for the sun glitter model by Cox and Munk (1954a). This factor corresponds to a drag coefficient of 0.0015 assuming neutral stratification (Ebuchi and Kizu, 2002). ECMWF data is imported at $1°$ and $6\,\text{h}$ spatial and temporal resolution, respectively. Spatially, nearest neighbour

interpolation is applied based on GOME-2 pixel centers.

Absorbing aerosol indices (AAI) are also used to mask measurements potentially biased by aerosol effects (AAI > 2). For this purpose, AAI data inferred from GOME-2 measurements at both MSC and PMD resolutions is used (de Graaf et al., 2018). Hence, no interpolation is required to merge MICRU and AAI data.

## 2.2   RT calculations and inversion of LER

The conversion between surface LER $T$ and measured TOA reflectance $R$ applies a look-up table (LUT) based on reduced reflectances $\hat{R} = I/E_0 = R\cos\theta_0/\pi$. The LUT entries are pre-computed using the SCIATRAN software version 3.7.1 (Rozanov et al., 2014; IFE-Bremen, 2018). The LUT has 5 dimensions: SZA, VZA , RAA, surface height $h$ and surface LER $T$. Table 4 compiles the LUT nodes as well as the wavelengths applied as described in Sect. 2.1.1. The LUT nodes in SZA and VZA direction are defined in reduced angles $\mu_0 = \cos\theta_0$ and $\mu = \cos\theta$, respectively, in order to provide more nodes at angles featuring





**Table 4.** Definition of LUT nodes: reduced SZA $\mu_0$, reduced VZA $\mu$, RAA $\phi$, surface height $h$, and surface LER $T$. The 5-D LUTs are calculated for 19 wavelengths $\lambda$.

| Parameter | Nodes |
|---|---|
| $\mu_0 = \cos\theta_0$ | 1.00, 0.975, 0.95, 0.90, 0.85, 0.80, 0.75, 0.70, 0.65, 0.60, 0.55, 0.50, 0.45, 0.40, 0.35, 0.30, 0.25, 0.20, 0.15, 0.10, 0.05 (21 total) |
| $\mu = \cos\theta$ | 1.000, 0.9875, 0.975, 0.950, 0.925, 0.900, 0.875, 0.850, 0.825, 0.800, 0.775, 0.750, 0.725, 0.700, 0.675, 0.650, 0.625, 0.600, 0.575, 0.550, 0.525, 0.500, 0.475, 0.450, 0.425, 0.400, 0.375, 0.350, 0.325 (29 total) |
| $\phi$ [°] | 0 . . . 180, in steps of 15 (13 total) |
| $h$ [km] | 0, 1.4, 3, 4.8, 7 (5 total) |
| $T$ | 0 . . . 1, in steps of 0.1 (11 total) |
| $\lambda$ [nm] | 375, 382, 388, 389.7, 413, 424.5, 425, 433.4, 440, 460, 519, 521.8, 554, 589, 639, 670, 681, 757.5, 799 (19 total) |

larger gradients. The linear interpolation between the nodes is performed in $\theta_0$ and $\theta$ space, respectively, in order to increase numerical stability at nadir.

The vector RT calculations are performed in spherical geometry based on a US standard atmosphere with 1013 hPa surface pressure at $h = 0$ m. The surface is treated as a Lambertian reflector. The model accounts for molecular absorption by $O_3$ and

$O_4$. The $O_3$ column is fixed to 250 Dobson Units (DU). Aerosols and Raman scattering are not included in the simulations.

From the LUT, the $\hat{R}(T)$ relation is interpolated for all observation geometries except for $h < 0$ km, which are tweaked to $h = 0$ km. $\hat{R}(T)$ is monotonic and, therefore, $T(\hat{R})$ can be readily inverted. We apply linear interpolation to infer $T(\hat{R})$.

## 2.3   $T_{\min}$ retrieval

The $T_{\min}$ MICRU algorithm requires a certain number of measurements in order to constrain its model parameters using obser-

vations not contaminated by clouds. For the description of the algorithm, we define a base set of measurements $\Omega_0$, which are spatially and temporally correlated. It is noted that $\Omega_0$ is a subset of all available measurements depending on grid resolution, measurement resolution, time period, surface structure, and cloud statistics. Section 2.5.1 describes the implementation of the subsetting process.

MICRU defines $T_{\min}$ depending on the measurement geometry ($\theta_0$, $\theta$, $\phi$), geolocation, and time $t$. $T_{\min}$ is not a true LER

because it contains geo-physical and instrumental information. This information is not separated within MICRU and will be treated simultaneously as the ultimate goal is to determine a parametrisation of $T_{\min}$ as accurately as possible. In general, it is not possible to parametrise $T_{\min}$ in full ($\theta_0$, $\theta$, $\phi$)-space due to the sun-synchronous orbit of GOME-2 (Sect. 2.1.1). At every latitude, the dependencies of SZA and RAA on both VZA and time repeat annually. The dependence of SZA and RAA on

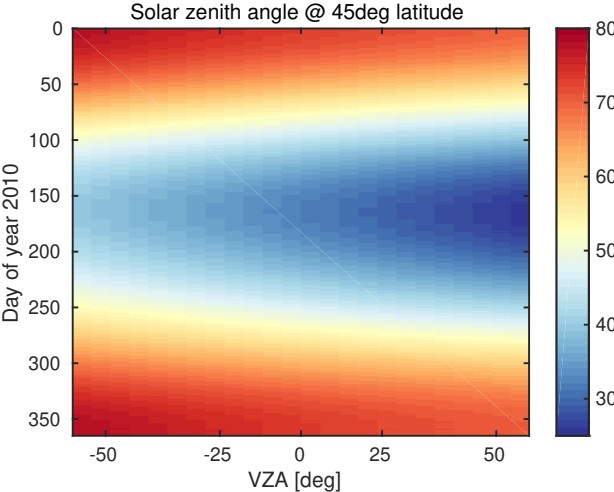

**Figure 5.** SZA depending on time and VZA at 45°N latitude. Data bins correspond to 24 discrete forward scan pixels and individual days in $x$- and $y$-direction, respectively. GOME-2A performs a sun-syncronous orbit at a fixed inclination to the sun. Hence, the SZA unambiguously depends on time, VZA, and latitude.

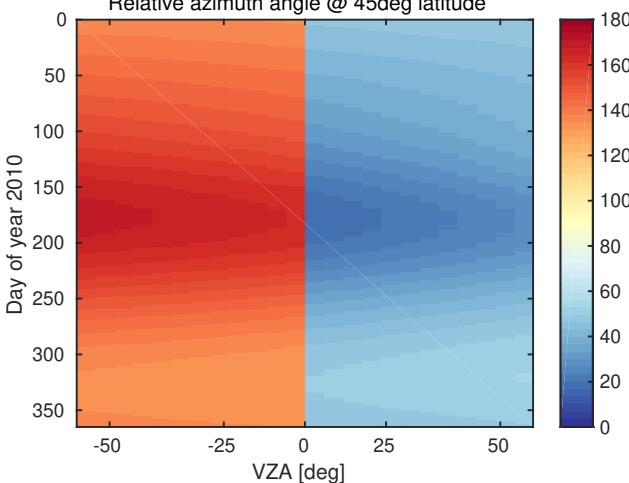

**Figure 6.** Same as Fig. 5, but showing the RAA at 45°N latitude.

VZA and time are exemplarily depicted for 45°N latitude in Figs. 5 and 6, respectively. It is therefore sufficient to parametrise the observation geometry (in each bin) by $\theta$ and $t$.

$\Omega_0$ typically contains a significant number of observations contaminated by clouds. Cloudy observations need to be filtered in order to retrieve a $T_{\mathrm{min}}$ parametrisation based on cloud-free observations. Therefore, an iterative filter algorithm to find the





lower accumulation point by Grzegorski et al. (2006) is presented in Sect. 2.3.2. Compared to HICRU, however, the MICRU algorithm generalizes from zero to four dimensions.

### 2.3.1 $T_{min}$ model

The MICRU $T_{min}$ model is

$$y(\hat{t}, \hat{\theta}, \theta_s, r_g) = a_0 + a_t\hat{t} + y_a(\hat{t}, \hat{\theta}) + a_s\cos\theta_s + a_g r_g \tag{4}$$

applying 4 independent variables $\hat{t}$, $\hat{\theta}$, $\theta_s$, and $r_g$ and 7 dependent variables $a_0$, $a_t$, $a_p$, $a_{a0}$, $a_{a1}$, $a_s$, and $a_g$, which are introduced in the following. Equation (4) is an empirical parametrisation accounting for actual and systematic effects of the lower threshold, which are not linearly independent in general.

Equation (4) applies normalized time $\hat{t}$ and normalized VZA $\hat{\theta}$ instead of $t$, and $\theta$ as independent model parameters to improve fit stability. Assuming $t$ measures in unit days, we then define the normalized time

$$\hat{t} = \frac{t - t_0}{365.25\,\text{days}} \tag{5}$$

measuring in unit years centred on $t_0$. In case of GOME-2A, $t_0$ is chosen so that $\hat{t} = 0$ for 1 January 2010. $\theta$ measured in unit degrees and the normalised VZA

$$\hat{\theta} = \theta/55° \tag{6}$$

ranges between -1 and 1.

The two further model parameters are the scattering angle $\theta_s$ defined by

$$\cos\theta_s = \sin\theta_0\sin\theta\cos\phi - \cos\theta\cos\theta_0 \tag{7}$$

and the reflected sun angle $\theta_r$ defined by

$$\cos\theta_r = \sin\theta_0\sin\theta\cos\phi + \cos\theta\cos\theta_0 \,. \tag{8}$$

Both angles are also illustrated in Fig. 1.

The right-hand side of Eq. (4) is the sum of the following terms.

1. The constant offset $a_0$ accounts for the mean surface LER.

2. Residual line-of-sight dependencies are modelled by a second order polynomial (Fig. 7), which is parametrised by the normalised apex angle $\hat{\theta}_a$ and curvature $a_p$:

$$y_a(\hat{\theta}) = a_p\left((\hat{\theta} - 2\hat{\theta}_a)\hat{\theta} + \hat{\theta}_a^2\right). \tag{9}$$



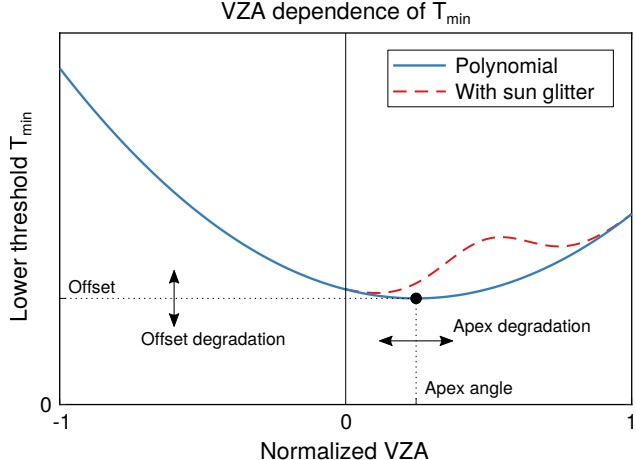

**Figure 7.** A second order polynomial parametrised by apex angle and curvature models systematic VZA dependencies of the lower threshold $T_{\min}$. Degradation may affect both apex angle and offset. The position, amplitude, and width of the sun glitter contribution depends on observation geometry and wind speed.

3. Temporal degradation is assumed by a linear offset degradation factor $a_t$ and the time dependent normalised apex angle

$$\hat{\theta}_a(\hat{t}) = a_{a0} + a_{a1}\hat{t} \tag{10}$$

as indicated by the arrows in Fig. 7. Tests applying a second order polynomial or an exponential to model degradation of GOME-2 MSC data were not successful. The former does not improve results significantly and the latter deteriorates the stability of the fit.

4. BRDF effects are modelled by an empiric $\cos\theta_s$ term. Its inverse shows a similar behaviour like the Li-dense kernel for closed canopy (Li and Strahler, 1992; Wanner et al., 1995) but does not require any further parameters. This term models the annual oscillations particularly visible at the western swath edge in Figs. 8(d,e) and 8(d,e).

The cosine normalises the parameter improving the fit stability. As a test, we replaced the empirical term with the precise Li-dense kernels and, secondly, with a reduced $\cos\theta_r$ term for surface effects. Both tests, however, resulted in inferior results. Furthermore, $\csc\theta_s$ and $\cos^2\theta_s$ terms were applied but both lead to slightly but systematically inferior results in a number of case studies.

5. The contribution of sun glitter on water surfaces is parametrised based on the isotropic sun-glitter model suggested by Cox and Munk (1954a, b). This model was found to be sufficiently accurate for MICRU and, according to Zhang and Wang (2010), performs reasonably well compared to competing models in their study. We apply the glitter reflectance $r_g$ as provided by Eqs. (1–4), (9), and (15) in (Zhang and Wang, 2010). According to Cox and Munk (1954a), the mean square slope of the clean surface is

$$\sigma^2 = 0.003 + 5.12 \times 10^{-3}W \tag{11}$$





**Table 5.** List of $T_{\min}$ model parameters with initial values ($\beta_0$) and parameter bounds of the constrained non-linear fit.

| Symbol | Comment | $\beta_0$ | Range |
|---|---|---|---|
| $a_0$ | offset | variable | $-\infty \ldots \infty$ |
| $a_t$ | offset degradation | 0 | $-0.02 \ldots 0.02^*$ |
| $a_p$ | polynomial amplitude | 0.02 | $0 \ldots 0.2$ |
| $a_{a0}$ | normalised apex angle | 0 | $-5 \ldots 10$ |
| $a_{a1}$ | apex degradation | 0.02 | $-0.5 \ldots 0.5^*$ |
| $a_s$ | scattering amplitude | 0 | $-0.5 \ldots 0.2$ |
| $a_g$ | glitter amplitude | 0.2 | $0 \ldots 0.4$ |

$*$: Degradation constrained to 0 for MSC channels 12, 13, and 14.

where $W$ is the wind speed at 41 ft ($\approx 12.5\,\text{m}$) above sea level, which is computed from $10\,\text{m}$-wind speeds as described in Sect. 2.1.2. The index of refraction is set to $n$=1.34 (Blum et al., 2012). Hence, $r_g$ as illustrated in Fig. 7 is a function of $\theta_0$, $\theta$, $\phi$, and $W$.

Table 5 summarizes the $T_{\min}$ model parameters and the bounds of the constrained fit. Clearly, this model is a trade-off
between accuracy and stability. Choosing a model based on more parameters would increase the accuracy of modelling the physical effects but degrade the stability of the fit limited by the number of cloud-free observations.

### 2.3.2   Iterative surface fitting

MICRU applies an iterative threshold technique to retrieve the lower accumulation point $T_{\min}$. The method is similar to the threshold method applied by HICRU (Grzegorski et al., 2006), where $R_{\min}$ is assumed to only depend on time and viewing
direction. For HICRU, this dependency could be resolved manually because the number of discrete VZAs was small, and, therefore, the lower accumulation point could be efficiently retrieved in an image processing manner. MICRU, however, assumes a more complex behaviour of $T_{\min}$ and therefore incorporates a nonlinear least-squares fit in every iteration of the lower accumulation point determination algorithm.

The basic idea behind iterative surface fitting is that the lower envelope of measurement set $\Omega$ (blue dots in exemplary
Figs. 8(b) and 9(b)) is approximated by iteratively filtering measurement tuples $(\hat{t}, \hat{\theta}, \theta_s, r_g, T) \in \Omega$ fulfilling

$$T > y(\beta) + \tau \tag{12}$$

where $y(\beta)$ is the fit result and $\tau$ is a positive threshold. $\beta$ denotes the result vector of Eq. (4)

$$\beta = [a_0, a_t, a_p, a_{a0}, a_{a1}, a_s, a_g]. \tag{13}$$

The accuracy and the convergence of the method depend on the choice of $\tau$ (Grzegorski et al., 2006), the applicability of the
$T_{\min}$ model, the stability of the surface fit, and the initial values $\beta_0$. Compared to HICRU, MICRU introduces two improvements: (1) an adaptive scaling of $\tau$ reducing the number of a-priory assumptions, and (2) the application of a surface fit. The



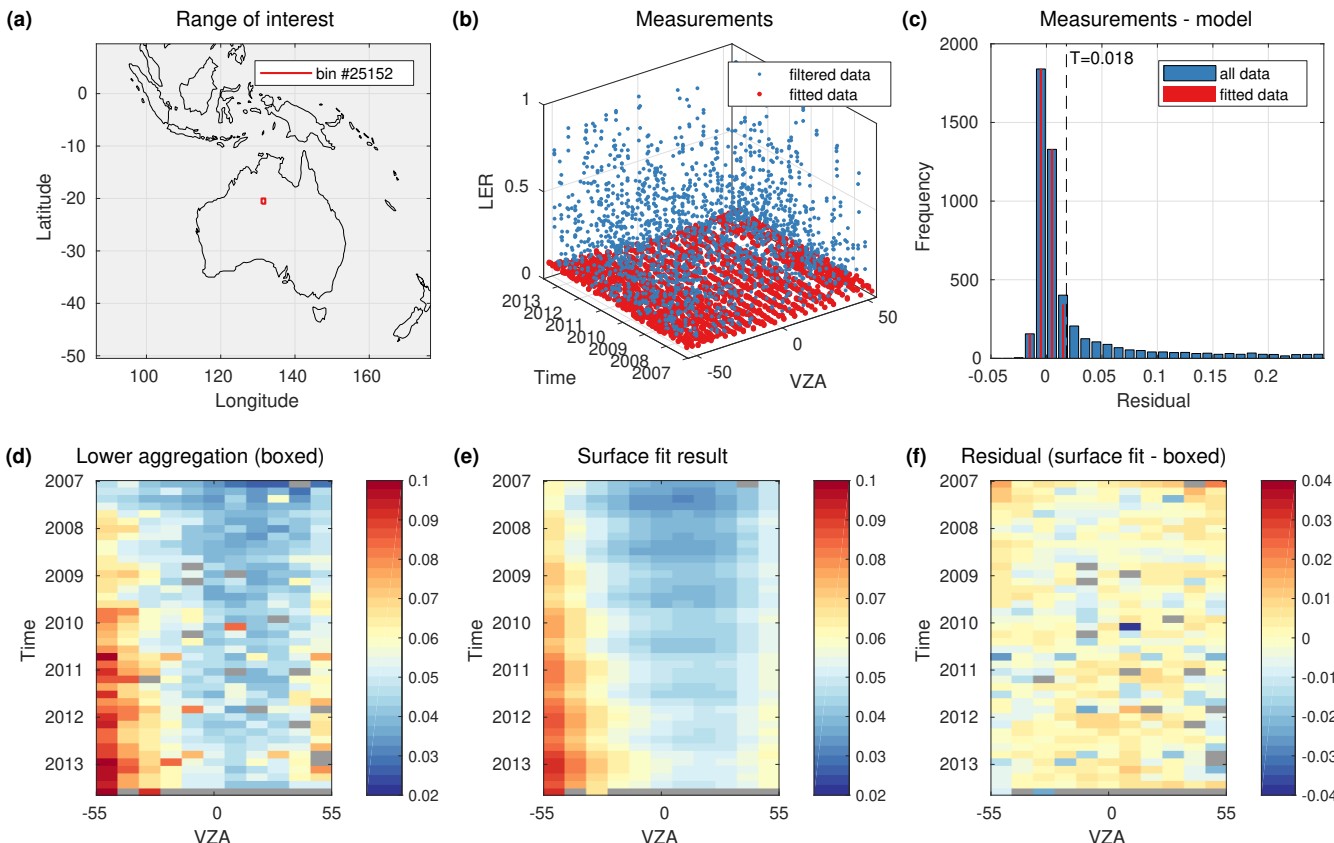

**Figure 8.** Example surface fit for MSC channel 2 at 382 nm and applying binning 2 over Australia: (a) The bin of interest ranges from 131°E to 132°E longitude and from 20°S to 21°S, (b) 3D representation of measurement set $\Omega_0$ (red) and finally fitted set $\Omega_I$ (blue), (c) frequency distribution of the fit residual, (d) lower accumulation point within each individual 0D-box, (e) surface fit result using model Eq. 4; and (f) difference of (d) and (e). The discretisation chosen for (d)–(f) displays a trade-off between accuracy and noise for the sake of clarity. Negative VZA denote the western half of the swath.

surface fit incorporates parameter constraints (Table 5) and therefore the trust-region-reflective algorithm (Coleman and Li, 1994, 1996) is applied.

The initialisation of the fit defines $\beta_0$ and an initial selection vector $V_0$. The selection vector $V_i$ defines the subset of measurements $\Omega_{i+1} = \Omega(V_i)$, on which the $(i+1)$-th iteration of the surface fit is applied. Table 5 provides initial values $\beta_0$ except for $a_0$, which is set to the median of $T(\Omega)$. $V_0$ is set true for all LER measurements fulfilling $T < a_0 + \sigma_0$, where $\sigma_0$ is the standard deviation of residual vector $\boldsymbol{R} = T - y(\beta_0)$. The initial threshold $\tau_0$ is set to 0.012.

The $i$-th iteration consists of the following steps: Fit Eq. (4) to $\Omega_i$ with initial guess $\beta_{i-1}$ yielding $\beta_i$. The residual vector is then $\boldsymbol{R} = T - y(\beta_i)$ defining the measurement set used in the next iteration through

$$\Omega_{i+1} = \{\Omega | \boldsymbol{R} < \tau \wedge \boldsymbol{R} > -3\sigma_{\boldsymbol{R}}\} \tag{14}$$



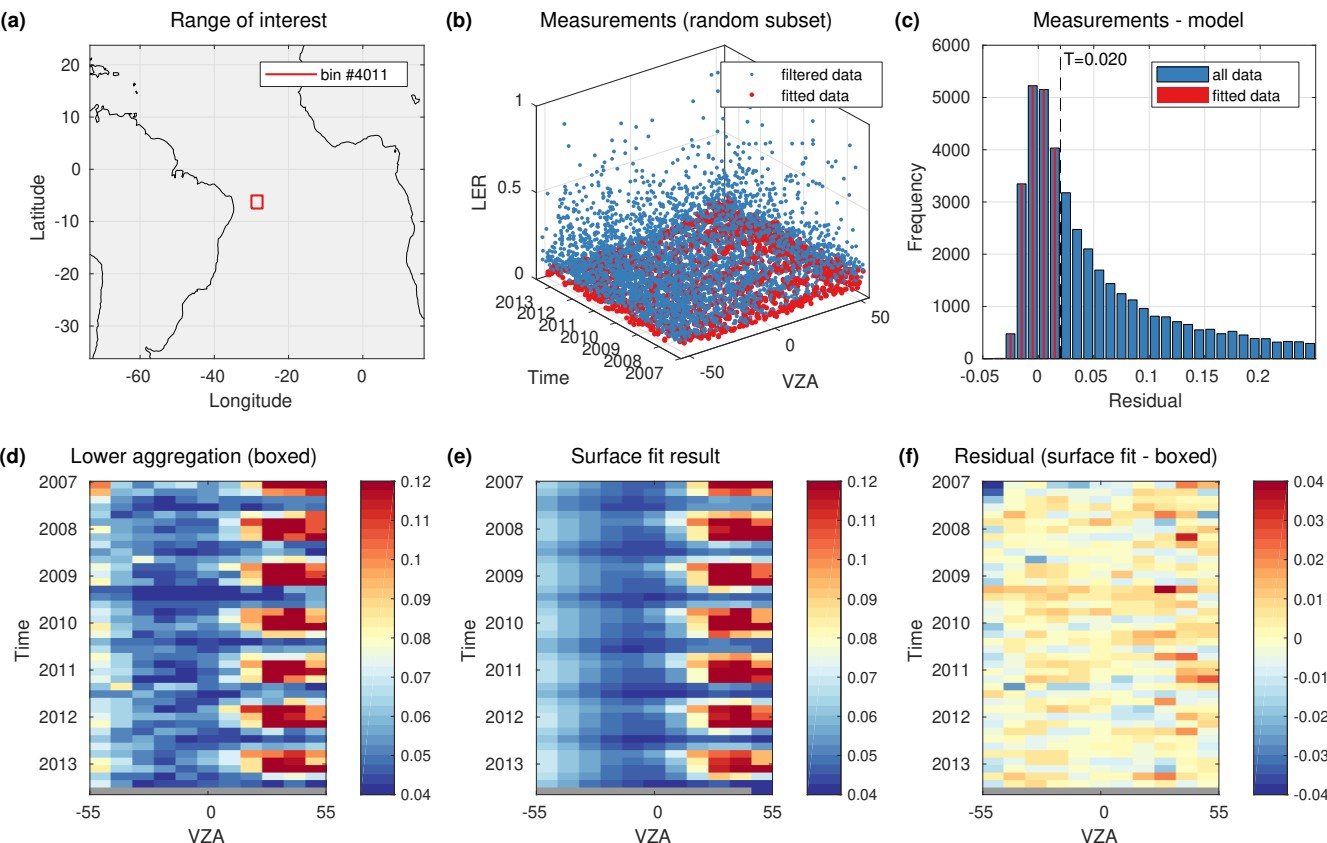

**Figure 9.** Same as Fig. 8, but for MSC channel 10 at 516.7 nm and applying binning 5 over the equatorial Atlantic. The set of measurements ranges from 30°W to 27°E longitude and from 7.55°S to 5°S.

where $\sigma_R$ is again the standard deviation of $\boldsymbol{R}$. The second condition in Eq. (14) filters outliers of the measurements distribution towards $-\infty$.

After that, the threshold for the next iteration is determined. $\tau_{i+1}$ depends on the retrieved mean $T$, that is the threshold becomes larger for brighter surfaces. The adjustment is retarded to steps of $\delta\tau = 0.002$. The upper limit $\tau_{\max}$ is defined. $\tau_{\max}$

5  increases linearly with $\overline{y(\beta_i)}$: $\tau_{\max} = \tau_0$ for $\overline{y(\beta_i)} = 0$ and $\tau_{\max} = 0.1$ for $\overline{y(\beta_i)} = 1$. Then, $\tau$ is increased if

$$\tau_i < \tau_{\max} \Rightarrow \tau_{i+1} = \tau_i + \delta\tau$$

or decreased if

$$\tau_i > \tau_0 \wedge \tau_i > \tau_{\max} + \delta\tau \Rightarrow \tau_{i+1} = \tau_i - \delta\tau.$$

Iterations terminate if at least one of the following four conditions is true: number of iteration steps exceeds $i_{\max} = 40$,

10  invariance of selection vector ($V_{i+1} = V_i$), invariance of result vector ($\beta_i = \beta_{i-1}$), or $\Omega_{i+1}$ includes less than 8 data points.





The result is $\beta = \beta_i$ corresponding to a remaining set $\Omega_i$ (red dots in Fig. 8(b)) defining $T_{\min}$ as a function of $\hat{t}$, $\hat{\theta}$, $\theta_s$, and $r_g$ in a specific geo-spatial bin. Diagnostic metrics for filtering (Sect. 2.5.3) are the number of elements in $\Omega_i$, its ratio to the number of elements in $\Omega_0$, and the number of iterations $i$.

## 2.4 Determination of $R_{\max}$

The upper threshold $R_{\max}$ is defined as the reflectance of a Lambertian surface with an albedo of 0.8 located at 7 km altitude. $R_{\max}$ is assumed independent of geolocation and time. A quantitative discussion of choosing $T_{\max} = 0.8$ as a cloud albedo for an Lambertian cloud model as upper threshold is provided by Stammes et al. (2008). As a consequence, however, very bright clouds exceeding $T_{\max} = 0.8$ will result in a MICRU CF $> 1$. It should be noted that potential instrumental degradation may introduce a systematic bias of the CF, which will be strongest for large CF. Most importantly for MICRU, the effect of this

simplification on the accuracy of small CF is negligible. Furthermore, $R_{\max}$ is calculated applying the look-up-tables described in Sect. 2.2 and Table 4.

## 2.5 Implementation

The MICRU algorithm consists of several consecutive steps: Import of data, RT simulation, merging of external data, determination of $R_{\min}$, determination of $R_{\max}$, and finally the computation of CF. The following subsections detail the implementation

of the methods described above in the MICRU framework.

### 2.5.1 Geospartial subsetting

The $T_{\min}$ algorithm (Sect. 2.3) requires a spatio-temporal subset of satellite measurements with a sufficient number of measurements. Section 2.1.2 describes how GOME-2 pixels are reduced to their centres. Hence, geospatial locations can be readily indexed and assigned to geospatial subsets. All MICRU computation refers to pixel centres rather than their actual area. This

simplification takes advantage of the fact that the surface is scanned by almost identically shaped ground pixels over the evaluated measurement period, and, therefore, measurements with identical pixel center are congruent.

Temporally, larger subsets should be favoured over smaller ones unless there are significant changes of surface properties or the instrument response degrades much differently than considered in the model (Sect. 2.3.1). Longer time-series increase the probability of including measurements not contaminated by clouds.

Spatially, a very small geospatial interval would be beneficial in order to increase the correlation between measurement and collocated $T_{\min}$ where the true surface is inhomogeneous. However, there is a trade-off because the probability of including enough cloud-free measurements decreases if the geospatial interval becomes too small. If there are not enough cloud-free measurements, the accuracy of the fit degrades. Furthermore, spatial subsampling can be avoided using spatial sampling intervals larger than the native resolution of the measurements defined by its PSF.

Within MICRU, the geospatial subsetting is called *binning*, which is performed on a longitude-latitude grid. Each binning corresponds to a global map at a different resolution. The $T_{\min}$ retrieval is independently applied on each binning, whose defini-





**Table 6.** Spacing of meridians ($\Delta\lambda$) and parallels ($\Delta\varphi$) for the definitions of tiling and binnings for MSC and PMD evaluations, respectively. Binning resolutions depend on surface type: land (s = 1) and ocean (s = 2).

|  | # | s | MSC $\Delta\lambda$ | MSC $\Delta\varphi$ | PMD $\Delta\lambda$ | PMD $\Delta\varphi$ |
|---|---|---|---|---|---|---|
| tiling |  |  | 45° | 15° | 15° | 5° |
| binning | 1 | 1 | 1° | 0.5° | 0.125° | 0.5° |
|  | 2 | 1 | 1° | 1° | 0.5° | 1° |
|  | 3 | 1 | 3° | 2.5° | 2.5° | 2.5° |
|  | 4 | 2 | 1° | 0.5° | 0.125° | 0.5° |
|  | 5 | 2 | 3° | 2.5° | 2.5° | 2.5° |
|  | 6 | 2 | 15° | 2.5° | 15° | 2.5° |
|  | 7 | 2 | 45° | 2.5° | 15° | 5° |

tion distinguishes between measurements over ocean and land. The results are then merged to form a complete parametrisations of $T_{\min}$ for ocean and land surfaces independently. There are several advantages of this approach.

1. If, for some reason, the fit fails using the highest resolution, the parametrisation results from evaluation using larger bins may be used instead.

2. The bin dimensions can be adapted to the surface type: smaller and approximately quadratic over land, larger and less depending on longitude over ocean.

3. It enables independent parametrisations for the two different surface types. Hence, $T_{\min}$ gradients at the coast can be mitigated.

Table 6 details and Fig. 4 illustrates the MICRU binnings for GOME-2 MSC and PMD evaluations, respectively. Figure 8(a)

illustrates the dimension and location of bin 25152 of MSC binning 2.

   The entire data set needs to be resorted with respect to geolocation instead of acquisition time for computational purposes. Therefore, input data is organised in geospatial *tiles* as defined in Table 6, which reduces the memory requirement for a process performing the iterative surface fitting per bin (Sect. 2.3.2). Tiling furthermore enables parallel processing of MICRU on a cluster because each sub-process only requires a small portion of the observational data. Hence, scaling MICRU to sensors

different from GOME-2 is straight-forward by adjusting the tile resolution.

### 2.5.2   $T_{\min}$ **maps**

The following filters are applied on the input data prior the $T_{\min}$-retrieval:

   – Filter measurement in ascending node to avoid ambiguities of the time and latitude dependent $\theta$-selection.





- Filter viewing modes other than the nominal 1920 km swath (EUMETSAT, 2015). For example, this filter excludes *nadir static* and *narrow swath* orbits as well as data recorded after June 2013.

- Filter SZA larger 85°.

- Filter data possibly affected by solar eclipses as defined in Appendix B of (Tilstra et al., 2017a).

- Filter times of instrumental malfunction as listed in (EUMETSAT, 2014).

- Filter measurements with $AAI > 2$.

- Filter measurements, which include neither $> 90\%$ land nor $> 90\%$ ocean.

- Over ocean, filter measurements with $\theta_r < 8°$ (sun glitter).

Then, the $T_{min}$-algorithm (Sect. 2.3) is applied on the measurement tuples within each bin. Figures 8 illustrate the $T_{min}$-
algorithm applied on MSC channel 2 for a $1° \times 1°$ bin over Australia. Similarly, Fig. 9 illustrates the same on MSC channel 10 data at 516.7 nm over the Atlantic Ocean. This step is repeated for all binnings listed in Table 6 and all channels listed in Tables 1 and 2 for MSC and PMD, respectively. For diagnostic purposes, the number of iterations $i$, the number of included measurements $N$, the number of fitted measurements $N^*$, threshold $\tau$, and residual statistics are intermediately stored alongside the fit result $\beta$ for diagnostic purposes.

Tiled $T_{min}$ results are stitched together to form global maps for all binnings and channels. For each channel, the maps of different resolutions need to be merged in order to obtain two complete and unambiguous $\beta$ maps, respectively one for ocean and one for land. The steps of the merging process are detailed in Appendix A. Appendix B presents exemplary results for MICRU channel 02.

### 2.5.3 CF calculation and flagging

Once $R_{min}$ and $R_{max}$ are determined, $c$ can be computed using Eq. (1) for all MICRU channels separately. The MICRU data set furthermore provides several quality flags listed in Table 7. MICRU MSC and PMD data are merged with three aliasing offsets $m = 0, 1, 2$ (Sect. 2.1.1) for the investigation of the spatial aliasing in Sect. 3.3.3.

### 2.6 Comparison data

For GOME-2 MSCs, FRESCO+ cloud fractions evaluated at the $O_2$-A band are probably the most commonly used CF product
(Wang et al., 2008; TEMIS, 2019). In this study, three different versions of the FRESCO cloud fractions are applied:

**FRESCO L1b** denotes the CF data shipped with the L1b files from EUMETSAT, also denoted FRESCO version 6. This FRESCO version applies a background map compiled from MERIS measurements over land and GOME-1 surface LER over ocean (Popp et al., 2011; Tilstra et al., 2017b).



**Table 7.** Flags applied by the MICRU algorithm allowing for individual filtering.

| Flag | Description |
| --- | --- |
| Coast warning | if > 10% land and >10% ocean |
| AAI warning | if $AAI > 2$ |
| Snow warning | if MODIS 8-day snow concentration $> 5\%$ |
| Sea ice warning | if micro-wave sea-ice concentration $> 5\%$ |
| Sunglint warning | if $\theta_r < 36°$ over ocean or $\theta_r < 8°$ over land |
| Sunglint risk | if $\theta_r < 8°$ over ocean |
| Statistics warning | if $N < 80$ or if the mode of $R > 0.1$ |
| Coarse mode | if the applied binning is neither 1 or 4 |
| Extrapolated parameters | if the applied parametrisation is extrapolated |

**FRESCO v7** is the first FRESCO version applying a background LER map derived from GOME-2 measurements themselves
(Tilstra et al., 2017b).

**FRESCO v8** is the most recent version applying a directional LER database compiled from GOME-2 measurements. The
resolution at the coast is increased to effects of both surface types within one GOME-2 pixel.

FRESCO L1b data of the entire evaluation period is included in this study. For FRESCO v7 and v8, however, comparisons
to MICRU are limited to selected months. For FRESCO v7, it is January, April, July, and October 2010. For FRESCO v8, it is
January to December 2010 plus April 2007 and 2013 (cf. Figs. 21(b) and (c)). The applied FRESCO products do not correct for
interferences with sun glitter. It is furthermore noted that MICRU data are ignored in the comparisons in Sect. 3.4 if respective
OCRA/FRESCO data are invalid.

MICRU PMD results are also compared to OCRA cloud fractions inferred from GOME-2 PMD measurements described by
Lutz et al. (2016). In its actual version 3.0, OCRA is mainly developed as the operational CF product for the S-5P/TROPOMI
mission. In contrast to FRESCO, OCRA applies an empirical correction scheme for the sun glint effect (Lutz et al., 2016).

    The selected cloud products define the upper threshold differently. FRESCO applies a Lambertian (or reflecting) cloud model
– like MICRU – and OCRA applies a volumetric (or scattering) cloud model (Wang et al., 2008; Lutz et al., 2016). Furthermore,
the treatment of extreme CF significantly differs between MICRU, FRESCO, and OCRA. Some FRESCO both and OCRA
provide normalised CF, which means that CF do not linearly scale with reflectance. OCRA, for example, normalises CF $< 0$ to
0 and CF $> 1$ to 1. For FRESCO, normalisation schemes defer between versions: FRESCO L1b sets CF $< 0$ to 0. For CF $> 1$,
all FRESCO version vary the cloud albedo to improve convergence. In contrary, MICRU does not apply any normalisation by
default leading to an unlimited CF distribution. We define *cropped* subsets of data: CF $< 0$ are set to 0 and CF $> 1$ are omitted
from statistical comparisons (Sect.3.4) in order to avoid systematic bias from different normalisation strategies. It shall be noted





that both normalisation and cropping lead to biased mean results. These biases propagate into trace gas retrievals if normalised CF data are applied.

Finally, measurements by the AVHRR/3 (Advanced Very High Resolution Radiometer version 3) instrument are applied as independent measurements for the detection of clouds from the MetOp satellite (Cracknell, 1997; NOAA, 2017; EUMETSAT, 2011). AVHRR is an imager with six spectral channels centred between 630 nm and 12 μm. The spatial sampling of GOME-2 and AVHRR is detailed by Sihler et al. (2017). In this study, AVHRR data of bands 1, 2, and 3a is applied to produce RGB false colour images. Furthermore, an artificial AVHRR cloudmask is constructed, where an AVHRR pixel is assumed cloudy if either the Albedo test, or the T4-T3 test, or the T4-T5 test indicates a cloudy scene (cf. EUMETSAT, 2011).



## 3 Results

This section starts off with results from the $T_{\min}$ retrieval (Sect. 3.1) and a comparison between MICRU, FRESCO, OCRA based on single, cloud-free swaths (Sect. 3.2). Subsequently, statistical ensembles are applied to intercompare MICRU CF results (Sect. 3.3), and to evaluate differences between the three CF algorithms (Sect. 3.4).

Studies on monthly statistics exclude data where

- SZA $\geq 84°$,

- latitudes $\geq 55°$,

- MODIS snow concentration $> 2\%$,

- AAI $> 2$,

– the sunglint risk flag is raised, or

- the coast warning is raised (exception: average maps)

in order to reduce interferences.

### 3.1   $T_{\min}$ retrieval

Figure 8 illustrates the input measurements and output results of the $T_{\min}$-retrieval (Sect. 2.3) applying MSC channel 2 at
382 nm and binning 2 over continental Australia. The blue dots in Fig. 8(b) denote the input data omitting scattering angle $\theta_s$ and glitter reflectance $r_g$ dimensions for the sake of clarity. Figure 8(d) shows a matrix of lower aggregation points of $T$ retrieved independently in discrete boxed defined in the $t/\theta$-plane. The lower aggregation points are retrieved with the same $\tau$ as resulting from the surface fit but without parametrisation. The matrix reveals an increasing trend with time and a significant VZA-dependence. The iterative surface fitting result using the same data (Fig. 8(e)) shows a similar but much smoother result
due to improved statistics by combining the information from all measurements and applying a parametrised surface model. Figure 8(f) shows the difference between boxed and fitted results indicating average deviations much smaller than 0.04, which is the targeted accuracy of MICRU CF. There are, however, small systematic deviations towards the edges indicating a slight overestimation of $T_{\min}$ at the beginning of the sensing period (2007) and at large VZA and slight underestimation at the end of the period (colder colors for 2013). The histogram of the residual $R$ of LER measurements and modelled $T_{\min}$ is plotted in
Fig. 8(c) using the respective colors as in Fig. 8(b). Measurements applied for the final iteration of the surface fit peak between -0.01 and 0, for which a final threshold of $\tau = 0.018$ is applied.

    Figure 9, in contrary to Fig. 8, illustrates the application to another MSC channel at 516.7 nm and surface type ocean. Compared to Australia, the fraction of fitted measurements is significantly lower due to a higher probability for clouds. The smaller fraction of fitted measurements is also indicated by a less pronounced peak in the histogram (Fig. 9(c)). There is a
significant contribution of sun glitter, which is visible by the annually appearing red areas at positive VZA in Figs. 9(d) and





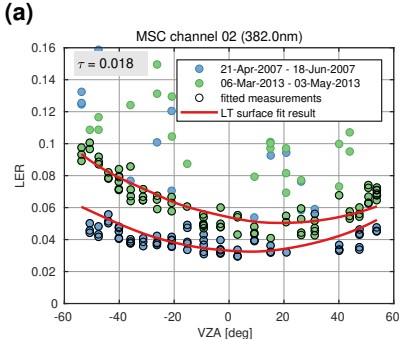 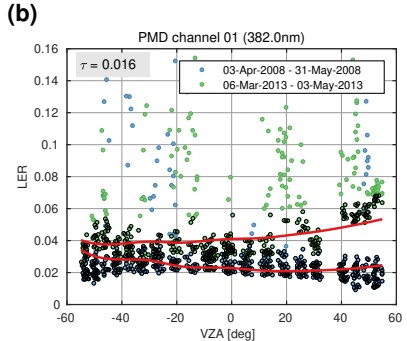 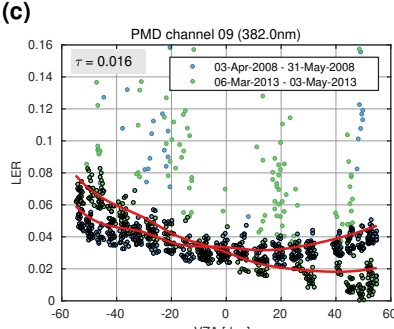

**Figure 10.** Detailed fit results from surface of three GOME-2 bands measured over Australia (cf. Fig. 8(a)), comparing the fit results with measurements selected from the beginning (blue dots) and the end (green dots): (a) MSC channel 02, the fit results (red lines) correspond to 8(f), (b) for PMD-PP, and (c) for PMD-SP. Note the significant differences between the residual VZA-dependences and their trends of three GOME-2 bands measuring at the same wavelength of 382 nm. Note that reflectance measurements below the red line will later result in negative CF.

(e). The scatter in Fig. 9(f) collocated with the regions affected by sun glitter is probably due to poor statistics when calculating the boxed comparison results. The CF in the first, western-most column of boxes seem to be biased low of the order of 0.01 compared to other viewing angles.

    Figure 10 investigates trends of the VZA-dependence for the three GOME-2 channels MSC, PMD-PP, and PMD-SP but
same spectral region, respectively. Figure 10(a) corresponds to the first and last row in Fig. 8(d/e), indicated by blue and green dots, respectively. Circled dots correspond to the red dots in Fig. 8(b). The red lines correspond to the fit results shown in 8(e). Figure 10(a) reveals a lower threshold increasing with time ($\approx 0.02$ in total) and a time-dependent VZA-dependence affecting apex VZA and curvature. Figures 10(b) and (c) illustrate the temporal behaviour of the corresponding PMD-PP and PMD-SP measurements. For PMD-PP, the VZA-dependence and its degradation is smallest compared to the other channels. PMD-PP,
however, features a significantly larger overall trend compared to PMD-SP in Fig. 10(c). For PMD-SP, the overall trend is small, but the VZA-dependence degrades significantly more than MSC channel 2.

## 3.2 Cloud-free observations

Figure 11 compares cloud-free measurements of MICRU, OCRA, and FRESCO over three exemplary sites featuring different surface cover (left to right): rainforest, continental mid-latitudes, and ocean. Independent AVHRR measurements are included
to identify essentially cloud-free scans (top row).

    MICRU MSC and PMD results show no significant cloud-cover where also AVHRR does not detect clouds. Purple colours at the swath edges, however, indicate biased low CF results (Figs. 11(d)–(i) in cloud-free observations over both land and ocean. Over land, the MICRU results are consistent with the AVHRR cloud mask, e. g. CF contributions of singular clouds smaller





than the GOME-2 pixel sizes are reproduced in Figs. 11(e) and (h) over North America. The bias of cloud-free observations is small and the colorbar range almost does not resolve the scatter around zero CF.

In case of sunglint, the situation is more complex. In Figs. 11(f) and (i), MICRU measurements scatter significantly as indicated within the area flagged with a sunglint warning (green edges). The scatter is only significant in the eastern swath

(orbit 17907) whereas the western swath (orbit 17908) shows almost no scatter. The analysis of the wind fields (not plotted) reveals that wind speeds vary between 0 and 5 m/s for the cloud-free region of the eastern swath and between 3 and 4 m/s in the western swath, respectively. The interpolation of the wind fields is apparently not accurate enough for the eastern swath. The interpolation is limited by a spatial resolution of $1°$ and a temporal resolution of 6 h. Hence, MICRU CF may be over- and underestimated in case of heterogeneous wind situations and especial for very low winds. We conclude, however, that MICRU

is capable to model moderate $R_{min}$ contributions by sun glitter. When sun glitter is large (in case of low winds), as indicated by yellow colours in the false colour background image of Fig. 11(c), this corrections is less reliable. Therefore, measurements flagged with a sunglint warning are screened from the statistical comparison studies below.

Figure 11 furthermore compares MICRU results to FRESCO and OCRA. Over land, the CF maps of FRESCO L1b and v7 measurements (Figs. 11(j), (k), (m), and (n)) reveal significant positive biases in the western part of the swath. Cloud fractions

larger than 20% are detected even though AVHRR and MICRU both detect no clouds. FRESCO v8 displays a significant improvement over Brazil (Fig. 11(p)), whereas CF over North America in Fig. 11(q) are still significantly biased in the west of the swath. Switching to OCRA, Fig. 11(s) reveals significantly smaller positive biases of OCRA over Brazil compared to FRESCO L1b and v7. Over North America (Fig. 11(t)), however, a positive bias and scatter are significant.

Over ocean (right column in Fig. 11), biases in regions possibly affected by sun glitter are obvious in the FRESCO and OCRA

data. FRESCO products are not correcting for this interference leading to systematic positive biases because the increased intensity is apparently interpreted as reflecting clouds (Figs. 11(l), (o), and (r)). In contrary to FRESCO, OCRA corrects for the sun glint effect in the centre of the affected region (Fig. 11(u)), but interferences still persist for $\theta_r < 36°$, which is flagged by MICRU as sunglint warning.





**Figure 11.** Detailed comparison between different cloud fraction products over cloud-free scenes over land and ocean: Brazil on 10 April 2010 (left column), North America on 8 April 2010 (second column), and the Indian Ocean on 2 April 2010. The rows display cloud information from different sensors/products (from top to bottom): AVHRR false colour RGB image indicating areas, where at least one AVHRR cloud test is positive (orange), MICRU MSC at 440 nm, MICRU PMD-PP at 460 nm, three FRESCO versions, and OCRA. The colour bar is defined to resolve cloud fractions particularly below 0.3 and negative values. Areas without data are indicated in grey. Images in the right column contain swath data of two orbits featuring different sun glitter scenarios due to different wind contitions (see text).

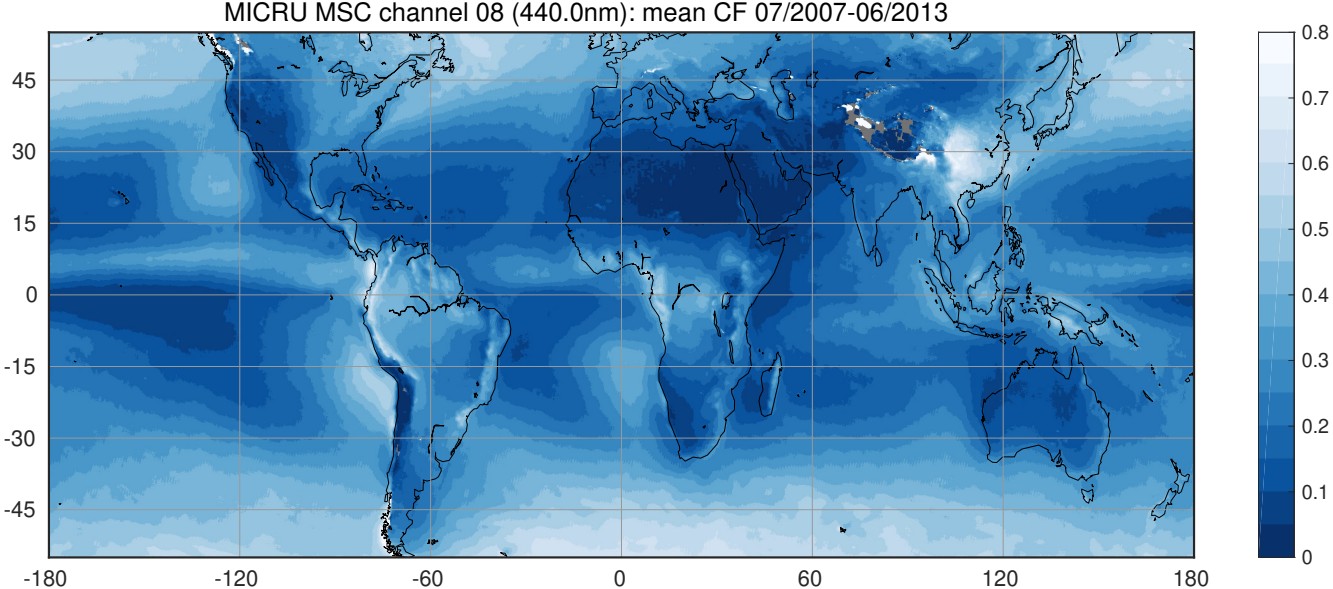

**Figure 12.** Six year average of MICRU MSC channel 8 cloud fraction measurements recorded between 1 July 2007 and 30 June 2013. Areas without data are plotted in gray. See Figs. E1 through E5 for comparison.

### 3.3 MICRU

#### 3.3.1 Global average cloud fraction

Figure 12 shows a global map of the average MICRU cloud fraction of six consecutive years using GOME-2A measurements. Therefore, it is a snapshot of the average CF at 9:30 in morning. The averaging period starts in July 2007 and ends in June 2013. The map clearly reveals a statistically increased CF at the Intertropical Convergence Zone (ITCZ), off the west coasts of continents in the sub-tropics, and the sub-polar oceans. The comparatively high average CF over China indicates a significant bias due to aerosol scattering. Similar plots for MICRU PMD, FRESCO and OCRA are described in Appendix E.

#### 3.3.2 MSC intercomparison

Figure 13 shows correlation density plots and bivariate fits (Cantrell, 2008) of MICRU CF for channel 2, 12, and 14 with respect to MICRU channel 8 (cf. Table 1).

The comparison between MICRU evaluations at 382 and 440 nm shows an almost perfect correlation of $r^2 = 0.997$ in Fig. 13(a). The slope reveals a minute positive bias. Figures 13(b) and (c) reveal significantly increased scatter for comparisons with longer wavelengths, which is dominated by increased heterogeneities of the land surface reflectance because the scatter is almost independent from wavelength over ocean (not shown). It is important to note that in Figs. 13(b) and (c) the scatter





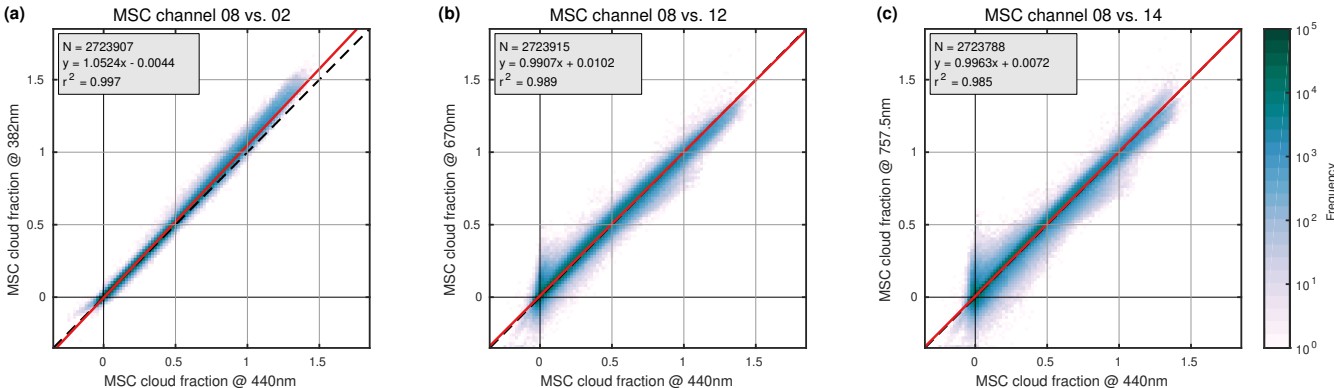

**Figure 13.** Comparison between MICRU MSC channel 8 at 440 nm and MSC channels (a) 2 at 382 nm, (b) 12 at 670 nm, and (c) 14 at 757.5 nm for April 2010. The plots correspond to the combinations indicated by circles in Fig. D1(a).

in $y$-direction for small $|c_x|$ is larger than the scatter in $x$-direction for small $|c_y|$. This indicates that the accuracy of CF is decreasing towards larger wavelengths.

Appendix D comprehensively compares results from MICRU channels and other cloud products.

### 3.3.3 MSC vs. PMD

5 Figure 14 shows CF comparison plots of two corresponding MSC/PMD evaluation. A spatial aliasing of $m = 0$ is chosen for the comparison at 382 nm in Fig. 14(a) and $m = 1\,519$ nm in Fig. 14(b), respectively. According to Fig. 3, these are the optimal choices of $m$, which is confirmed by the circled values in Fig. D2(a) indicating a significantly higher correlation between MSC and PMD channels compared to the results neighbouring to the left and right.

Furthermore, the matrix results in Figs. D1(b) and D2(a) can be used to compare the influence of spatial versus spectral 10 aliasing. Due to the different readout scheme (Fig. 3) both cannot be perfectly fulfilled simultaneously, either spectral or the spatial alignment between MSC and PMD can be achieved. In Fig. D1(b), the standard deviation of MSC measurements at 382 nm is slightly smaller (0.008) than for those at 424.5 nm (0.009). At 519 nm, however, the MSC values at 516.7 nm feature a smaller standard deviation (0.008) compared to those at 521.8 nm (0.009). Hence, spectral alignment seems favourable for PMD channel 1 at 382 nm whereas spatial alignment seems favourable for PMD channel 4 at 519 nm. The linear coefficients of 15 correlation in Fig. D2(a), however, indicate, that PMD channel 1 correlates slightly better to MSC channel 5 (0.998) compared to MSC channel 2 (0.997). Hence, spatial alignment is favourable if $r^2$ should be optimal. For PMD channel 4, the deviation of the respective $r^2$ values is $< 0.001$ (both 0.998). It shall be noted here that Fig. D2(a) also illustrates the importance for the correct choice of $m$. The correlation of PMD-PP at 519 nm with itself is significantly reduced from 1 to 0.994 if the value of $m$ deviates by $\pm 1$.

20 The coefficient of correlation between MSC channel 10 and PMD channel 4 ($m = 1$) of $r^2 = 0.998$ is optimal allowing a direct comparison and the investigation of the accuracy of small cloud fractions in the MSC product assuming that zero CF



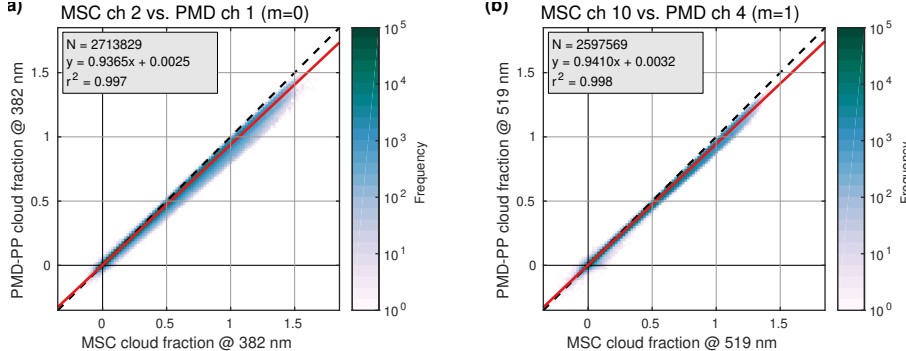

**Figure 14.** Comparison between MICRU MSC and MICRU PMD cloud fractions for April 2010: (a) at 382 nm and (b) at 519 nm. The plots correspond to the combinations indicated by circles in Fig. D2(a).

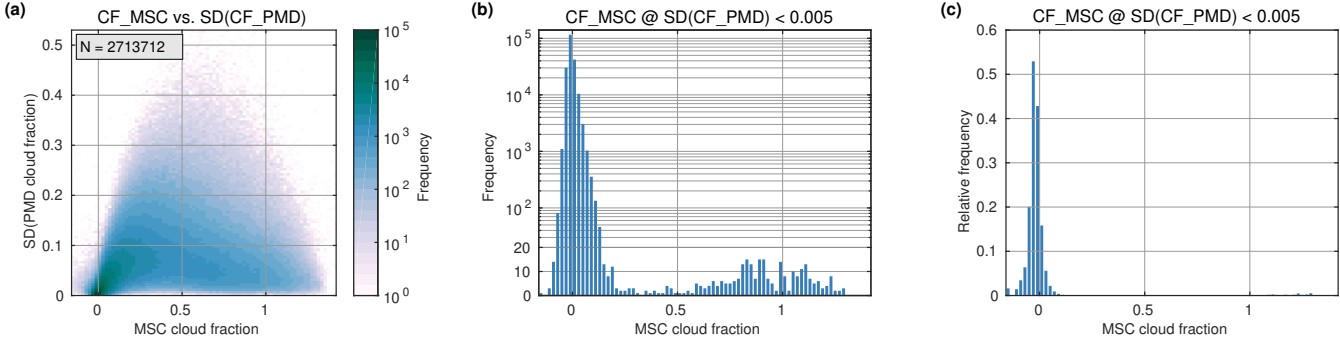

**Figure 15.** Standard deviation (SD) of PMD cloud fractions within one collocated MSC Pixel recorded in April 2010. PMD cloud fractions are obtained from PMD-PP channel 10 ($m=1$) with minimal spatial aliasing to MSC channel 10, which are both centred at 516.7 nm. (a) Density of PMD standard deviation and MSC cloud fraction; (b) histogram of MSC cloud fraction for a standard deviation of the collocated PMD measurements $< 0.005$, that is the lowermost row in (a); (c) relative frequency corresponding to (b). Note that the lower part of (b) is in linear, the upper part in logarithmic scale.

is physically only possible if all including PMD CFs are also zero. Therefore, standard deviations of PMD CFs within each MSC pixel are computed. Figure 15 shows the density of CF standard deviations versus MSC cloud fractions. The maximum standard deviation is minimal for small and large CF and maximal for CF of approximately 0.5. The absolute and relative distributions of MSC measurements corresponding to a standard deviation of PMD CF $< 0.005$ are plotted in Figs. 15(b) and (c), respectively. The width of the histogram bins is 0.02 CF. Figure 15(b) peaks at -0.01 CF and Fig. 15(c) at -0.03 CF, which can be interpreted as an estimate for the accuracy of small MICRU CF.





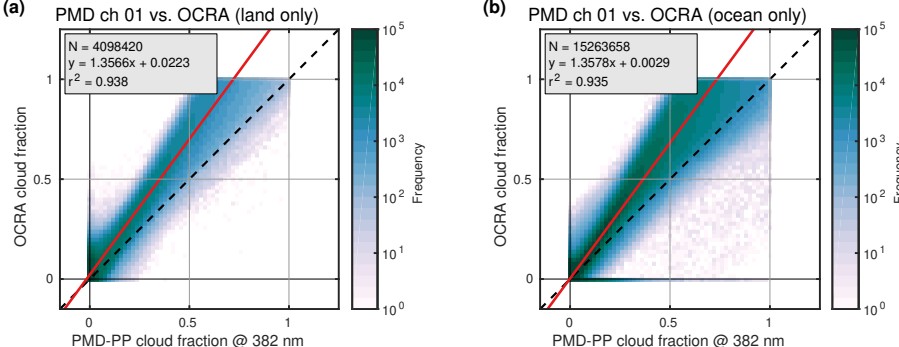

**Figure 16.** Comparison between MICRU MSC at 440 nm and OCRA based on CF data from April 2010: measurements over (a) land and (b) ocean. This comparison applies cropped data.

## 3.4 Comparison to other CF algorithms

### 3.4.1 MICRU vs. OCRA

Figures 11(s) and (t) in Sect. 3.2 suggest that OCRA measurements may be biased high in the western part of single cloud-free swaths over land. This observation is now investigated further based on monthly statistics. Figures 16(a) and (b) compare
MICRU and OCRA measurements recorded over land and ocean, respectively. Most apparently, the slope of 1.35 indicates differently definitions of the upper threshold. Figure D2(c) consistently confirms that OCRA overestimates large CF compared to FRESCO and MICRU CF. This is probably due to OCRA applying a scattering cloud model to define the upper threshold whereas FRESCO and MICRU apply a reflecting cloud model. It needs to be noted that this comparison between OCRA and MICRU applies cropped data as described in Sect. 2.6, which may furthermore affect the overall slope.

Secondly, it is focussed on the scatter of very small cloud fractions. Figure 16 clearly shows that OCRA CF for very small MICRU CF scatter more than MICRU CF for very small OCRA CF (cf. Fig. D1(b)). Figures 16(a) and (b) investigate this feature depending on surface type land and ocean, respectively. Small OCRA CF over land scatter significantly more for small MICRU CF when compared to measurements over ocean. This behaviour may again be explained by interferences from spatial heterogeneities of the land surface reflectivity. OCRA also applies measurements in the red spectral range, and, hence, is
affected by these heterogeneities.

Figure 17 shows a detailed comparison between OCRA and MICRU over land where the data is sorted according to the viewing direction west, nadir, and east, respectively. Focussing again on small CF, a significant bias of 0.066 may be detected when averaging over the 8 westernmost GOME-2 pixels of the swath (Fig. 17(a)), whereas the bias towards nadir and east is negligible (0.02). Furthermore, the scatter of small OCRA CF is larger towards western compared to eastern viewing directions.





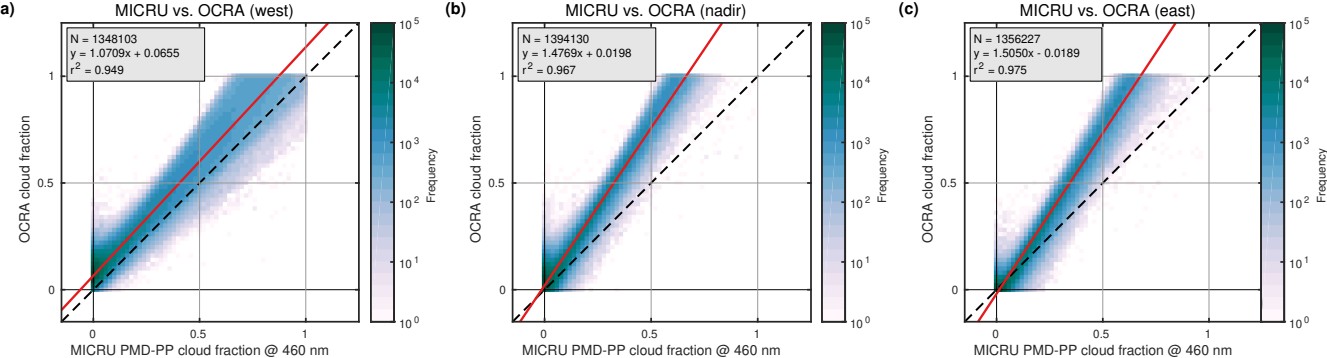

**Figure 17.** Same data as Fig. 16(a) but for different viewing angles: (a) west, (b) nadir, and (c) east. This comparison applies cropped data.

### 3.4.2   MICRU vs. FRESCO

Figure 18 compiles the comparison between MICRU CF at 440 nm and three FRESCO versions depending on surface type land or ocean, respectively. Overall, all FRESCO versions reveal a significantly higher scatter of small CF compared to MICRU, which is similar to the comparison to OCRA. Furthermore, the scatter over land is consistently larger compared to over ocean

due to the increased albedo effect at 757 nm applied by FRESCO. The slope of the comparisons is close to unity due to a similar definition of the upper threshold. But there are specific differences between the FRESCO versions.

The comparisons to FRESCO L1b in Fig. 18(a) and (b) feature smaller biases for small CF than FRESCO v7 and v8, which is in agreement with Fig. D1(b). The respective comparisons applying FRESCO v7 data in Fig. 18(c) and (d) reveal significantly larger scatter over both land an ocean. The bias over land is larger than 0.08, which seems to be a specific feature for FRESCO

v7 as all other investigated products are less biased against each other (cf. Fig. D2(b)). The comparison over land in Fig. 18(c) includes FRESCO CF up to 0.5 for MICRU CF=0, which may be attributed to albedo effects along coasts as investigated below (Fig. 20). Figure 18(d) than shows a small quantity of unrealistic CF smaller 0.2 over ocean for MICRU CF up to 0.8. Next, the comparison between FRESCO v8 and MICRU features a bias smaller compared to v7 but still larger than L1b. The scatter is significantly improved compared to v7, but the scatter of small FRESCO CF for small MICRU CF is still much larger

than vice-versa. In Fig. 18 may be furthermore observed that both FRESCO versions v7 and v8 feature CF <0. This is an improvement considering that physical measurements are affected by noise.

Similar to the comparison to OCRA, the comparisons between FRESCO and MICRU over land are now investigated depending on viewing geometry. Figure 19 consistently demonstrates that the western part of the swath are biased due to BRDF effects. Largest biases are observed towards east for FRESCO L1b and v7 (Figs. 19(d) and (g)), whereas the comparisons

between FRESCO v8 and MICRU are almost independent from the viewing direction (Figs. 19(g)-(i)). It is furthermore noted that the comparisons between MICRU and FRESCO L1b for nadir and eastern viewing geometries in Figs. 19(b) and (c) reveal an almost identical scatter of small MICRU CF for small FRESCO CF and vice-versa. This feature, however, is not visible in





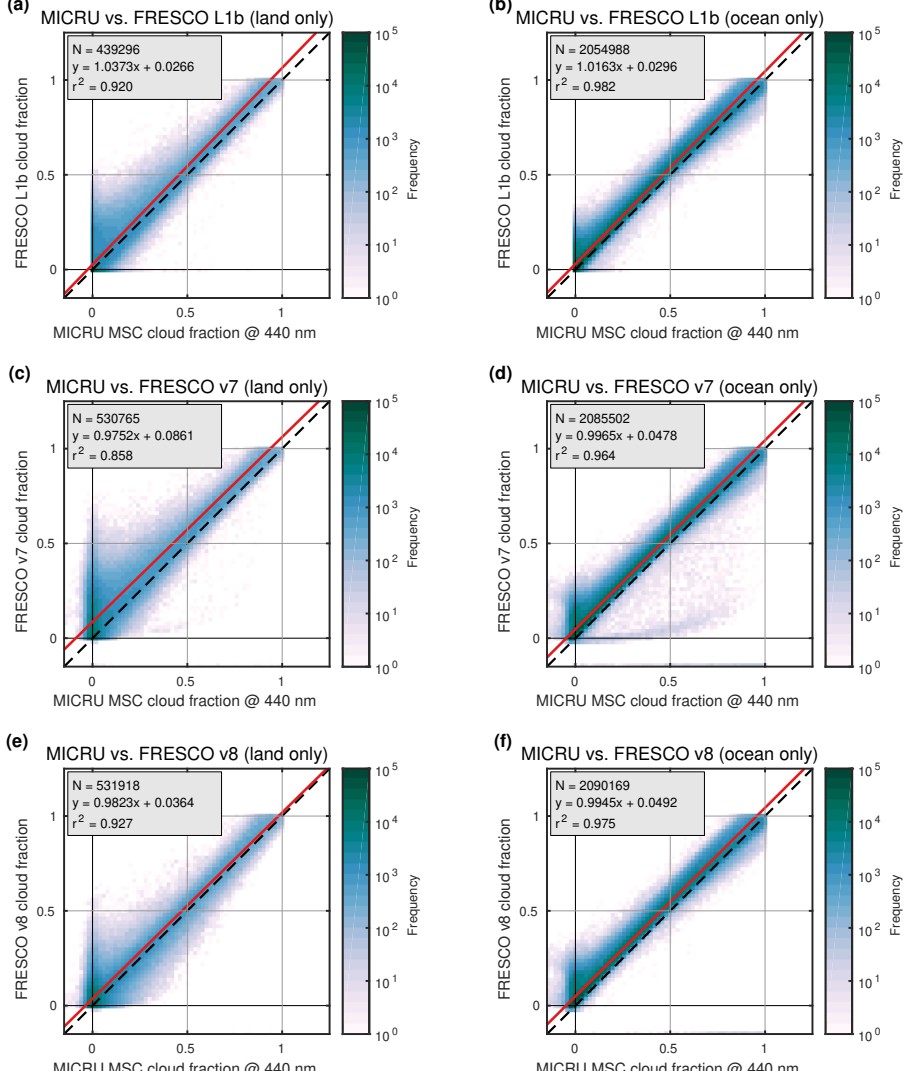

**Figure 18.** Comparison between MICRU MSC at 440 nm and three different FRESCO versions based on CF data from April 2010: measurements over land (left column) and ocean (right column); FRESCO L1b (top row), FRESCO v7 (middle row), and FRESCO v8 (bottom row). The comparison to FRESCO L1b applies cropped data. The comparisions in (c)–(f) only exclude CF > 1.

the comparison with the two more recent FRESCO versions, for which a significant amount of small CF are systematically biased (Figs. 19(e),(f), (h), and (i)).

A final comparison between MICRU and FRESCO focusses on spatial features in average CF maps. Figure 20 compares average MICRU CF maps derived at 440 nm with three corresponding average maps from FRESCO data. The maps zoom on
5 Mexico and its Pacific coast where interferences due to the land-sea contrast can be expected. All maps are computed using the same selection of data: January, April, July, and October 2010. In order to reduce potential interferences from BRDF effects (cf.





**Figure 19.** Viewing angle dependence of the comparison between MICRU at 440 nm and three different FRESCO versions over land. Same data as Fig.18(a), (c), and (e), respectively.

Fig. 19), only the central third (nadir) of the swath is considered where $|\theta| < 23.5°$. Significant differences between the 4 CF products are visible in the mean CF maps in the left column of Fig. 20 even though major features are similar. The difference plots in the right column quantify the differences. Comparing MICRU to FRESCO L1b in Fig. 20(c) reveals relatively high spatial gradients, especially at the coast of the peninsula. Furthermore, MICRU seems to be biased low in comparison to

5    FRESCO L1b over the Pacific and mainland Mexico, especially in the north-east of the zoom image. The comparison to FRESCO v7 in Fig. 20(e) differs significantly. Here, FRESCO v7 is biased high throughout the image. Furthermore, FRESCO v7 CF are biased low by more than 15% along the coasts, which may be attributed to the low resolution sampling applied





in the LER computation for this FRESCO version in combination with the larger albedo contrast between land and ocean in the wavelength rage applied by FRESCO. For FRESCO v8 (Fig. 20(g)), this issue is partially reduced by applying LER maps with an increased resolution along coasts. Another feature in the comparison to FRESCO v7 and v8 in Figs. 20(e) and (g) is pointed out. Both average FRESCO data are biased high with respect to MICRU in central Mexico east of Torreón
Municipality, where the surface albedo is significantly higher compared to the surrounding. Hence, this feature may again be due to the comparatively low spatial sampling of the background LER maps and the red wavelength range, where the influence of the albedo on CF is larger compared to shorter wavelengths. In contrary, this interference is not visible in Fig. 20(c) where a MERIS background map sampled at higher spatial resolution is applied.

### 3.4.3  Temporal evolution and degradation

So far, the statistical evaluations are carried out on monthly data aggregates. Now, the temporal evolution of the different CF products is investigated with respect to small CF. Here, the temporal evolution is studied based on the 15th percentile of monthly CF measurements framed by the 55° parallels. Preceding test showed that 10 to 15% of GOME-2 MSC measurements are effectively cloud-free. The selection of the 15th percentile showed optimal contrast for this study and avoids saturation or normalisation effects. Figure 21 compiles the temporal evolution of selected CF products and groups them in order to highlight
different comparative aspects.

There are, however, no significant trends visible in the investigated time period.

Figure 21(a) compares the time series of different cloud products: MICRU MSC channel 8, the same MSC channel but applying only one year of data (dashed blue), the two MICRU PMD channels at a similar wavelength of 460, nm (cf. Table 2), OCRA, and the three FRESCO versions. First off, the two MICRU MSC versions align almost perfectly. Apparently, reducing
the time interval used to derive the LT parametrisation has an almost negligible effect in this comparison. Furthermore, MICRU MSC and PMD data also align almost perfectly, with the PMD results being positively biased of $\approx 0.005$. The amplitude of bi-annual variations is of the same order. In contrary to the MICRU results, OCRA and FRESCO results feature a larger amplitude of at least 0.01 and an increased annual instead of bi-annual oscillation. For all products, the overall trends are negligible compared to the annual variations.

Figure 21(b) details the CF statistics depending on VZA over land (cf. Figs. 19 and 17). The plot confirms the aforementioned results: MICRU shows negligible VZA-dependence of the 15th percentile. For FRESCO v8, the 15th percentile of nadir measurements (solid gray line) is approximately 0.01 smaller compared to measurements in the eastern and western third of the swath. The VZA-dependence in the OCRA percentiles is significantly larger compared to MICRU and FRESCO. OCRA data reveals a clear east to west trend, while FRESCO v8 features minimal CF values in the centre third of the swath. The CF
in the western third of the swath of OCRA (dotted orange line) average to 0.06 while the average is close to zero in the eastern third of the swath.

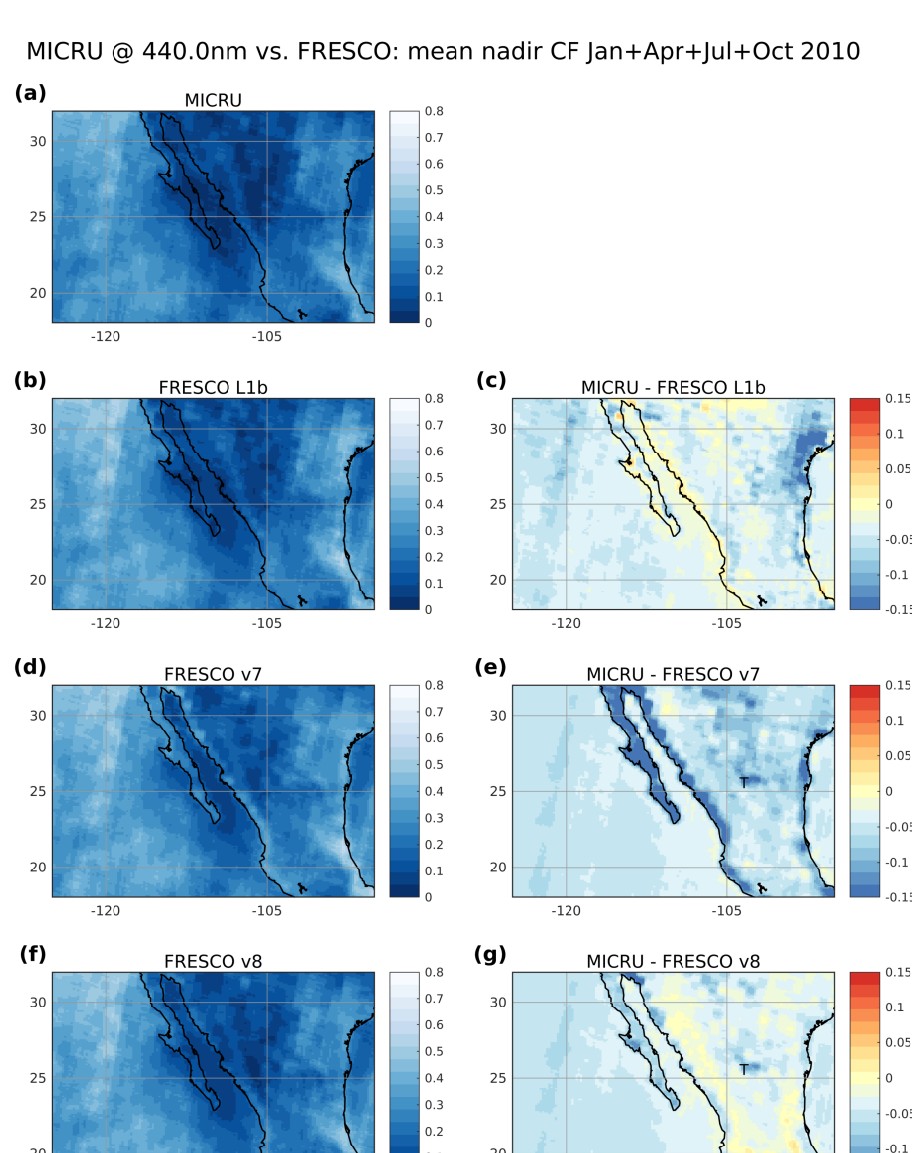

**Figure 20.** Comparison between average cloud fractions from different algorithms: (a) MICRU PMD channel 8 at 440 nm; (b), (d), and (f) FRESCO L1b, v7, and v8, respectively; (c), (e), and (g) respective differences between MICRU and FRESCO. Averages include all nadir measurements ($|\theta| < 23.5°$, same as in Figs. 19(b), (e), and (h), respectively) recorded in January, April, July and October 2010. The T in (e) and (g) marks the location of Torreón Municipality in Comarca Lagunera, Mexico (see text).



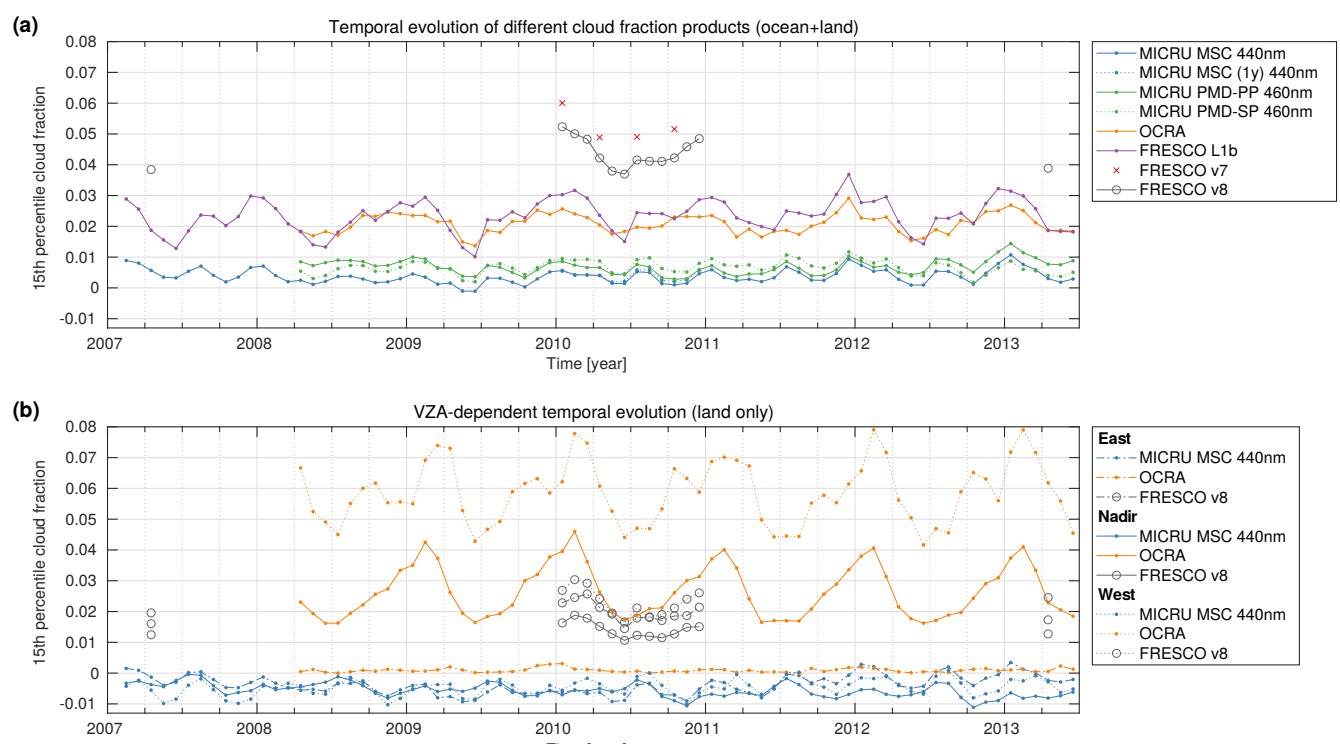

**Figure 21.** Temporal evolution of the 15th percentile of monthly cloud fraction distributions: (b) comparison between selected MICRU channels, OCRA, and 3 FRESCO versions and (b) viewing angle dependency of MICRU, OCRA and FRESCO v8. Note that the offset between the blue line in (a) and the average of the blue lines in (b) is due to a different data subset, that is all vs. land only, respectively.





## 4   Discussion

### 4.1   MICRU

Figures 8 and 9 illustrate the $T_{\min}$ retrieval at two different sites and wavelengths. The residuals in both examples (Figs. 8(f) and 9(f)) are on average significantly smaller than the targeted 0.04 CF accuracy. Systematic contributions from degradation,

seasonal variability, VZA-dependence, and sun glitter are small. This indicates that Eq. (4) is sufficient to parametrise $T_{\min}$ for GOME-2A. Figures. 11 and 21(b) also support this conclusion.

Equation (4) models the residual VZA dependence. Figure 10 shows that the VZA dependence and its temporal dependence vary between GOME-2 channels even though they are measuring in the same wavelength region. This indicates that the instrumental contribution to the residual VZA dependence is at least of the same order than possible inaccuracies of RT used to

invert the surface LER $T$. Hence, this issue can not be solved by a more accurate RT but needs to be corrected for empirically. Furthermore, degradation needs to be accounted for and it needs to be noted that the design of MICRU would allow degradation models different to Eqs. (4) and (10). For example, discrete functions and prescribed degradation are both possible to be included.

Considering the influence of surface BRDF effects on MICRU CF, our results support the discussion by Lorente et al. (2018)

that the LER model systematically underestimates the surface reflectance in forward direction corresponding to the eastern part of the GOME-2 swath. Figures B1(f) and C1(c) show that the average apex offset is biased high. The minimum gradient of the polynomial is more frequently in the eastern part of the swath. This suggests stronger contributions from the surface BRDF in the western part of the swath in accordance with Lorente et al. (2018). Therefore, MICRU CF in Figs. 11, 17, and 19 are consistently smaller in the western part of the swath when compared to OCRA and the three FRESCO versions because OCRA

and FRESCO both apply a constant surface reflectance model whereas MICRU applies an empiric VZA dependence model.

Lorente et al. (2018) furthermore claim that surface effects are stronger for longer wavelengths where atmospheric scattering is weaker. Figure C1(b) may support this claim: the average curvature of the residual VZA dependency decreases with wavelength but its variance increases. This observation is consistent for all three channels and suggests that, at shorter wavelengths, the residual VZA dependency is due to a combination of instrumental and RT effects, which are similar for the ensemble of

measurements. At larger wavelengths, however, the surface introduces a larger variance due to stronger spatial and season heterogeneities. However, the wavelength dependency of the apex offset in Fig. C1(c) is ambiguous. The average apex offset peaks between 450 and 700 nm for two out of three GOME-2 channels. The variance, on the other hand, is minimal for shorter wavelengths. Therefore, retrieving CF in the UV/blue spectral range is in any case beneficial in order to reduce interferences with surface albedo and type.

In addition to interferences with BRDF effects, there is another drawback of retrieving CF at larger wavelengths (cf. Sect.3.3.2), which is caused by the spectral surface albedo and its heterogeneity increasing with wavelength. CF retrieved at 382 and 440 nm correlate better compared to those retried at the $O_2$-B and $O_2$-A band at 670 and 757.5 nm, respectively. This effect is studied more systematically in Fig. D1(b) revealing a significant increase of the standard deviation of small CF between 521.8 and 670 nm.





Another aspect of the MICRU MSC channel intercomparison is a slope deviating from unity as, for example, shown in Fig. 13(a). CF at 382 nm are biased high with respect to those retrieved at 440 nm while the intercept at zero CF is negligible. Hence, the definition of $T_{max}$ apparently deviates between MICRU channels. Figure D2(c) compares the slopes of all MICRU channels. There is a significantly biased slope for MSC channels 1–4 retrieved at 389.7 and below. This step between MSC

channels 4 and 5 may be attributed to the application of different GOME-2 bands, specifically bands 2B and 3, from which the MICRU channels are extracted (cf. Table 1). We would like to note that we observed also the CF accuracy degrading near GOME-2 band edges when fine-tuning the MSC channel definitions (Table 1). The degradation depends only weakly on kernel width leading to the conclusion that this is a broadband effect, possibly caused by instrumental straylight. Furthermore, interferences with molecular absorption and atmospheric scattering resulting in a wavelength dependent $R$ may also cause a

systematic slope bias.

From the systematic studies compiled in Figs. D1 and D2(c) we may conclude that MSC channels between 424.5 and 521.8 nm are most consistent. The varying slope in Band 2B may be influenced by instrumental effects. For cloud height retrievals using the $O_2$-A band, MICRU channels 4 and 7 centred at 389.68 ad 433.4 nm, respectively, may be a good option for GOME-2 as they offer reasonable spatial aliasing (Fig. 3). There is a slight priority for channel 7 when considering the

mean and standard deviation of small CF in Figs. D1(a) and (b).

The comparison between MSC and PMD channel results with minimal spatial alignment in Fig. 14 shows an almost perfect correlation and biases $\leq 0.4$. The correlation in the UV is slightly lower compared to the visible, which may be caused by the inferior spatial aliasing in the UV. The comparisons between MSC and PMD furthermore reveal a slope significantly smaller than 1. Figure D2(b) reveals that this is a minor feature and may be explained by calibration differences between the different

GOME-2 channels. However, the slope is dominated by differences of the definition of the UT, which are not accounted for by MICRU. This behaviour has a minor effect on the accuracy of the LT.

The comparison between MSC and PMD measurements allows to estimate the influence of spatial (i. e. temporal) and the spectral alignment. Figure D2(a) confirms a maximum correlation for optimal $m$ as expected. The influence of the spatial alignment parameter $m$ is found significant and deviations by ±1 degrade the correlation from 0.998 to 0.992 for MSC at

516.7 nm. Results comparing spatial versus spectral alignment are not very clear. At least for PMD channel 4 (PMD-PP centred at 519 nm) spatial alignment seems slightly favourable over spectral alignment, which would be perfect for MSC channel 11 (Fig. D1(b)).

Hence, MSC channel 10 is selected over channel 11 to be compared to PMD-PP channel 4 with $m = 1$ in order to investigate the absolute accuracy of small CF. The comparison between the standard deviation of PMD CF within one MSC pixel in Fig. 15

indicates that the systematic bias of MICRU CF is of the order of -0.03.

Figure 21 finally shows that small MICRU CF have negligible trend over the investigated period of more than 6 years. The variations of the 15th percentile are smaller than 0.01. Clearly, CF at different viewing directions are significantly more consistent for MICRU compared to OCRA and FRESCO.





## 4.2 OCRA

OCRA CF are compared to MICRU PMD results based on singular orbits and exemplary monthly statistics. Both approaches consistently reveal that OCRA CF are biased high for observations towards west (Figs. 11(s), 11(t), and 17(a)). This indicates that BRDF effects have a stronger influence on OCRA results for observation geometries opposing the sun. OCRA CF values

in the eastern third of the GOME-2 swath have similar statistics like MICRU (Fig. 21(c)).

Considering the bias of small MICRU CF of -0.03, the bias of OCRA measurement in the western third of the swath can be estimated to 0.095 on average over land. These biases may be even larger depending on observation geometry and surface type (Fig. 11(t)).

The overall statistics in Fig. 16 indicate that small OCRA CF are on average less biased when taking into account the

negative systematic bias of MICRU (Sect. 3.3.3). The accuracy of singular OCRA measurements, however, is significantly and consistently lower compared to MICRU as revealed by the larger scatter of OCRA CF for very small MICRU CF than vice versa (Figs. 16, 17, and D1(b)).

From the investigation of OCRA CF over ocean in Fig. 11(u) it can be concluded that OCRA's empirical correction algorithm a bit too optimistic. While large contributions by sun glitter seem to be removed, the area at larger $\theta_r$ are still positively biased.

It is noted that the conservative MICRU sunglint warning flag contains the affected regions.

## 4.3 FRESCO

Three different versions of FRESCO CF are compared to MICRU MSC results based on singular orbits, exemplary monthly statistics, and average nadir maps. The different approaches for the background maps of the FRESCO versions are clearly visible in Figs. 19 and 20. Small FRESCO CF over land are significantly biased high for western observation geometries by

0.17 and 0.21 on average for FRESCO L1b and v7, respectively. In contrary, the comparison between FRESCO v8 and MICRU is almost independent of the VZA. Therefore, we conclude that the consideration of BRDF effects in FRESCO v8 displays a significant improvement compared to preceding versions of the product.

On the other hand, however, Fig. 20 shows that interferences with the coast are minimal for FRESCO L1b whereas both newer FRESCO versions are significantly biased high at coasts an inland. This issue is slightly improved in FRESCO v8, where

GOME-2 background LER data is sampled at a four times higher resolution at coasts. This specific positive bias along coasts may interfere with the processing and evaluation of tropospheric trace gas products from GOME-2 applying FRESCO because it leads to filtering a significant amount of measurements there when a CF-threshold filter is applied. Assuming a significant fraction of the worlds population resides along coasts this interference is considered significant. Less coastal measurements would be applied using FRESCO L1b, but this would come with the cost of filtering many measurements in the western part

of the swath as investigated by Fig. 19).

Compared to MICRU, FRESCO results are biased high as indicated in the top right section of Fig. D1(a). The mean biases of FRESCO v7 and v8 are between 0.05 and 0.07, which is unrealistic even when considering the systematic bias of MICRU CF. In this light, FRESCO L1b CF, however, are probably less biased on average compared to MICRU.





The scatter of FRESCO CF for very small MICRU CF, however, is consistently larger for all FRESCO versions than vice versa. This may be attributed to the application of radiances close to the $O_2$-A band as discussed above. From the comparison of MICRU CF at different wavelengths we may furthermore conclude that the attempt by Desmons et al. (2019) to apply FRESCO at the $O_2$-B band may not fully mitigate significant interferences with heterogeneities of the absolute surface albedo.

5    In general, it appears that features in the comparison between one FRESCO version and MICRU do not appear in the comparison with the other FRESCO versions. We therefore conclude, that MICRU reveals actually less systematic features than all three considered FRESCO versions.



## 5 Conclusions

MICRU is a cloud fraction retrieval algorithm based on satellite radiance measurements of backscattered solar radiation. The MICRU algorithm achieves an accuracy of 0.04 in calculating small effective cloud fractions (CF) from spectroscopic satellite measurements. This is a prerequisite for accurate trace gas retrievals because clouds in most cases determine the radiative

transport within each satellite measurement. The unique feature of MICRU is the application of an empirical BRDF surface model accounting for viewing angle dependencies in the cloud retrieval.

As a proof-of-concept, we applied MICRU to GOME-2A data, but the algorithm is also applicable to similar spectroscopic satellite missions like SCIAMACHY, OMI, TROPOMI/S-5P, and Sentinel-5. Furthermore, MICRU can also process UV/vis imager data like AVHRR, MERIS, MODIS, or Sentinel-2 due to its scalable design.

Our results confirm that MICRU is capable to accurately retrieve small CF over a wide spectral range, which renders it an optimal choice for tropospheric trace gas retrievals. These should use a cloud product based on radiance measurements of similar wavelength and spatial sampling so that the correlation between cloud and trace gas retrievals are optimised. For example, satellite instruments like Sentinel-5 have an inter-band offset of up to 30%, which deteriorates the applicability of particularly small cloud fractions retrieved in the red spectra range (e.g. FRESCO) to DOAS products like $NO_2$ and HCHO

retrieved in the UV/blue spectral region.

In conclusion, applying radiances recorded in the UV/blue spectral range is advantageous over the red spectral range in order to reduce surface effects. Furthermore, spatial alignment effects between MSC and PMD channels may be minimised by choosing appropriate spectral convolution kernels. In order to test the applicability of MICRU to data from recent satellite missions spanning less than 6 years, it was applied on a data set reduced to one year. Results of this alternative retrieval are

found to provide sufficient accuracy. Hence, MICRU may be applied on satellite missions offering less data than the GOME-2A mission with confidence.

*Data availability.* All spectral data is contained in the level 1b (L1b) data provided by EUMETSAT. GOME-2 data is available at EO Portal at eumetsat.int. MODIS snow data has been provided by Hall and Riggs (2016) Sea Ice data has been provided by Cavalieri et al. (1996) Global 30 Arc-Second Elevation (GTOPO30) Digital Object Identifier (DOI) number: /10.5066/F7DF6PQS Global Self-consistent, Hierarchical,

High-resolution Geography Database (GSHHG) can be downloaded from NOAA NCEI (NOAA, 2018). ERA Interim data used in this study were provided by the European Centre for Medium Range Weather Forecasts (ECMWF).





## Appendix A: $T_{\min}$ merging steps

The merging process consists of the following steps:

- Extrapolate all results to highest resolution.

- Discard result bins fulfilling

    - $i < 4$, or

    - $a_0 \leq \text{-0.5}$ for $\varphi > 80°S$.

- Additionally, filter oceans bins where

    - $N^* \leq 120$, or

    - $N^* \leq 1000$ for $\frac{N^*}{N} < 0.02$ and $|\varphi| \leq 65°$

    10    is fulfilled, except in the coarsest resolution realised in binning 7.

- Additionally, filter bins over land with $N^* \leq 80$.

- Filtered maps are merged from coarse to fine resolution in order to maintain the highest resolution possible.

- Empty bins – or gaps – are extrapolated as follows:

    - For $\varphi \leq 80°S$, zonal means are copied into the gaps, if there are no data at the same latitude at all, $\beta$ from one row
      towards North are applied

    - Remote gaps, like islands or lakes, which are more than 6° longitude and 3° latitude away from a valid bin, are
      filled with zonal mean values

    - Remaining gaps, e. g. at coasts, are filled by successive applying a Gaussian convolution kernel with a $1\sigma$ diameter
      of 3° longitude and 1° latitude.

- Set $a_g = 0$ for $|\varphi| > 65°$

## Appendix B: $T_{\min}$ parameter map

Figure B1 shows an exemplary stitching result for MSC channel 2. The spatial origin of the finally applied fit results is compiled
in Fig. B1(a): Fit results from the highest resolution - binning 1 over land and binning 4 over ocean – are in dark blue colour.
Extrapolated bins are in coloured yellow. Coastlines typically apply coarser resolutions because of the poorer statistics and the

25    90% cut-off criterion for the respective surface types. Polar oceans appear in light blue or orange colours because frequent
clouds and bright sea ice significantly reduce $N^*$.



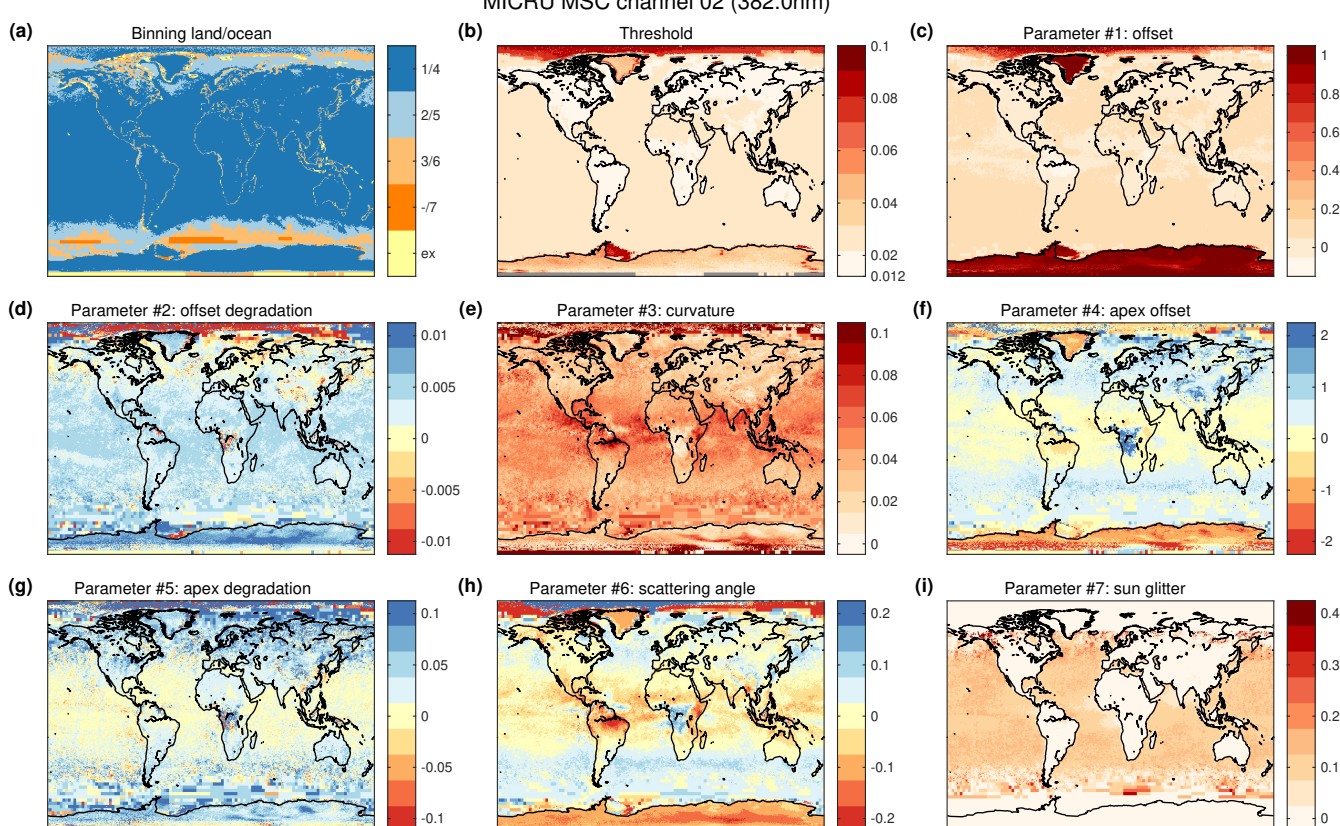

**Figure B1.** Parametrisation of the lower threshold (LT) for MSC channel 02 after post-processing: (a) binning over land and ocean, (b) final threshold $T$, (c) offset, (d) offset degradation, (e) curvature, (f) apex angle, (g) apex degradation, (h) scattering angle, (i) sun glitter. In (a), colors other than dark blue indicate regions, where the resolution is reduced or extrapolation (ex) is applied. Coast containing pixels display land results.

Figure B1 is a merged $T_\mathrm{min}$ map of all binned $T_\mathrm{min}$-results for MSC channel 2. It is the base LUT for calculating $R_\mathrm{min}$ globally. Figure B1(b) indicates, that $\tau$ increases over water and desert regions with larger LER. Figures B1(c) through (i) illustrate the geographical dependence of the fitted surface parameters. A significant latitudinal dependence and interferences with the increased cloud probability at the ITCZ can be observed for all parameters. A significant land-sea contrast is visible for the apex degradation in Fig. B1(g): the apex seems to shift systematically less over land. Furthermore, there is a significantly different behaviour of the apex angle and scattering angle dependence over the rain-forest regions between South America and Africa in Figs. B1(f) and (h), respectively. Both regions feature a comparatively large cloud probability, and, hence, less statistics available for the $T_\mathrm{min}$ retrieval.





It is noted that Figs. B1(d) and (g) reveal that both trend terms are latitudinally correlated with opposite sign. A discussion of independent terms is therefore difficult. This observation, however, confirms an overall degradation of the VZA dependence as shown in Fig. 7.

Sunglint is not constant throughout Fig (i) and, therefore, cannot be corrected for a-priori. Specifically, (i) indicates lower values in the ocean west of equatorial Africa and India. Towards the poles scatter increases as sunglint contribution diminishes. Over the amazon, interestingly, there is a clear signal from sun glitter, which was not expected a-prior, illustrating the gain by fitting this parameter also over land.

## Appendix C: Wavelength dependency of MICRU results

Figure C1 compiles the average wavelength dependency of the $T_{\min}$ parameters for MSC and PMD MICRU channels. Apparently, offset degradation is much more an issue for MSC and PMD-PP when compared to PMD-SP (Fig. C1(a)), which is in accordance to the findings from Fig. 10. The VZA-dependence is consistently stronger in the UV compared to MICRU channels at longer wavelengths (Fig. C1(b)) but relatively smaller for PMD-PP as already discussed above. The wavelength dependence of the apex degradation in Fig. C1(d) seems to be much more complex, changing signs below 500 nm. The wavelength dependence of $a_s$ is small compared to the overall variability in Fig. C1(e). $a_g$ in Fig. C1(f) are increasing with wavelength suggesting Rayleigh scattering is damping the contribution from sun glitter in the UV.

## Appendix D: Matrix comparison between MICRU, FRESCO, and OCRA

Figures D1 and D2 show a selection of comparisons between different MICRU MSC, MICRU PMD, FRESCO and OCRA cloud products for April 2010. Figures D1(a) and (b) show the mean CF and standard deviation of selected CF for the product in $x$-direction. The selection includes only those measurements fulfilling $|c| < 0.01$ of the product in $y$-direction. As a reference, results in Fig. 13, that are results where MICRU MSC channels 2, 12 and 14 are compared to MICRU MSC channel 8 evaluated at 440 nm, correspond to the results in the second row as indicated by circles in Fig. D1 (a).

Focussing on the inter-MSC comparison, the upper left block in Figs. D1 and D2, one observes that the MICRU results are relatively consistent for all MICRU channels. The standard deviation in Fig. D1(b) reveals a significant jump between 521.8 and 670 nm, which was already observed in Fig. 13. Figures D2(a) and (b) show that the correlation degrades and the deviation of small CF increases with increasing spectral distance. The slope, which is dominated by $R_{\max}$, reveals a significant jump between 389.7 and 425.5 nm in Fig. D2(c) coinciding with a step from MSC band 2B to band 3 (cf. Table 1).

### MSC applying less data

In addition to the standard MSC evaluation integrating 6 years of measurements (Sect. 2.1.1), an evaluation using only one year of data is performed in order to simulate the performance of MICRU if applied to shorter data sets. We chose the year 2010 because it ranges approximately in the centre of the standard evaluation period where CF accuracy may be assumed optimal





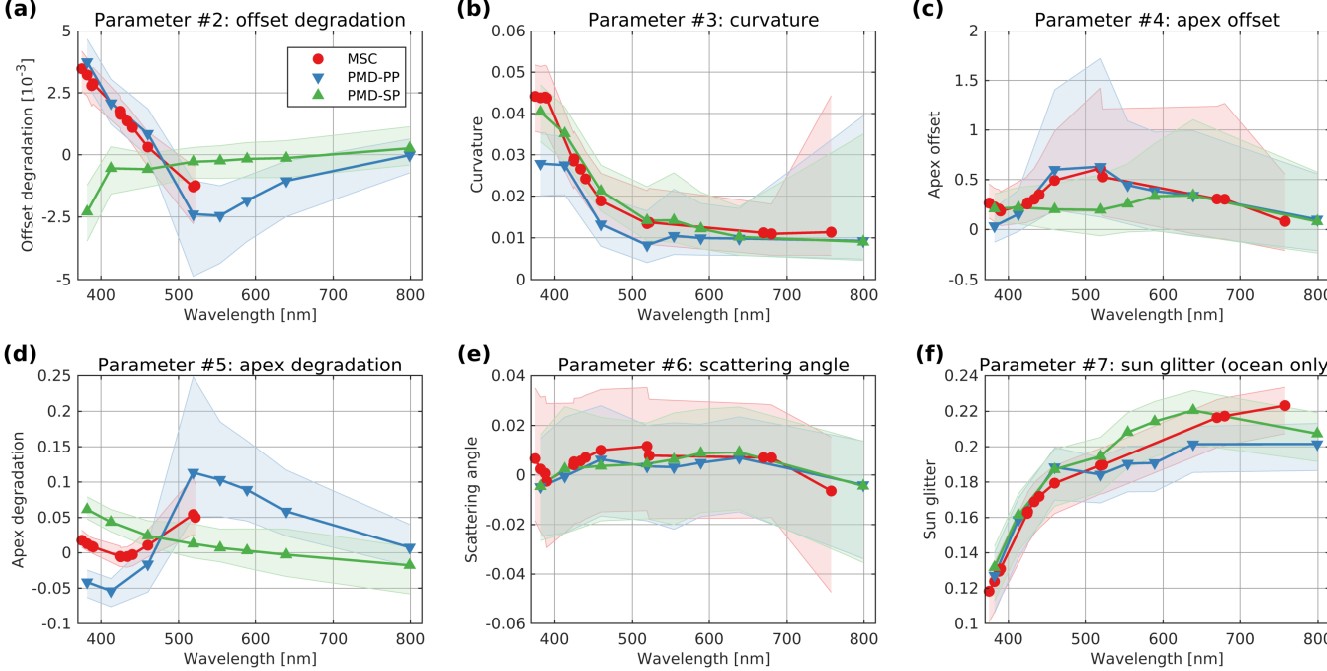

**Figure C1.** Wavelength dependency of $T_{min}$ parameters: (a) offset degradation, (b) curvature, (c) apex offset, (d) apex degradation, (e) scattering angle, and (f) sun glitter. MSC channels in red, PMD channels 1–8 in blue and PMD channels 9–16 in green. Interconnected data points are median fit parameters. Transparent areas include 50% of all results with boundaries at the 25th and 75th percentiles.

(cf. Sect. 3.4.3). For this special evaluation, the parameters accounting for degradation ($a_t$ and $a_{a1}$ in Table 5) are constrained to 0 for all MSC channels. The results of this simulation are included in the result matrices in Figs. D1 and D2 and denoted by *1 year*.

The circles in Fig. D1(b) indicate the entries, where the new results are compared to the standard results, which are located

5   one column to the left. All circled values are larger compared to the standard evaluation. This indicates that small cloud fractions are less accurate if a shorter evaluation period is chosen. The corresponding matrix entries in Fig. D1(a) indicate that small cloud fractions retrieved at 382 nm are more deviated compared to the standard evaluation (-0.3) compared to those at longer wavelengths, where corresponding entries are negligible (denoted by ·).



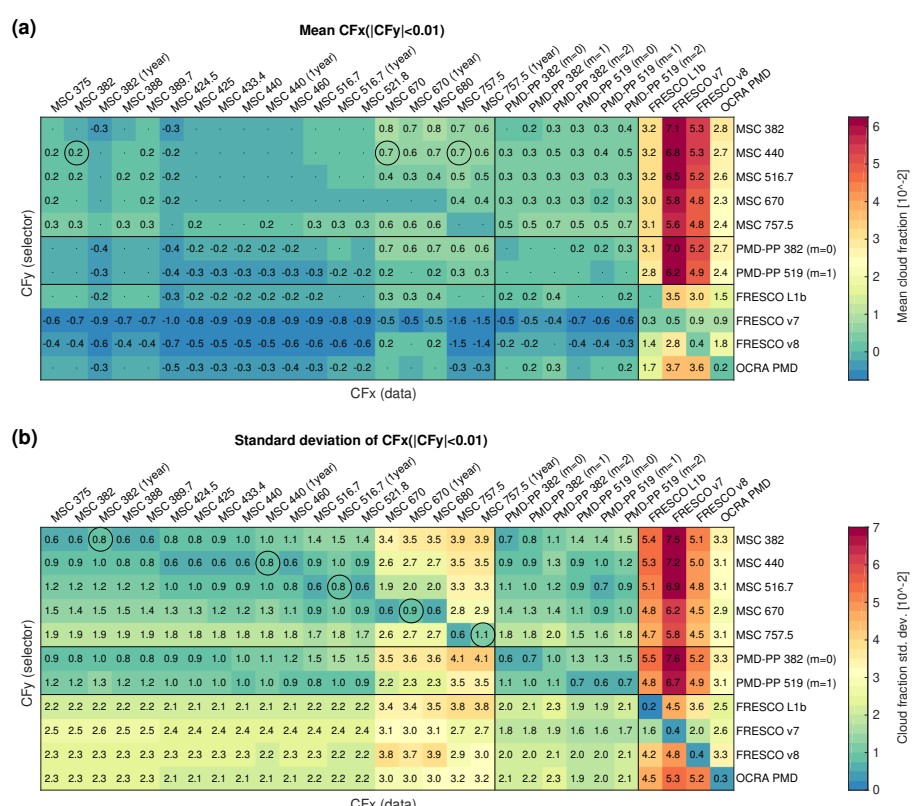

**Figure D1.** Tabular inter-comparison of MICRU and external cloud fraction results for April 2010: (a) Mean CF of product listed in x direction selected by corresponding product in y direction with an absolute CF<1%, (b) same as (a) but listing the standard deviation. The circled values in (a) correspond to Fig. 13. The circled values in (b) indicate values comparing data of the same MICRU channel but different evaluations periods: one year versus the standard data set applying 7+ years.

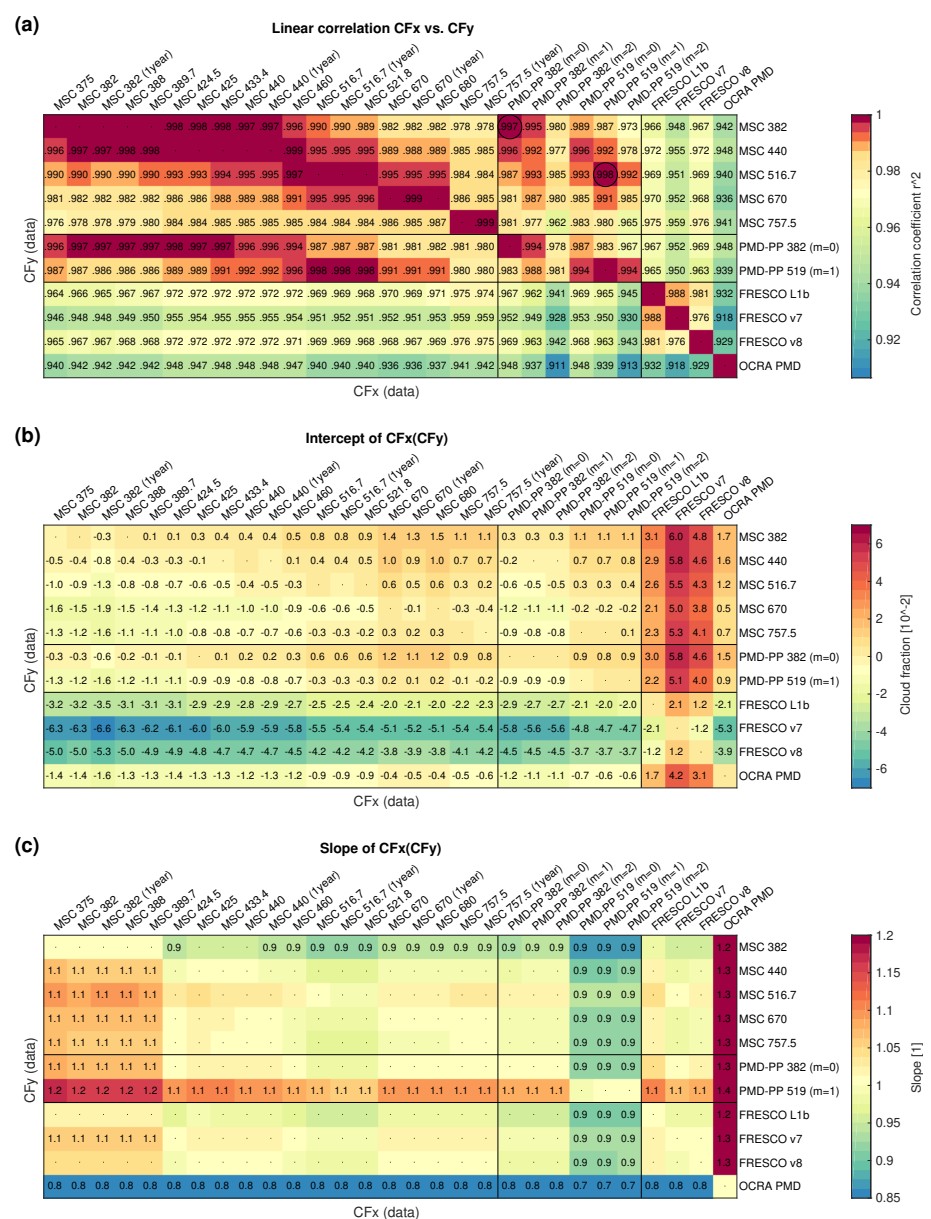

**Figure D2.** Continuation of Fig. D1: (a) linear coefficient of correlation $r^2$, (b) and (c) intercept and slope of bi-variate linear fit, respectively. Numbers of small deviations from 0 and 1, respectively, are omitted for the sake of clarity. The circled values in (c) correspond to Fig. 14.

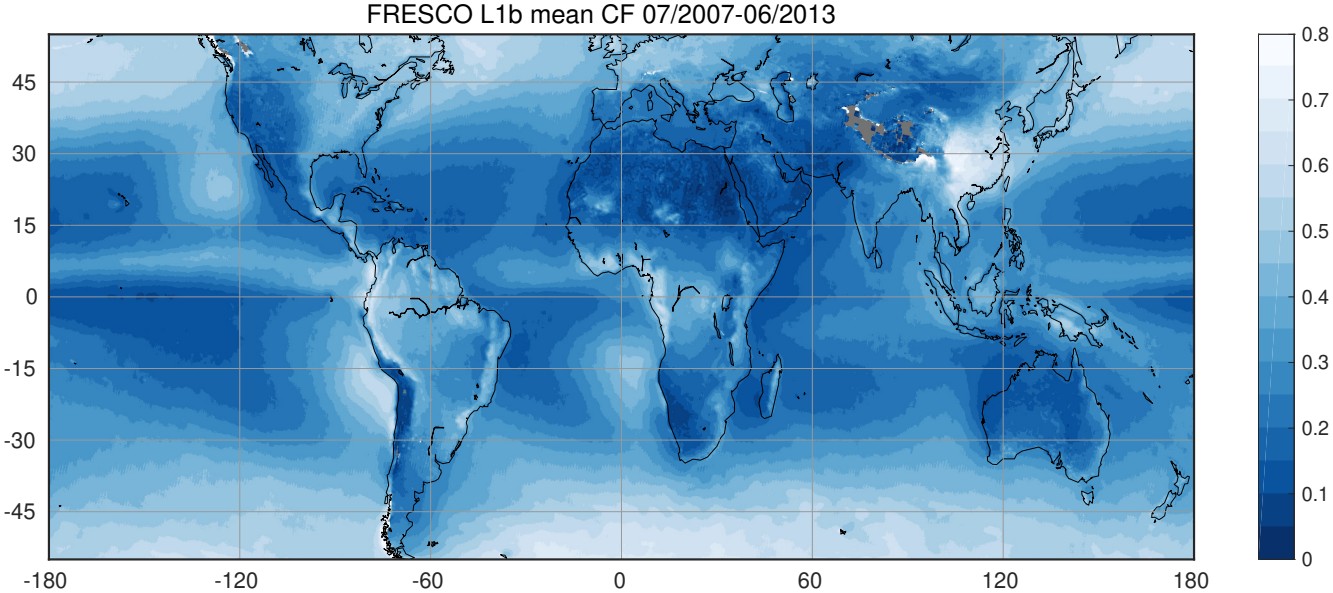

**Figure E1.** Same as Fig. 12 but six year average of FRESCO L1b cloud fraction measurements recorded between 1 July 2007 and 30 June 2013. Areas without data are plotted in gray.

## Appendix E:  Global average cloud fractions

This section compiles global average cloud fraction maps of FRESCO L1b (Fig. E1), FRESCO v7 (Fig. E2), FRESCO v8 (Fig. E3), MICRU PMD channel 4 (Fig. E4), and OCRA (Fig. E5) similar to Fig. 12 in Sect. 3.3.1 applying MICRU MSC channel 8 in the main body of the paper. All maps are compiled from 6 years of data collected between July 2007 and June 2013, except for Figs. E2 and E3. Figure E2 comprises 4 months, Fig. E3 comprises year 2010, onky.

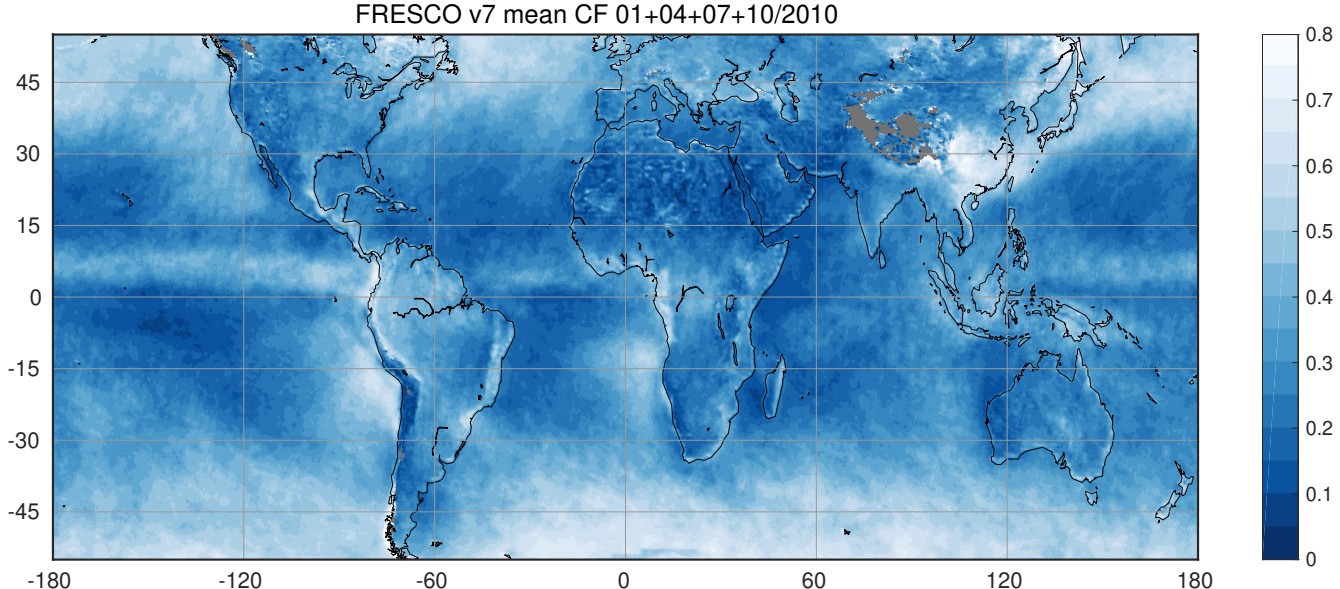

**Figure E2.** Same as Fig. 12 but four month average of FRESCO v7 cloud fraction measurements recorded in January, April, July, and October 2010. Areas without data are plotted in gray.

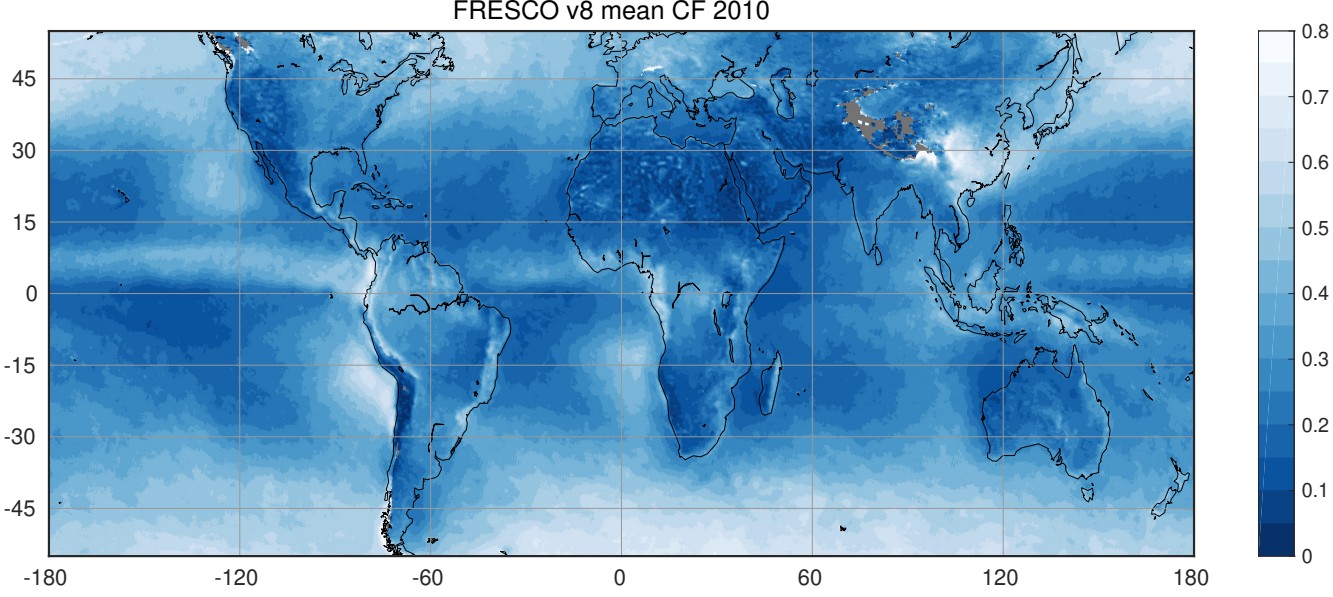

**Figure E3.** Same as Fig. 12 but one year average of FRESCO v8 cloud fraction measurements recorded in 2010. Areas without data are plotted in gray.



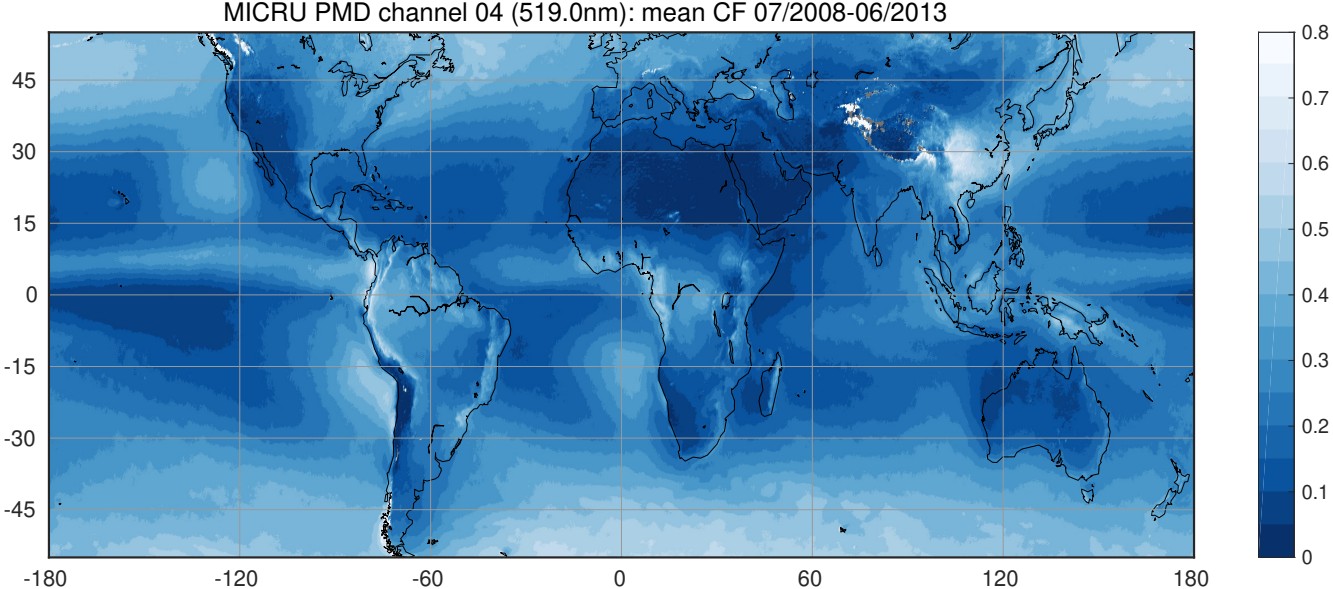

**Figure E4.** Same as Fig. 12 but five year average of MICRU PMD channel 04 cloud fraction measurements recorded between 1 July 2008 and 30 June 2013. Areas without data are plotted in gray.

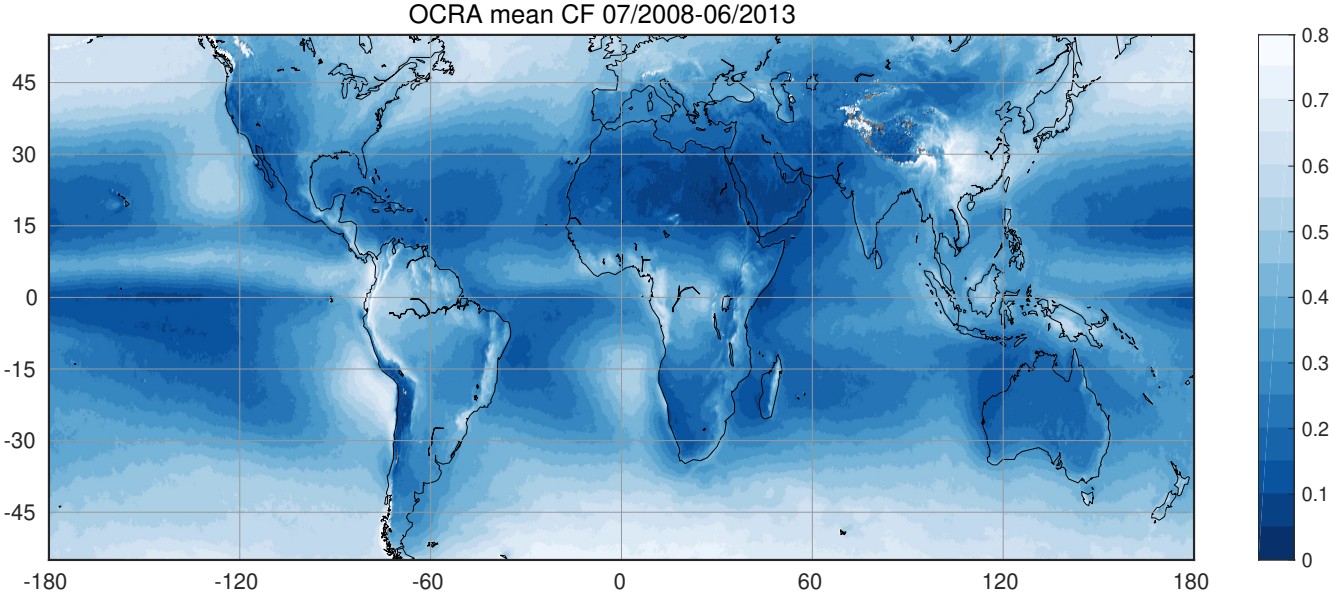

**Figure E5.** Same as Fig. 12 but five year average of OCRA cloud fraction measurements recorded between 1 July 2008 and 30 June 2013. Areas without data are plotted in gray.





*Competing interests.* The authors declare that they have no further conflict of interest.

*Acknowledgements.* Michael Grzegorski, the main developer of HICRU, for inspiration and discussion. Andreas Richter (IUP Bremen) for fruitful discussions and first application of the data. Furthermore, Piet Stammes (KNMI), Ping Wang (KNMI), Ronny Lutz (DLR), Diego Loyola (DLR) Rüdiger Lang (EUMETSAT) are gratefully acknowledged. EUMETSAT and NASA are acknowledged for providing data.

5   The program developers at IUP/IFE Bremen are acknowledged for providing the SCIATRAN software. This study was funded by DLR Bonn under contract 550EE1247 as part of the Tropomi/S-5P verification and by ESA under contract ST-ESA-S5L2PP-CON-003 as part of the Sentinel-5 verification.





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
