# Peer review of "MICRU: an effective cloud fraction algorithm designed for UV/vis satellite instruments with large viewing angles"

_Atmospheric Measurement Techniques, 2020_

## Referee Comment (RC1) · Anonymous Referee #1 · 24 Jun 2020

Summary:

Section 1 introduces a basic motivation and explains why cloud fraction retrievals are an important ingredient to trace-gas retrievals. It further provides an overview of existing algorithms and their respective heritage. A particular importance is rightfully directed to surface contributions and to the recent developments in the field to address BRDF effects. Instrument characteristics and the cumbersome relations between MSC and PMD read-outs are well explained (section 2.1). All used auxiliary data are briefly introduced and their treatment (spatio-temporal interpolation) is justified. The main part deals with the determination of the lower threshold T_min (section 2.3). The re-

sult for the presented approach of a surface fit is shown for two example geolocations (Australia, Atlantic) at two different wavelengths (382 nm and 516.7 nm), respectively. Finally, the data sets where MICRU is compared against are briefly described (section 2.6). Section 3 shows results of MICRU for three example scenes of GOME-2 (Brazil, North America, Indian Ocean) and compares those to different FRESCO versions and OCRA. The various MICRU versions (MSC, PMD, different wavelengths) are also inter-compared to highlight their differences. The comparison to FRESCO and OCRA is also extended to monthly statistics and the temporal evolution is investigated. An interpre-tation of the comparisons and findings for several selected individual cases and larger statistics is presented in section 4, both for the various MICRU applications and also for the comparison algorithms. Section 5 reminds that the main topic of investigation are small cloud fraction regions and recalls the novelty approach of an empirical BRDF surface model. It concludes a transferability of MICRU to other spectroscopic satellite missions and imager data. It is finally recommended to prefer the UV/blue spectral region over the red spectral region to reduce surface effects. Finally, Appendices A to E provide further valuable information for the reader.

As an overall conclusion, the manuscript is well written and structured, provides a relevant and very interesting contribution to the scientific topic addressed and therefore I recommend its publication after the general comments are addressed.

general comments:

GC1): While a lot of effort is put into the investigation and determination of the lower threshold, the description and assumptions of the upper threshold are quite brief. The authors may consider to expand a bit on the justification of the chosen simplified ap-proach for the upper threshold and for which types of clouds it may be justified and for which not.

GC2): Since many cloud retrieval algorithms struggle particularly over very bright sur-faces, has the MICRU performance also been tested e.g. over snow/ice conditions?

Also the behavior over different strong aerosol events (desert dust, urban pollution etc.) could be interesting to analyze.

GC3): The consequences of using a fixed cloud albedo of 0.8 (p2, l30; section 2.4, etc.) should be discussed in more detail in the paper. For example, how should the trace gas retrieval use a MICRU CF>1 (p19, l8)? Are the MICRU CF applicable only to tropospheric trace gas retrievals?

GC4): A direct comparison between MICRU CF and FRESCO CF is possible because both algorithms use a fixed cloud albedo of 0.8. OCRA doesn't have this constraint, therefore the comparison between MICRU and OCRA (Section 3.2, 3.4.1., 3.4.3, 4.2, Fig. 11, Fig. 16, Fig. 17, Fig. 21 Appendix D,Appendix D, Fig. D1, Fig. D2, Fig. E5) should be extended by adding an additional 'OCRA CF_fixed_albedo' by converting the OCRA CF to a magnitude similar to the MICRU CF using the following approximation:

OCRA_CF * ROCINN_CA $\sim$ MICRU_CF * 0.8

OCRA_CF_fixed_albedo = OCRA_CF * ROCINN_CA/0.8

where ROCINN_CA is the cloud albedo retrieved with ROCINN. This adaptation can only be done at the MSC level because the ROCINN parameters are only provided for the MSC footprints and not at PMD level, hence for the OCRA PMD cloud fraction, this modification cannot be applied. However, the manuscript focuses on providing accurate cloud information for the retrieval of trace gases. Since the trace gases are retrieved for the MSC data, I would strongly suggest to add to the comparisons also the OCRA cloud fraction for the MSC data including the adaptation with the ROCINN cloud albedo as outlined above. In summary, I would recommend to

a) add to the comparisons also the "OCRA_CF_fixed_albedo" for the MSC data as outlined above, and

b) add to the conclusions for the OCRA PMD vs MICRU PMD comparisons a statement that the modification with the cloud albedo as outlined above cannot be done at PMD
level and this might be a potential source for discrepancies in the comparison.

Further specific comments and technical corrections below refer to p(age) and l(ine) of manuscript amt-2020-182.pdf:

specific comments:

p3, l4: There are also new retrieval algorithms that combine the LER models with a geometry-dependent BRDF correction, see for example (Loyola et al., 2020) https://doi.org/10.5194/amt-13-985-2020

p3, l32; p5, l1; and p22, l11: The third version of OCRA was not applied to TROPOMI but to OMI. The fourth version of OCRA (Loyola et al., 2018) https://doi.org/10.5194/amt-11-409-2018 is applied to TROPOMI, but this reference is missing in all three paragraphs.

p4, l34: To be more precise, for the TROPOMI/S5-P mission OCRA is part of the S5P L2 CLOUD product and FRESCO is an auxiliary cloud product used for the S5P L2 NO2, ALH, CH4 and O3 profile products.

p6, l23: Could you specify, which data version was acquired from EUMETSAT? Was it the reprocessed AC-SAF data set?

p10, Figure 4: The last sentence of the figure caption might be confusing and binning #1 might be misunderstood as native resolution. Please rephrase to avoid confusion.

p10, l5: Forward and backward scans were not mentioned before. A short introduction and explanation, why backward scans are discarded could be beneficial for the reader here.

p15, caption of Figure 7: In addition to observation geometry and wind speed, doesn't the sun glitter contribution also depend on geolocation and time? E.g. in summer the sun glitter appears at different latitudes than in winter. Please clarify.

p15, l8: Same reference is given twice: "...swath edge in Figs. 8(d,e) and 8(d,e)"

p15, l9-13: Additional tests are described but discarded because of inferior results in a number of case studies. In my opinion it could be elaborated a bit more on the reasoning of selecting these choices in order to give the reader a bit more background information.

p16, table 5: Why is degradation for MSC channels 12, 13, 14 constrained to 0? Please add short explanation.

p17, Figure 8(b): Is there a mixup, where in the text ". . . measurement set Omega_0 (red) and finally fitted set Omega_I (blue)", the colors red and blue should be swapped? The figure legend says that the fitted data are the red points.

Figures 8e and 9e: The surface fits for MSC 2 over Australia (8e) and MSC 10 over Atlantic (9e) show very different patterns. The former is strongly pronounced in the western half of the swath and becomes stronger in time while the latter is restricted to the east half of the swath and present for all years 2007-2013. Is there a simple explanation for the driving factor behind these differences (except geolocation and wavelength)? Edit: Ok, the pattern of the latter is later addressed on p24, l30 and assigned to sun glitter but what could be the explanation for the increasing trend in the first example?

p19, l8: How are MICRU CFs > 1 treated? Are they cropped to 1 or flagged? Please clarify.

p19, section2.4: The upper threshold is assumed to be fixed with a cloud albedo of 0.8 at an altitude of 7km without a dependency on geolocation and time. While a lot of effort has been put into the various dependencies for the lower threshold, the assumptions on the upper threshold are few. It is argued that MICRU focuses on small CFs, but how valid is e.g. the assumption of cloud albedo = 0.8 in the case of an optically very thin cloud close to the surface or very high in the troposphere? Could it be specified for which type of clouds this simplified assumption on the upper threshold is valid and justified and for which not?

[Figure]

p21, l24: The sentence "probably the most commonly used CF product" is not correct. Most of the operational GOME-2 AC-SAF trace gas products are based on the OCRA/ROCINN algorithm.

p22, l14: "and OCRA applies a volumetric (or scattering) cloud model". Technically, the scattering cloud model is only relevant for the ROCINN part of the OCRA/ROCINN algorithm combination, which retrieves cloud top height and cloud optical thickness. For the OCRA cloud fraction using the color space approach, the relevant assumption is a spectral independence of a fully cloudy reflectance across the UV/VIS wavelength range. Therefore, "and OCRA applies a color space approach for the upper threshold (Lutz et al., 2016)" seems more fitting.

p23, l1-2: The sentence "These biases propagate into trace gas retrievals if normalized CF data are applied" may apply to FRESCO but is not correct for OCRA/ROCINN. Any possible bias on the OCRA CF will be compensated in the ROCINN cloud albedo and therefore possible bias will be not propagated into trace gas retrievals (Loyola et al., 2007) http://dx.doi.org/10.1109/TGRS.2007.901043

p31, Figure 16: Title and axis labels say "PMD ch 1" and "PMD-PP cloud fraction @ 382 nm" while the figure caption reads "MICRU MSC at 440 nm". Please clarify.

p31, l7: see comment to p22, l14

p37, Figure 21: Is there a reason that subplot (a) shows all three FRESCO versions L1b , v7 and v8 while in subplot (b) only the FRESCO v8 are shown?

p37, Figure 21: Are sun glitter scenes included in the 15th percentile cloud fractions shown? The difference for the OCRA East might be due to the fact that sun glitter appears only in the east half of the swath and OCRA sets scenes affected by sun glitter to zero by default. This might contribute to the low bias.

p38, l19-20: Please note that the VZA dependence in OCRA is also evaluated empirically with a monthly temporal resolution.

p40, l13-15: It is true that the OCRA sun glitter removal at areas with larger theta_r may be positively biased (particularly visible in the left swath of Figure 11(u)). However, it could also be pointed out here that in regions of very strong sun glitter (yellow in Figure 11(c)), OCRA seems to properly account for this effect (visible in the right swath of Figure 11(u)). Furthermore, OCRA includes a sun glint flag.

p45, l6: This is a very interesting detail. Could this terrestrial sun glitter signal over the Amazonas be related to high oriented ice crystals as suggested by Marshak et al. based on EPIC/DSCOVR? (Terrestrial glint seen from deep space: Oriented ice crystals detected from the Lagrangian point https://doi.org/10.1002/2017GL073248)

technical corrections:

p2, l25: change "the time dependent the" to "the time dependency of the"

p2, l28: change "a-priory" to "a-priori"

p2, l30: change "albedo of 0.8 Stammes et al. (2008) rendering" to "albedo of 0.8 as in Stammes et al. (2008), rendering"

p7, l16: change "channels 2,5, 10" to "channels 2, 5, 10" (blank space missing before 5)

p16, l21: change "a-priory" to "a-priori"

p20, l1: change "to form a complete parametrisations" to "to form complete parametrisations"

p20, l17: change "prior the T_min retrieval" to "prior to the T_min retrieval"

p22, l4: something is missing in "is increased to effects of". Maybe "is increased to cover effects of"?

p22, l15: Sentence "Some FRESCO both and OCRA. . ." sounds weird. Is the following meant: "Both FRESCO and OCRA. . ."?

p24, l17: change "in discrete boxed defined" to "in discrete boxes defined"

p29, l5: change "evaluation" to "evaluations"

p29, l6: change "and m = 1 519nm" to "and m = 1 at 519nm"

p31, l6: change "differently" to "different"

p32, l19: "Figs. 19(d) and (g))": Shouldn't this be "Figs. 19(c) and (f)"?

p37, caption of Figure 21: first line ends with "(b): comparison between selected MI-CRU", while it should be "(a): comparison between selected MICRU".

p38, l32: change "retried" to "retrieved"

p40, l5: "(Fig. 21(c))": Fig. 21 has no subplot (c). Is Fig. 21(b) meant?

p40, l24: "at coasts an inland": Is "at coasts and inland" meant?

p40, l30: "as investigated by Fig. 19)." The closing bracket has no opening bracket.

p48, caption of Figure D2: "circled values in (c)": The circles are in (a)

p49, l5: at the end of line change "onky" to "only"

---

## Referee Comment (RC2) · Anonymous Referee #2 · 29 Jun 2020

Review of "MICRU background map and effective cloud fraction algorithms designed for UV/vis satellite instruments with large viewing angles" by Sihler et al.

The manuscript describes a model for accounting for anisotropic reflection of solar light from Earth's surface in an effective cloud fraction algorithm designed for UV/Vis satellite instruments. Results of the application of the algorithm to GOME-2 data are compared with other cloud fraction algorithms. Appendices provide technical details of the developed algorithm. The manuscript is clearly relevant for AMT. Even though the material is not a significant advance in remote sensing of clouds it could be published to document the GOME-2 cloud fraction algorithm in the literature. The abstract provides a concise and complete summary of the paper. The earlier work is properly credited. I recommend publication of this manuscript only after major revisions which address the following comments.

General comments

1. The authors do not clearly state what are the main improvements of the proposed algorithm as compared with the existing cloud algorithms which also accounts for surface BRDF. It would be useful to summarize those improvements in Conclusions.
2. Low values of LER are of the primary interest for the construction of a minimum LER map (background map) which is the core of the developed algorithm. The existing surface reflectance data sets (see e.g. Kleipool et al., 2008) show that an overwhelming fraction of Earth's surface has reflectance lower than 0.1-0.15 in the UV/Vis spectral range with wavelengths shorter than 500 nm. This spectral range is most important for trace-gas retrievals. The background map is constructed using a look-up table that relates top-of-the-atmosphere radiance and LER. Table 4 lists the nodes of this look-up table. The step of 0.1 in LER nodes in Table 4 is quite insufficient for calculations in the low LER range. Any interpolation with so sparse nodes would lead to high errors in the low LER range thus in the minimum LER map. The authors should add more nodes of LER for its low values and provide an estimate of interpolation errors. The paper cannot be recommended for publication without addressing this comment.
3. In Appendix C, the authors consider the spectral dependence of BRDF model parameters. Those internal parameters are used to build the minimum LER map. It would be useful if the authors would consider the spectral dependence of the final product of the developed algorithm, namely the effective cloud fraction. There is some contradiction in interpreting the spectral dependence of the effective cloud fraction. Formally, the effective cloud fraction is wavelength dependent because it is defined by

spectral quantities (Stammes et al., JGR, 2008). However the radiative transfer simulations show that the cloud fraction is nearly invariant with wavelength over a wide spectral range (Gupta et al., AMT, 2016).

4. I strongly recommend to show and analyze the cross-track dependence, i.e. dependence on VZA, of the cloud fraction. Accounting for BRDF effects on the cloud fraction would flatten the cloud fraction cross-track dependence reducing possible biases related to not accounting for anisotropic reflection of solar light from Earth's surface. Particularly, it is important for the ocean where the sun glitter can significantly affects cloud pressure retrievals. The authors are encouraged to compare their results with those in the following paper:

Fasnacht et al., A geometry-dependent surface Lambertian-equivalent reflectivity product for UV-Vis retrievals – Part 2: Evaluation over open ocean, Atmos. Meas. Tech., 12, 6749–6769, 2019.

5. In my opinion, the manuscript is too long and somewhat overloaded with technical details. More technical details could be moved in Appendices. For instance, Section 2.3.2 can be either cut down or moved to an appendix. Section 4 returns to Fig. 8-20 which were already discussed in the previous section. I would recommend to combine Sect. 3 and Sect. 4 to avoid possible duplication.

Specific comments

The title does not clearly reflect the contents of the paper. MICRU is not a common acronym. It is not clear what "background map" means. The title does not reflect that the paper is dealing with accounting for anisotropic reflection (BRDF) of solar light from Earth's surface in cloud algorithms.

P.6, L4 and elsewhere. The letter T is commonly used to denote the transmittance in radiative transfer. To avoid confusion it is desirable to select a different symbol for LER.

P.10, Fig.4. $T_{min}$ in the figure capture is not defined yet.

P.11, L.1. It is not clear how the land sea mask is applied to a nominal GOME-2 pixel? Is a land/sea fraction within a pixel known? Please clarify.

P.11, L.7. Please provide a reference to GTOPO30.

P.11, L.26. Please specify the wavelengths at which the absorbing aerosol index is defined. Its threshold value used for filtering the data depends on the wavelengths.

P.12, L.3. While doing RT computations in a spherical atmosphere the authors do not account for the atmospheric refraction. Please provide a justification for neglecting the refraction effect?

P.12, L.5. The use of a single value of 250 DU for total ozone column may not be sufficient for wavelengths within the Chappuis absorption bands in case of high solar zenith angles (Table 4 lists the angles up to 87 deg.).

P.14, L.6. Why is the glitter reflectance, r_g, defined as an independent variable? It depends on the sun-view geometry, e.g. on the viewing zenith angle which is specified as an independent variable. Please clarify.

P.19, L.23-24. The authors say " Longer time-series increase the probability of including measurements not contaminated by clouds." Please provide actual numbers that characterize the duration of time-series.

P.22, L.4. Please give a reference to FRESCO v8.

P.24, l.21-22. Fig. 8(f) shows the minimum LER residuals, T_min (stated in Line 14). However, the authors say that "… average deviations much smaller than 0.04, which is the targeted accuracy of MICRU CF". Please clarify how the LER residual of 0.04 is related to the targeted accuracy of cloud fraction.

P.25, L.2. In the discussion of Fig. 9, CF is mentioned. Please clarify what parameter (LER or CF) is shown in Fig. 9.

Section 3.2 compares cloud fractions from different algorithms. Please specify the wavelengths at which the cloud fractions are retrieved. Can the observed differences between MICRU and FRESCO/OCRA CF retrievals be due to the wavelength difference?

Section 3.3.2. Please explain why there are CF differences retrieved at different wavelengths for high values of CF. For high values of CF, possible surface effects could be neglected. Given the cloud backscatter spectrally independent, the CF values at different wavelengths seem to be same.

Section 3.4.1. What is a conclusion of the comparison of MICRU and OCRA? Do the authors attribute the differences between the algorithms to the different treatment of surface BRDF?

P.31, L.6. "… different definition of the upper threshold." What do you mean? What is the upper threshold for OCRA?

Section 3.4.2. Please formulate a purpose of comparing MICRU with three versions of FRESCO. Why do not select just the latest version of FRESCO?

---

## Short Comment (SC1) · 9 Jul 2020

The authors need to clarify definitions of relative azimuth angle (RAA) and scattering angle (SA). It seems as though the authors performed their CF calculations using correct RAA and SA in the paper; however, I noticed the definitions of RAA in Figure 1 and SA in Equation (7) do not agree with each other.

Suppose SZA = 30, VZA = 60, and RAA = 180 in Figure 1. Then, we can estimate SA = 90 from Figure 1.

On the other hand, plugging the above SZA, VZA, and RAA into Equation (7) gives:

SA = acos{-cos(VZA)*cos(SZA) + sin(SZA)*sin(VZA)*cos(RAA)}

= acos{-cos(60)*cos(30)+sin(30)*sin(60)*cos(180)} = 150.

Therefore, RAA in Figure 1 should be |VAA - (SAA - 180)| or |VAA - (SAA + 180)| instead of |VAA – SAA| where SAA is solar azimuth angle and VAA is viewing azimuth angle.

---

## Author Comment (AC1) · 8 Dec 2020

**Authors' response to the comments of Referee #1 on "MICRU background map and effective cloud fraction algorithms designed for UV/vis satellite instruments with large viewing angles" by Holger Sihler *et al.**

We would like to thank Referee #1 for the review of our submission to AMTD and for contributing helpful comments and suggestions to improve the quality and clarity of our manuscript.

For reference, the original Referee comments below are typeset in black, our responses in blue. Modifications of the original manuscript (green) are indicated in red.

Summary:
Section 1 introduces a basic motivation and explains why cloud fraction retrievals are an important ingredient to trace-gas retrievals. It further provides an overview of existing algorithms and their respective heritage. A particular importance is rightfully directed to surface contributions and to the recent developments in the field to address BRDF effects. Instrument characteristics and the cumbersome relations between MSC and PMD read-outs are well explained (section 2.1). All used auxiliary data are briefly introduced and their treatment (spatio-temporal interpolation) is justified. The main part deals with the determination of the lower threshold T_min (section 2.3). The result for the presented approach of a surface fit is shown for two example geolocations (Australia, Atlantic) at two different wavelengths (382 nm and 516.7 nm), respectively. Finally, the data sets where MICRU is compared against are briefly described (section 2.6). Section 3 shows results of MICRU for three example scenes of GOME-2 (Brazil, North America, Indian Ocean) and compares those to different FRESCO versions and OCRA. The various MICRU versions (MSC, PMD, different wavelengths) are also intercompared to highlight their differences. The comparison to FRESCO and OCRA is also extended to monthly statistics and the temporal evolution is investigated. An interpretation of the comparisons and findings for several selected individual cases and larger statistics is presented in section 4, both for the various MICRU applications and also for the comparison algorithms. Section 5 reminds that the main topic of investigation are small cloud fraction regions and recalls the novelty approach of an empirical BRDF surface model. It concludes a transferability of MICRU to other spectroscopic satellite missions and imager data. It is finally recommended to prefer the UV/blue spectral region over the red spectral region to reduce surface effects. Finally, Appendices A to E provide further valuable information for the reader.

As an overall conclusion, the manuscript is well written and structured, provides a relevant and very interesting contribution to the scientific topic addressed and therefore I recommend its publication after the general comments are addressed.

general comments:

GC1): While a lot of effort is put into the investigation and determination of the lower threshold, the description and assumptions of the upper threshold are quite brief. The authors may consider to expand a bit on the justification of the chosen simplified approach for the upper threshold and for which types of clouds it may be justified and for which not.

This issue is also addressed by GC3 and in the specific comments and technical corrections. Please see our combined responses below.

GC2): Since many cloud retrieval algorithms struggle particularly over very bright surfaces, has the MICRU performance also been tested e.g. over snow/ice conditions?

We thank the Referee for this question. Actually, snow/ice conditions were not tested because these are not discriminated by the algorithm. Hence, snow/ice conditions are potentially biased high and frequently not reliable. Certainly, this issue is worth to be improved in a revised version of the MICRU algorithm. For now, however, we included a flag for snow and ice coverage from auxiliary sensors, respectively (cf. Table 7).

We interpret the Referees comment such that we should emphasise more the capabilities of MICRU over bright surfaces such as deserts. However, we refrained from including another specific study/figure to the paper as it is already quite long. From the figures in the paper we may already follow that

- MICRU performs reliable over bright surfaces, especially at smaller wavelengths due to the smaller surface albedo.
- FRESCO and OCRA both apply data from red spectral region significantly affecting CF retrievals.
- Figure 13 partially includes measurements over East India and North America, which are significantly brighter than the measurements over Brazil. This allows for a relevant comparison between MICRU, FRESCO, and OCRA.
- The averaged biases over deserts may now be estimated from the revised global average maps in Appendix E (see below).

This issue is now included in Section 3.2 of the revised manuscript based on the new Figure 13:

Figure 13 furthermore compares MICRU results to FRESCO and OCRA. Over land, the CF maps of FRESCO L1b and v7 measurements (Figs. 13(j), (k), (m), and (n)) reveal significant positive biases in the western part of the swath. Cloud fractions larger than 20% are detected even though AVHRR and MICRU both detect no clouds. FRESCO v8 displays a significant improvement over Brazil (Fig. 13(p)), whereas CF over North America in Fig. 13(q) are still significantly biased in the west of the swath. Over East India (orbit #17907), however, all FRESCO versions are significantly biased high (Figs. 13(l), (o), and (r)). Switching to OCRA, Fig. 13(s) reveals significantly smaller positive biases of OCRA over Brazil compared to FRESCO L1b and v7. Over North America (Fig. 13(t)), however, a positive bias and scatter are significant. The biases of OCRA_fixed_albedo at MSC resolution are significantly smaller, especially over Brazil (Fig. 13(v)). Similarly, Fig. 13(x) shows significantly smaller biases over East India compared to the native OCRA results at PMD resolution in Fig. 13(u).

The influence of bright surfaces is now also discussed in Section 4.1 of the revised manuscript as specified in our answer to our following answer regarding strong aerosol events.

Furthermore, 3 more MICRU MSC maps (see below) are added to Appendix F of the revised manuscript, allowing for a more detailed comparison between different wavelengths and products, especially over the bright desert surfaces in the centre of the maps.

Figure F1: Six year average of MICRU MSC channel 2 cloud fraction measurements recorded between 1 July 2007 and 30 June 2013. Areas without data are plotted in gray.

[Figure]

Figure F2: Six year average of MICRU MSC channel 12 cloud fraction measurements recorded between 1 July 2007 and 30 June 2013. Areas without data are plotted in gray.

[Figure]

Figure F3: Six year average of MICRU MSC channel 14 cloud fraction measurements recorded between 1 July 2007 and 30 June 2013. Areas without data are plotted in gray.

[Figure]

Also the behavior over different strong aerosol events (desert dust, urban pollution etc.) could be interesting to analyze.

We agree with the Referee and appended the following passage to the fifth section of the discussion of MICRU results in Section 4.1 (p. 38, l.30)

The average CF maps Figs. F1, 14, F2, and F3 of MICRU CF at 382, 440, 670 and 757.5 nm, respectively, reveal significant systematic biases of the bright tropic deserts at 670 and 757.5 nm, notably aver Africa and Australia. At 382 and 440nm, the MICRU results over bright desert areas are significantly smaller than all other results and may, therefore, be assumed more reliable. This assumption is confirmed by Fig. 13, where MICRU CF results over North America and East India, which are significantly brighter compared to Brazil, are not significantly biased. These maps furthermore indicate systematic CF biases from anthropogenic aerosols over East Asia and residual clouds in the tropics of South America and Africa.

Events of desert dust are, however, not considered here.

GC3): The consequences of using a fixed cloud albedo of 0.8 (p2, l30; section 2.4, etc.) should be discussed in more detail in the paper. For example, how should the trace gas retrieval use a MICRU CF>1 (p19, l8)? Are the MICRU CF applicable only to tropospheric trace gas retrievals?

In the following, we would like reply to the Referee's concerns about the upper threshold, which she/he also addressed in GC1 and in the specific comments and technical corrections.

In our view, using a fixed cloud albedo as upper threshold provides several advantages for our algorithm when compared to other options:
- Compared to an empirical upper threshold, or any model with a varying cloud albedo, a fixed cloud albedo represents a more transparent choice, which is independent from instrumental effects, geo-location, observation geometry and cloud properties.
- Compared to a volumetric cloud model, a Lambertian cloud model requires less parameters and is more straight-forward to implement in RT models.
- There is quite number of algorithms for cloud fractions, cloud properties, and trace gases based on the assumption of a cloud with a fixed cloud albedo. Hence, MICRU results are applicable as input to these algorithms right away.
- Additionally, it needs to be noted that the choice of cloud model only has a small effect on the accuracy of small cloud fractions, which is the main domain of MICRU.

In order to make our choice more transparent, we apply the following changes to the manuscript:

Citations to McPeters et al., 1996 and Vasilkov et al., 2017 are added to the introduction (p.2, l.30).

Section 2.4 now reads:
In MICRU, the upper threshold $R_{max}$ is defined as the reflectance of a Lambertian surface with an albedo of 0.8 located at 7 km altitude. This simple cloud model, which was adopted from McPeters et al. (1996) and Koelemeijer et al. (2001), improves applicability to retrievals building on MICRU cloud products because cloud correction algorithms in many trace gas retrievals apply the same model (Vasilkov et al., 2017). Furthermore, the assumptions on cloud RT need to be consistent between cloud and trace gas retrievals for AMF calculation. Volumetric clouds, on the other hand, are more complex to simulate and would require more parameters, which are unknown a-priori. $R_{max}$ is assumed independent of geolocation and time and calculated applying the look-up-tables described in Sect. 2.2 and Table 4. A quantitative discussion of choosing $T_{max} = 0.8$ as a cloud albedo for an Lambertian cloud model as upper threshold is provided by Koelemeijer et al. (2001),

Ahmad et al. (2004), and Stammes et al. (2008). As a consequence, however, very bright clouds exceeding $T_{max}$ = 0.8 will result in a MICRU CF > 1. Some CF algorithms normalize CF>1 to 1, but MICRU rather provides these exceptionally high values as additional output.
It needs to be noted that instrumental degradation may introduce a systematic bias of the CF, which will be strongest for large CF. Most importantly for MICRU, the influence of the applied cloud model and on the CF accuracy decreases with CF.

GC4): A direct comparison between MICRU CF and FRESCO CF is possible because both algorithms use a fixed cloud albedo of 0.8. OCRA doesn't have this constraint, therefore the comparison between MICRU and OCRA (Section 3.2, 3.4.1., 3.4.3, 4.2, Fig. 11, Fig. 16, Fig. 17, Fig. 21 Appendix D,Appendix D, Fig. D1, Fig. D2, Fig. E5) should be extended by adding an additional 'OCRA CF_fixed_albedo' by converting the OCRA CF to a magnitude similar to the MICRU CF using the following approximation:

OCRA_CF * ROCINN_CA ∼ MICRU_CF * 0.8

OCRA_CF_fixed_albedo = OCRA_CF * ROCINN_CA/0.8

where ROCINN_CA is the cloud albedo retrieved with ROCINN. This adaptation can only be done at the MSC level because the ROCINN parameters are only provided for the MSC footprints and not at PMD level, hence for the OCRA PMD cloud fraction, this modification cannot be applied. However, the manuscript focuses on providing accurate cloud information for the retrieval of trace gases. Since the trace gases are retrieved for the MSC data, I would strongly suggest to add to the comparisons also the OCRA cloud fraction for the MSC data including the adaptation with the ROCINN cloud albedo as outlined above. In summary, I would recommend to

a) add to the comparisons also the "OCRA_CF_fixed_albedo" for the MSC data as outlined above,

We thank the Referee for this comment and added this additional comparison to the paper. We performed our evaluations adding the new term and changed figures accordingly. However, as CF at PMD output is the principal OCRA output, we still focus on the uncorrected OCRA. The changes to the manuscript are only briefly summarized here for the sake of clarity. The latexdiff-document details all modifications to the manuscript.

Changes to the manuscript:
- Description of OCRA input modified and parameter OCRA_fixed_albedo added to Section 2.6 "Comparison data"
- An additional row with OCRA_fixed_albedo is now added to Figure 11 and its caption is respectively updated
- Added OCRA_fixed_albedo figure respective to Fig 16→ new Figure 19

[Figure]

Caption: Comparison between MICRU MSC at 382 nm and OCRA_fixed_albedo based on CF data from April 2010: measurements over (a) land and (b) ocean. This comparison applies cropped data.

- Figures in Appendix D (now E) now also include OCRA_fixed_albedo.
- Added global average map of OCRA_fixed_albedo to Appendix E (now Appendix F)

We appended to the first paragraph of OCRA results (Sect. 3.4.1):

And indeed, the comparisons between MICRU MSC and OCRA_fixed_albedo in Fig. 19 result in more moderate slopes of 0.9 and 0.86 for land and ocean, respectively. Also the linear coefficient of correlation is slightly higher compared to the uncorrected OCRA values in Fig. 18.

and to the second paragraph:

The same but less pronounced behaviour may be observed for OCRA_fixed_albedo in Fig. 19.

Edits to Section 3.4.3, the discussion (Section 4.2) and the Conclusions are complex and compiled in the latexdiff.

The edits according to new Figures 11 and 12 showing VZA dependency  are detailed in our answers to Referee #2

and
b) add to the conclusions for the OCRA PMD vs MICRU PMD comparisons a statement that the modification with the cloud albedo as outlined above cannot be done at PMD level and this might be a potential source for discrepancies in the comparison.

The comparison to OCRA and FRESCO is not detailed in our conclusions, yet. We therefore added to the discussion of the comparison between MICRU and OCRA (Section 4.2):

It needs to be noted, however, that comparisons to OCRA_fixed_albedo may only be performed at MSC resolution because the required ROCINN cloud albedo values are not available in PMD resolution. Therefore, all comparisons to OCRA at PMD resolution are affected by  a different cloud albedo.

Further specific comments and technical corrections below refer to p(age) and l(ine) of manuscript amt-2020-182.pdf:

p3, l4: There are also new retrieval algorithms that combine the LER models with a geometry-dependent BRDF correction, see for example (Loyola et al., 2020) https://doi.org/10.5194/amt-13-985-2020

The Reference is added to the manuscript. Furthermore, the following subclause is added at the respective location:

, or apply a geometry-dependent LER model (Vasilkov et al., 2018; Loyola et al., 2020)

p3, l32: p5, l1; and p22, l11: The third version of OCRA was not applied to TROPOMI but to OMI. The fourth version of OCRA (Loyola et al., 2018) https://doi.org/10.5194/amt-11-409-2018 is applied to TROPOMI, but this reference is missing in all three paragraphs.

The Reference to (Loyola et al., 2018) is added to the manuscript.

At the first occurrence on page 3
The fourth version of OCRA applied to TROPOMI is described by Loyola et al. (2018).

is now added.

On page 5,
(Lutz et al., 2016)
is changed to
(Loyola et al., 2018)

On page 22,
(Loyola et al., 2018)
is appended to the sentence starting in line 11.

p4, l34: To be more precise, for the TROPOMI/S5-P mission OCRA is part of the S5P L2 CLOUD product and FRESCO is an auxiliary cloud product used for the S5P L2 NO2, ALH, CH4 and O3 profile products.

We agree with the Referee.
The operational cloud fraction algorithms for these missions are OCRA (Lutz et al., 2016) and FRESCO (Wang et al., 2008), respectively.
Is changed to
The operational cloud fraction algorithms for TROPOMI/S5P is OCRA (Loyola et al., 2018) and FRESCO is an auxiliary cloud product used for selected level 2 products (Wang et al., 2008), respectively.
in the revised manuscript.

p6, l23: Could you specify, which data version was acquired from EUMETSAT? Was it the reprocessed AC-SAF data set?

For the spectral data, reprocessed and near-real-time data based on level 0 to 1b processor version 5.3 is applied.

To specify this, we changed
All spectral data is contained in the level 1b (L1b) data provided by EUMETSAT.
to
All spectral data is contained in the level 1b (L1b) data (processor version 5.3) provided by EUMETSAT.
in the revised manuscript.

p10, Figure 4: The last sentence of the figure caption might be confusing and binning #1 might be misunderstood as native resolution. Please rephrase to avoid confusion.

We agree with the Referee that the term "native" may be misleading. The last sentence of the caption is rephrased to (also see comment by Referee #2 on same caption)
The highest resolutions for MSC and PMD lower threshold maps are 0.1°×0.05° and 0.0125°×0.05°, respectively, and denoted binning #1 as in Table 6.

p10, l5: Forward and backward scans were not mentioned before. A short introduction and explanation, why backward scans are discarded could be beneficial for the reader here.

We agree that the backward and forward scans should be introduced. However, we think that this information should not be provided in Section 2.1.2 but rather in Section 2.1.1 "GOME-2 data". We replace (p. 6, l. 25)
One nominal swath consists of 24 MSC or 192 PMD pixels, respectively.

by
One swath consists of 32 MSC or 256 PMD pixels, respectively. One swath is divided into a forward and a backward scan. The scanner turns three times faster during the backward scan resulting in three times larger pixels, which are discarded altogether in the following. Hence remaining are 24 MSC or 192 PMD pixels, respectively.

p15, caption of Figure 7: In addition to observation geometry and wind speed, doesn't the sun glitter contribution also depend on geolocation and time? E.g. in summer the sun glitter appears at different latitudes than in winter. Please clarify.

We agree that this formulation may be misleading. We erased observation. The revised caption now reads
the sun glitter contribution depends on geometry and wind speed
which is a more general statement because geometry naturally depends on geolocation, observation geometry and time

p15, l8: Same reference is given twice: ". . .swath edge in Figs. 8(d,e) and 8(d,e)"

The references are updated to
Figs. 8(d,e) and 9(d,e)
in the revised manuscript.

p15, l9-13: Additional tests are described but discarded because of inferior results in a number of case studies. In my opinion it could be elaborated a bit more on the reasoning of selecting these choices in order to give the reader a bit more background information.

We agree with the Referee. In the revised manuscript, the paragraph now reads:

The cosine normalises the parameter improving the fit stability. As tests, we replaced the empirical term either with the precise Li-dense kernels, a reduced $\cos \theta_r$ term for surface effects, $\csc \theta_s$, or $\cos^2 \theta_s$ but all of these test resulted in less accurate surface fits and, hence, increased $T_{min}$ noise compared to our final choice.

p16, table 5: Why is degradation for MSC channels 12, 13, 14 constrained to 0? Please add short explanation.

In the revised manuscript, the table footnote is changed to
∗: Degradation constrained to 0 for MSC channels 12, 13, and 14 (see text).

In the main body of the manuscript (page 16, line 4), we changed
[…] Clearly, this model is […]
to
[…] The parameters $a_t$ and $a_{a1}$ are constrained to 0 for MSC channels 12, 13, and 14 as preliminary evaluations showed that degradation can be neglected for these channels, which furthermore improved signal-to-noise. Clearly, the above model is […]

p17, Figure 8(b): Is there a mixup, where in the text ". . . measurement set Omega_0 (red) and finally fitted set Omega_I (blue)", the colors red and blue should be swapped? The figure legend says that the fitted data are the red points.

It is true that the colors were swapped. The figure caption is revised to
(b) 3D representation of measurement set $\Omega_0$ (blue) and finally fitted set $\Omega_I$ (red)

Figures 8e and 9e: The surface fits for MSC 2 over Australia (8e) and MSC 10 over Atlantic (9e) show very different patterns. The former is strongly pronounced in the western half of the swath and becomes stronger in time while the latter is restricted to the east half of the swath and present for all years 2007-2013. Is there a simple explanation for the driving factor behind these differences (except geolocation and wavelength)? Edit: Ok, the pattern of the latter is later addressed on p24, l30 and assigned to sun glitter but what could be the explanation for the increasing trend in the first example?

We thank the Referee for this particular comment. Both case studies differ not only in geolocation and wavelength, but also surface type and MSC band.

In order to clarify this issue, the first paragraph of Section 4.1 is extended in the revised manuscript: The periodic structures in the west of Fig. 8(e), which are also visible but less pronounced in Fig. 9(e), may be mostly attributed to the anisotropic reflectivity over land. Figure 11 suggests that this anisotropy may be underestimated by the operational CF products. In Fig. 8(e), there seems to be an upward trend and a shift of the apex towards east over time. Both may be attributed to the degradation of this particular MICRU channel (cf. Figs. D1(a) and (d)). Furthermore, also local changes of land use, vegetation type, or precipitation climatology may lead to shifts in the lower threshold, which would be linearised by the applied $T_{min}$-model. These additional local trends may be reproduced by the MICRU algorithm as Figs. C1(d) and (g) illustrate.

p19, l8: How are MICRU CFs > 1 treated? Are they cropped to 1 or flagged? Please clarify.

In this study, "raw MICRU CFs" with values <0 and >1 are applied unless noted otherwise. The public data contains data normalised to CF∈[0,1]. The files furthermore contain the raw MICRU CFs, which are uncropped. The flags contained in the data are compiled in Table 1 and do not contain an additional flag indicating normalisation to avoid redundancy.

p19, section2.4: The upper threshold is assumed to be fixed with a cloud albedo of 0.8 at an altitude of 7km without a dependency on geolocation and time. While a lot of effort has been put into the various dependencies for the lower threshold, the assumptions on the upper threshold are few. It is argued that MICRU focuses on small CFs, but how valid is e.g. the assumption of cloud albedo = 0.8 in the case of an optically very thin cloud close to the surface or very high in the troposphere? Could it be specified for which type of clouds this simplified assumption on the upper threshold is valid and justified and for which not?

This issue is also addressed by GC1 and GC3. Please see our responses above.

p21, l24: The sentence "probably the most commonly used CF product" is not correct. Most of the operational GOME-2 AC-SAF trace gas products are based on the OCRA/ROCINN algorithm.

We replaced
probably the most commonly used CF product
by
widely-used product
in the revised manuscript.

p22, l14: "and OCRA applies a volumetric (or scattering) cloud model". Technically, the scattering cloud model is only relevant for the ROCINN part of the OCRA/ROCINN algorithm combination, which retrieves cloud top height and cloud optical thickness. For the OCRA cloud fraction using the color space approach, the relevant assumption is a spectral independence of a fully cloudy

reflectance across the UV/VIS wavelength range. Therefore, "and OCRA applies a color space approach for the upper threshold (Lutz et al., 2016)" seems more fitting.

We agree with the Referee. The sentence in the revised manuscript now reads:
The selected cloud products define the upper threshold differently. FRESCO applies a Lambertian (or reflecting) cloud model – like MICRU – and OCRA applies a color space approach for the upper threshold (Wang et al., 2008; Lutz et al., 2016).

p23, l1-2: The sentence "These biases propagate into trace gas retrievals if normalized CF data are applied" may apply to FRESCO but is not correct for OCRA/ROCINN. Any possible bias on the OCRA CF will be compensated in the ROCINN cloud albedo and therefore possible bias will be not propagated into trace gas retrievals (Loyola et al., 2007)
http://dx.doi.org/10.1109/TGRS.2007.901043

We thank the Referee for this comment. We revised our citation of Loyola et al. (2007) who state that "ROCINN partly compensates the cloud fraction underestimation from OCRA by overestimating the cloud-top albedo and by underestimating the cloud-top height." In the present study, however, it is observed that OCRA systematically *over*estimates CFs towards the west. In Figure 11 of the revised manuscript, it may be observed that ROCINN actually compensates some of the overestimation, but certainly not all.

According changes to the manuscript are detailed in our answers to Referee #2 suggesting a detailed study of the VZA dependence, in which we included both OCRA and OCRA_fixed_albedo results.

p31, Figure 16: Title and axis labels say "PMD ch 1" and "PMD-PP cloud fraction @ 382 nm" while the figure caption reads "MICRU MSC at 440 nm". Please clarify.

The caption now reads MICRU PMD at 382nm

p31, l7: see comment to p22, l14

See our answer above.

p37, Figure 21: Is there a reason that subplot (a) shows all three FRESCO versions L1b , v7 and v8 while in subplot (b) only the FRESCO v8 are shown?

Yes, there is. From Figure 19 it is clear, the older two FRESCO versions are much more affected by the LOS-dependency than FRESCO v8. In order to reduce complexity of Figure 21(b) (Figure 24 in the revised manuscript), of the three different cloud products (OCRA, FRESCO, and MICRU) only the version least affected by the LOS-dependency are compared. Therefore, FRESCO L1b and v7 are omitted.

We appended to the caption of the revised figure (now Fig. 24):
FRESCO L1b and v7 results are omitted in (b) for the sake of clarity.

p37, Figure 21: Are sun glitter scenes included in the 15th percentile cloud fractions shown? The difference for the OCRA East might be due to the fact that sun glitter appears only in the east half of the swath and OCRA sets scenes affected by sun glitter to zero by default. This might contribute to the low bias.

We guess, the Referee refers to the differences between OCRA East and OCRA Nadir in subfigure 21(b) (Figure 24 in the revised manuscript). In that Figure, however, ocean scenes are omitted

(denoted: land only) and, therefore we do not believe that the Referees rationale sufficiently explains the observed biases between OCRA viewing angles.

Please note that the revised manuscript applies OCRA_fixed_albedo CFs in Figure 21(b), which reveal a much smaller but still significant gradient between east and west compared to the OCRA PMD values.

p38, l19-20: Please note that the VZA dependence in OCRA is also evaluated empirically with a monthly temporal resolution.

We agree with the Referee. We added to the specified paragraph:

It needs to be noted, however, that also OCRA applies an empirical VZA correction based on global monthly means (Lutz et al., 2016).

p40, l13-15: It is true that the OCRA sun glitter removal at areas with larger theta_r may be positively biased (particularly visible in the left swath of Figure 11(u)). However, it could also be pointed out here that in regions of very strong sun glitter (yellow in Figure 11(c)), OCRA seems to properly account for this effect (visible in the right swath of Figure 11(u)). Furthermore, OCRA includes a sun glint flag.

We agree with the Referee. We added to the revised manuscript

At regions of very strong sun glitter (yellow areas in Fig. 13(c), OCRA seems to properly account for this effect on PMD resolution.

and

and that also OCRA contains a sunglint flag

p45, l6: This is a very interesting detail. Could this terrestrial sun glitter signal over the Amazonas be related to high oriented ice crystals as suggested by Marshak et al. based on EPIC/DSCOVR? (Terrestrial glint seen from deep space: Oriented ice crystals detected from the Lagrangian point https://doi.org/10.1002/2017GL073248)

We thank the Referee for introducing this interesting aspect. However, we do not believe that we found a sign of oriented ice crystals due to two reasons: (a) the signal discussed by Marshak et al. would increase the radiance, and hence, would be filtered by the lower threshold retrieval algorithm, and (b) the spatial structures we observed correspond to the location of water bodies, which we consider spatially only weakly correlated to the location of atmospheric ice platelets due to their significant altitude and the temperature profile in the Amazon Basin.

technical corrections:

p2, l25: change "the time dependent the" to "the time dependency of the"

Done.

p2, l28: change "a-priory" to "a-priori"

Done.

p2, l30: change "albedo of 0.8 Stammes et al. (2008) rendering" to "albedo of 0.8 as in Stammes et al. (2008), rendering"

Done.

p7, l16: change "channels 2,5, 10" to "channels 2, 5, 10" (blank space missing before 5)

Done.

p16, l21: change "a-priory" to "a-priori"

Done.

p20, l1: change "to form a complete parametrisations" to "to form complete parametrisations"

Done.

p20, l17: change "prior the $T_{min}$ retrieval" to "prior to the $T_{min}$ retrieval"

Done.

p22, l4: something is missing in "is increased to effects of". Maybe "is increased to cover effects of"?

The sentence in the revised manuscript now reads
The resolution at the coast and over specific regions is increased to reduce interference from different surface types within one GOME-2 pixel (Wang et al., 2018).

p22, l15: Sentence "Some FRESCO both and OCRA. . ." sounds weird. Is the following meant: "Both FRESCO and OCRA. . ."?

In the revised manuscript, the sentence begins with
Both FRESCO and OCRA [..]

p24, l17: change "in discrete boxed defined" to "in discrete boxes defined"

Done.

p29, l5: change "evaluation" to "evaluations"

Done.

p29, l6: change "and $m = 1$ 519nm" to "and $m = 1$ at 519nm"

Done.

p31, l6: change "differently" to "different"

Done.

p32, l19: "Figs. 19(d) and (g))": Shouldn't this be "Figs. 19(c) and (f)"?

Thanks for pointing out this error. Actually, largest biases are found in the west. Hence, the beginning of the sentence is now changed to

Largest biases are observed towards west for FRESCO L1b and v7 (Figs. 19(a) and (d)), [..]

p37, caption of Figure 21: first line ends with "(b): comparison between selected MICRU", while it should be "(a): comparison between selected MICRU".

Done.

p38, l32: change "retried" to "retrieved"

Done.

p40, l5: "(Fig. 21(c))": Fig. 21 has no subplot (c). Is Fig. 21(b) meant?

Yes, done.

p40, l24: "at coasts an inland": Is "at coasts and inland" meant?

Yes, done.

p40, l30: "as investigated by Fig. 19)." The closing bracket has no opening bracket.

Done, closing bracket erased.

p48, caption of Figure D2: "circled values in (c)": The circles are in (a)

Done.

p49, l5: at the end of line change "onky" to "only"

Done.

---

## Author Comment (AC2) · 8 Dec 2020

**Authors' response to the comments of Referee #2 on "MICRU background map and effective cloud fraction algorithms designed for UV/vis satellite instruments with large viewing angles" by Holger Sihler *et al.**

We would like to thank Referee #2 for the review of our submission to AMTD and for contributing helpful comments and suggestions to improve the quality and clarity of our manuscript.

For reference, the original Referee comments below are typeset in black, our responses in blue. Modifications of the original manuscript (green) are indicated in red.

Review of "MICRU background map and effective cloud fraction algorithms designed for UV/vis satellite instruments with large viewing angles" by Sihler et al.

The manuscript describes a model for accounting for anisotropic reflection of solar light from Earth's surface in an effective cloud fraction algorithm designed for UV/Vis satellite instruments. Results of the application of the algorithm to GOME-2 data are compared with other cloud fraction algorithms. Appendices provide technical details of the developed algorithm. The manuscript is clearly relevant for AMT. Even though the material is not a significant advance in remote sensing of clouds it could be published to document the GOME-2 cloud fraction algorithm in the literature. The abstract provides a concise and complete summary of the paper. The earlier work is properly credited. I recommend publication of this manuscript only after major revisions which address the following comments.

**General comments**

1. The authors do not clearly state what are the main improvements of the proposed algorithm as compared with the existing cloud algorithms which also accounts for surface BRDF. It would be useful to summarize those improvements in Conclusions.

To the knowledge of the authors, there are currently no cloud products for GOME-2 featuring BRDF effects for the background map. The improvements of this feature of MICRU compared to OCRA and FRESCO products are investigated in the manuscript.

MICRU features a background map computed from the measurements themselves also considering two parameters for degradation. These measures have the potential to improve the accuracy of small CF significantly.

The proposed MICRU implementation also features cloud fractions measured at different acquisition times minimizing systematic errors due to spatial aliasing, which is typical for GOME/GOME-2 instruments.

Hence, MICRU is a universal algorithm, which may be consistently applied to other spectrometers and also imaging instruments as well.

We edited the conclusions of the revised manuscript:

The unique feature of MICRU is the application of an empirical BRDF surface model accounting for viewing angle dependencies in the cloud retrieval. The paper demonstrates that MICRU CF depend significantly less on VZA compared to other available CF products for GOME-2 and,

hence, are significantly more accurate. MICRU determines the lower threshold from the measurements themselves furthermore reducing biases due to calibration and degradation issues.

2. Low values of LER are of the primary interest for the construction of a minimum LER map (background map) which is the core of the developed algorithm. The existing surface reflectance data sets (see e.g. Kleipool et al., 2008) show that an overwhelming fraction of Earth's surface has reflectance lower than 0.1-0.15 in the UV/Vis spectral range with wavelengths shorter than 500 nm. This spectral range is most important for trace-gas retrievals. The background map is constructed using a look-up table that relates top-of-the-atmosphere radiance and LER. Table 4 lists the nodes of this look-up table. The step of 0.1 in LER nodes in Table 4 is quite insufficient for calculations in the low LER range. Any interpolation with so sparse nodes would lead to high errors in the low LER range thus in the minimum LER map. The authors should add more nodes of LER for its low values and provide an estimate of interpolation errors. The paper cannot be recommended for publication without addressing this comment.

We thank the Referee for highlighting this issue. We conducted some test in beforehand and we were also surprised, how small the error caused by this simplification actually is. We performed the reflectance to LER conversion for one orbit of GOME-2 MSC data for four different wavelengths using two different LUTs, one as described in the paper and one, where 9 additional nodes between 0 and 0.1 at steps of 0.01 are included. The differences of the inverted LER are:

[Figure]

Obviously, the error introduced by linear interpolation applying the LUTs in question is always smaller than 0.001. The error for the CF must be even smaller, as the same systematic error is performed twice (background map calculation and its later Look-up), which should cancel out almost completely due to a change in sign. Furthermore, we would like to note that interpolation errors applying basic linear interpolation for strictly monotonic functions - as the TOA radiance to LER dependence – are small. Hence, we decided against performing our calculation with larger LUTs but rather informing the reader of the interpolation error.

We add the following paragraph to Section 2.2 the revised manuscript:

It needs to be noted that the resolution of the LUTs in LER direction may appear rather coarse. However, the difference of the obtained results to preliminary RT computations featuring a 10 times higher resolution were found to be < 0.001 in the UV and even one order of magnitude less in the red spectral region.

3. In Appendix C, the authors consider the spectral dependence of BRDF model parameters. Those internal parameters are used to build the minimum LER map. It would be useful if the authors would consider the spectral dependence of the final product of the developed algorithm, namely the effective cloud fraction. There is some contradiction in interpreting the spectral dependence of the

effective cloud fraction. Formally, the effective cloud fraction is wavelength dependent because it is defined by spectral quantities (Stammes et al., JGR, 2008). However the radiative transfer simulations show that the cloud fraction is nearly invariant with wavelength over a wide spectral range (Gupta et al., AMT, 2016).

We thank the Referee for this interesting question. We, however, doubt that our results are suitable to address this issue for a number of reasons:
- MICRU is optimised for retrieving small CFs at high accuracy over the entire swath of GOME-2. Larger CF have larger errors.
- The applied GOME-2 level 1b suffers from degradation and its absolute accuracy is not ideal for such comparisons.
- The lower threshold maps are not absolute LER as they also account for degradation effects, which differ between MICRU channels.
- GOME-2 is a scanning instrument resulting in different measurement PSFs at different wavelengths, that is spatial aliasing.

4. I strongly recommend to show and analyze the cross-track dependence, i.e. dependence on VZA, of the cloud fraction. Accounting for BRDF effects on the cloud fraction would flatten the cloud fraction cross-track dependence reducing possible biases related to not accounting for anisotropic reflection of solar light from Earth's surface. Particularly, it is important for the ocean where the sun glitter can significantly affects cloud pressure retrievals. The authors are encouraged to compare their results with those in the following paper:
Fasnacht et al., A geometry-dependent surface Lambertian-equivalent reflectivity product for UV-Vis retrievals – Part 2: Evaluation over open ocean, Atmos. Meas. Tech., 12, 6749–6769, 2019.

We thank the reviewer for this valuable suggestion. We performed an analysis of the statistics depending on viewing direction and included the following two figures to the revised manuscript.

New figure 11 (land)

[Figure]

Caption: Comparison of viewing angle dependence of small CFs over land between the 55° parallels: (a) MICRU MSC at 440 nm, (b) OCRA MSC, (c) OCRA_fixed_albedo, (d) FRESCO

L1b, (e) FRESCO v7, and (f) FRESCO v8. Statistics based on April 2010 data.

New figure 12 (ocean)

[Figure]

Caption: Same as Fig. 11 but over ocean. The influence of sun glitter is evident for positive VZA for most products, even though measurements flagged sunglint risk (Table 7) are filtered.

Accordingly, we added to Section 3.1:

Another estimate of the residual VZA-dependence may be assessed by analysing the cross-track dependence of the lower CF accumulation point displayed in Figs. 11(a) and 12(a) for land and ocean surfaces, respectively. Over land, the small CF accumulate between -0.02 and 0.03 almost evenly over the entire swath. Over ocean, small CF are slightly more scattered. The distribution dilutes significantly and reveals a slight positive bias towards west (negative VZA in Fig. 12).

and to Section 3.4.1:

An additional view on the VZA-dependence of MICRU and OCRA CF is provided by Figs. 11(a)–(c) and 12(a)–(c) for land and ocean surfaces, respectively. Over land, the accumulation points of small OCRA MSC are significantly biased high for all negative VZA. The albedo correction for OCRA_fixed_albedo (Fig. 11(c)) improves the situation significantly confining the accumulation point to CF < 0.1, which is still significantly larger than the MICRU CF in Fig. 11(a). Over ocean, both OCRA investigated versions reveal a significant bias from sun glitter for positive VZA in Figs. 11(b) and (c). Towards the west (negative VZA), the lower accumulation point of OCRA_fixed_albedo is more populated than MICRU.

and to Section 3.4.2:

Figures 11(d)–(f) and 12(d)–(f) detail the VZA-dependence of the three FRESCO versions over land and ocean, respectively. Over land, the accumulation points of small CF are significantly biased high for negative VZA, especially for FRESCO L1b and v7. The distribution of FRESCO v8 CF (Fig. 11(f)), however, is almost independent of VZA. Over ocean, the differences between the

FRESCO versions are small and the bias from sun glitter is again significant, which is consistent with other results.

While we acknowledge the results by Fasnacht et al., they may not easily be compared to our results for two reasons. Firstly, MICRU retrieves CF using a lower threshold, which may not be readily compared to the GLER retrieved by the cited work. Secondly, it is not straightforward to compare OMI and GOME-2 due to their different orbital parameters and measurement times. Unfortunately, the authors of Fasnacht et al. responded to our request of GOME-2 data that they did not yet apply their algorithm to GOME-2.

However, we include a reference to Fasnacht et al. to our short review in the introduction (p.4, l. 10) to acknowledge their results.

5. In my opinion, the manuscript is too long and somewhat overloaded with technical details. More technical details could be moved in Appendices. For instance, Section 2.3.2 can be either cut down or moved to an appendix. Section 4 returns to Fig. 8-20 which were already discussed in the previous section. I would recommend to combine Sect. 3 and Sect. 4 to avoid possible duplication.

We understand the Referees concerns regarding the structure and length of the manuscript. We agree that it is not brief, but we believe its length is justified because we compare our results to several other products. This careful comparison is at least partially motivated by our activities in the verification of the operational TROPOMI/S5P algorithms. Our feedback from the TROPOMI/S5P and S5 community is quite positive and, therefore, we would like to not shorten our result section further.

Furthermore, we refrained from combining Sects. 3 and 4 because we see significant scientific advantage in separating the result from their discussion because we want to base our thorough discussion on all results, which, logically, need to be presented before. Short comments in Section are merely signposts to guide the reader.

As suggested by the Referee, the iterative surface fitting section (Sect. 2.3.2) is moved to the Appendix in the revised manuscript. Citations to this section are edited from Sect. to Appendix, respectively.

**Specific comments**

The title does not clearly reflect the contents of the paper. MICRU is not a common acronym. It is not clear what "background map" means. The title does not reflect that the paper is dealing with accounting for anisotropic reflection (BRDF) of solar light from Earth's surface in cloud algorithms.

We opted for a brief title and omitted the specific feature of MICRU background maps:
MICRU: a sophisticated effective cloud fraction algorithm designed for UV/vis satellite instruments with large viewing angles

P.6, L4 and elsewhere. The letter T is commonly used to denote the transmittance in radiative transfer. To avoid confusion it is desirable to select a different symbol for LER.

We agree with the Referee, that the choice of the symbol $T$ for LER was not ideal and may also lead to confusion with the lower threshold $T_{min}$. Instead of replacing $T$ we decided to remove this symbol from the manuscript altogether and replacing it by the term (the) LER as often applied in AMT articles.

P.10, Fig.4. T_min in the figure capture is not defined yet.

The last sentence of the caption is rephrased in the revised manuscript (also see comment by Referee #1 on same caption):

The highest resolutions for MSC and PMD lower threshold maps are 0.1°×0.05° and 0.0125°×0.05°, respectively, and denoted binning #1 as in Table 6.

P.11, L.1. It is not clear how the land sea mask is applied to a nominal GOME-2 pixel? Is a land/sea fraction within a pixel known? Please clarify.

The algorithm to determine the land/sea fraction in each pixel is described in the second and third paragraph in Sect. 2.1.2 (P. 10). The second paragraph of Section 2.1.2 of the revised manuscript is changed to:

MICRU features a separate $T_{min}$ parametrisation for measurements over land and ocean, respectively. An accurate description of the land and water transition is therefore crucial for the accurate interpolation of $T_{min}$ at coasts. An algorithm specifically developed for MICRU derives the fraction of water and land in each satellite pixel at high resolution. As input, the land sea mask (LSM) compiled from revision 679 of the GSHHG coast line database (Wessel and Smith, 1996; NOAA, 2018) is applied. The polygon data from intermediate GSHHG resolution neglecting polygons smaller in area than one GOME-2 pixel is first sampled at 0.1°×0.05° and 0.0125°×0.05° for MSC and PMD, respectively, and then convolved with the corresponding PSF (cf. Figure 4). The convolution yields a global map of fractional land cover ranging between 0 and 1 representing complete water and land coverage, respectively. Hence, $T_{min}$ values for land and ocean may be interpolated for each satellite pixel based on the convolved land cover map. The interpolated fractional land cover values are later also used for flagging (Sect. 2.5.3).

where the sentence

The LSM is processed at eight times higher longitudinal resolution for PMD compared to MSC taking advantage of the smaller PMD pixel size.

is erased.

P.11, L.7. Please provide a reference to GTOPO30.

Citation to https://doi.org/10.5066/F7DF6PQS added.

P.11, L.26. Please specify the wavelengths at which the absorbing aerosol index is defined. Its threshold value used for filtering the data depends on the wavelengths.

We agree with the Referee that the wavelength should be specified in the manuscript. The revised manuscript now includes:

The reflectances used for the determination of the AAI at MSC resolution are centred at 340 and 380 nm. For the AAI at PMD resolution, PMD-PP bands 4 and 6 at 338 and 382 nm are applied, respectively.

P.12, L.3. While doing RT computations in a spherical atmosphere the authors do not account for the atmospheric refraction. Please provide a justification for neglecting the refraction effect?

Actually, SCIATRAN is able to do calculations in spherical geometry either with or without refraction. For our LUT calculations, refraction was turned on. Furthermore, we would like to note that the influence of refraction on our calculation, which do not include the limb geometry, is minimal.

In order to clarify this setting, we appended
and accounting for atmospheric refraction
after h=0m (p.12, l.4).

P.12, L.5. The use of a single value of 250 DU for total ozone column may not be sufficient for wavelengths within the Chappuis absorption bands in case of high solar zenith angles (Table 4 lists the angles up to 87 deg.).

We thank the Referee for addressing our simplification of the RT. Our motivation is to decrease the amount of input information on MICRU and, hence, to actually provide an independent product/piece of information. In order to estimate the influence of variations of the ozone column on the CF retrieved within the Chappuis band, we want to perform a worst case estimation:

- The MICRU channels most affected by the Chappuis band absorption are PMD channels 6 and 14 measuring between 568-613 nm, where the O3 cross section $\sigma=4.8e-21$ cm^2/molec. For MSC channels 11 and 12, the cross section is less than 2e-21.
- The worst case air mass factor up to 55° latitude computes from an SZA=86 and VZA=54 using AMF = 1/cos(SZA) + 1/cos(VZA) to 16
- RT calculations are performed using 250 DU=6.75e18 molec/cm^2. We will compare the calculated TOA reflectances to those at 500DU=6.75e18. Multiplied by the AMF, this yields slant columns of S1=1.1e+20 and S2=2.2e+20 molec/cm^2, respectively.
- The Transmission at 250DU then computes to T(250DU)=exp(-S1*\sigma)=0.59 and T(500DU) = 0.35, respectively. The difference corresponds to a reduction to approximately 60% TOA reflectance compared to a measurement affected by 250DU.
- For MICRU, the goal is to achieve 4% accuracy on the lower threshold, which is the minimum TOA reflectance. At 600nm, the minimum TOA reflectance is below 0.1 for 80% of the measurements (determined from an April example orbit, partially over Africa).
- With an average upper threshold of 0.73, this can yield a CF error of 0.1*0.6/0.63 = 10% for the lower threshold. This exceeds the 4% error goal of MICRU for the accuracy of small cloud fractions.

It needs to be noted, that this 10% error is a conservative estimate of the error caused by Ozone column variations. The issue is compensated by the following factors:

- The AMF of 16 is quite an extreme case, which only seldomly occurs for tropospheric measurements at latitudes lower than 55°.
- The column variation of Ozone at the same latitude are usually much smaller than 250DU. Exception: Ozone hole conditions.
- The empirical approach of MICRU may also compensate systematic influences of the RT. For example, the offset may compensate an errorenous average ozone column and the fitted parabola may account for its influences for larger viewing angles. Furthermore, the manuscript discusses that systematic instrumental effects have, for GOME-2, a larger effect on the CF accuracy compared to RT errors.

In total, we only find a negligible effect in the MICRU results. The correlation matrices for all PMD channels corresponding to Figures D2(a) and (b) are

[Figure]

Here, channels 6 and 14 do not show a significant variations compared to their respective neighbors. Reduced linear correlation with increased channel difference may be explained by spatial aliasing. The CF intercepts are mostly less than 1% and all well below the 4% limit.

In order to convey our considerations to the reader, the revised manuscript is edited to

The $O_3$ column is fixed to 250 Dobson Units (DU) in order to reduce the number of required input parameters. This simplification may affect MICRU retrievals within the Ozone Chappuis band, most notably PMD channels 6 and 14, and, to a lesser extent, MSC channels 11 and 12. Preliminary results, however, showed that errors are on average negligible as the empirical approach of MICRU reduces the influence of systematic errors.

P.14, L.6. Why is the glitter reflectance, r_g, defined as an independent variable? It depends on the sun-view geometry, e.g. on the viewing zenith angle which is specified as an independent variable. Please clarify.

MICRU applies an empirical surface reflectance model, where the parameter specifying the contribution by sun glitter is an independent parameter. The sun glitter itself is a function of viewing geometry and wind speed, and, hence, linear independent from the other three independent variables: time, VZA, and scattering angle.

P.19, L.23-24. The authors say " Longer time-series increase the probability of including measurements not contaminated by clouds." Please provide actual numbers that characterize the duration of timeseries.

This issue is thoroughly discussed by Krijger et al., 2007. We include this citation in the revised manuscript. The paragraph at the specified location now reads in the revised manuscript:

Temporally, larger subsets should be favoured over smaller ones unless there are significant changes of surface properties or the instrument response degrades much differently than considered in the model (Sect. 2.3.1). For example, for MSC binning 1 there are 400 equatorial bins over land with 15 or less measurements considered cloud-free by the $T_{min}$-retrieval despite applying a study period of 77 months. Longer time-series increase the probability of including measurements not contaminated by clouds (Krijger et al., 2007).

P.22, L.4. Please give a reference to FRESCO v8.

Actually, there is no peer reviewed paper about FRESCO v8 for GOME-2. The algorithm, however, is described in a EUMETSAT report about FRESCO+ v2 using DLER, which is the same as FRESCO v8 at KNMI and in our work. The report is available from the TEMIS website

Wang, P., Tuinder, O., and (KNMI), P. S.: FRESCO+ version 2 for GOME-2 Metop-C processing, Internet, http://www.temis.nl/fresco/frescopv2_metopc_WP1_report_20181026.pdf, last access: 16 November 2020, 2018.

and we appended the reference (Wang et al., 2018) to line 4 on page 22 of the revised manuscript.

P.24, l.21-22. Fig. 8(f) shows the minimum LER residuals, T_min (stated in Line 14). However, the authors say that "... average deviations much smaller than 0.04, which is the targeted accuracy of MICRU CF". Please clarify how the LER residual of 0.04 is related to the targeted accuracy of cloud fraction.

We thank the Referee for this helpful comment. We agree that this statement may be confusing for the reader. We decided that the subclause

, which is the targeted accuracy of MICRU CF

is actually redundant and removed it from the revised manuscript.

P.25, L.2. In the discussion of Fig. 9, CF is mentioned. Please clarify what parameter (LER or CF) is shown in Fig. 9.

We thank the Referee for pointing this out. We replaced CF by LER in the revised manuscript.

Section 3.2 compares cloud fractions from different algorithms. Please specify the wavelengths at which the cloud fractions are retrieved. Can the observed differences between MICRU and FRESCO/OCRA CF retrievals be due to the wavelength difference?

We thank the Referee for this thoughtful consideration. However, we draw a different conclusion from our results, because CF from all MICRU channels are consistent (Figures 15, E1, and E2). The intercomparison of MICRU results furthermore illustrates that noise increases towards larger wavelengths due to larger reflectivity and increased uncertainty. FRESCO and OCRA both apply measurements from the red spectral region, which may partially explain the differences. Furthermore, as presented in Section 3.2, MICRU performs much more reliable over cloud-free scenes compared to FRESCO and OCRA because it applies a more elaborate parameterisation of the lower threshold, which is an effect independent from wavelength.

In order to clarify this issue, the applied wavelengths are added to Section 3.2 of the revised manuscript as requested by the Referee.

MICRU MSC and PMD results are specifically obtained at 440 and 460nm, respectively. OCRA results are based on PMD measurements between 321 and 804nm. FRESCO applies the O2A-band at 757.5nm.

and added to the third paragraph of Section 4.1 (Discussion) of the revised manuscript:

This effect is almost independent from wavelength and consistent for all MICRU channels (cf. Fig.

15 and Appendix E).

Section 3.3.2. Please explain why there are CF differences retrieved at different wavelengths for high values of CF. For high values of CF, possible surface effects could be neglected. Given the cloud backscatter spectrally independent, the CF values at different wavelengths seem to be same.

We agree with the Referee that this issue should be addressed in the manuscript. We are certain that this effect is dominated by data and instrumental deficiencies. We added to Section 3.3.2 of the revised manuscript:

Figure 15 furthermore indicates, that the CF slope differs between MICRU channels, which is discussed in Sect. 4.1.

We furthermore modified the respective paragraph in Section 4.1:

Another aspect of the MICRU MSC channel intercomparison are differencies at different wavelengths for high values of CF and, hence, slopes deviating from unity as, for example, shown in Fig. 15(a). CF at 382 nm are biased high with respect to those retrieved at 440 nm while the intercept at zero CF is negligible. Hence, the definition of $T_{max}$ apparently deviates between MICRU channels, which should be independent from surface effects. Figure E2(c) comprehensively compares the slopes of all MICRU channels. There is a significantly biased slope for MSC channels 1–4 retrieved at 389.7 and below. This step between MSC channels 4 and 5 may be attributed to the application of different GOME-2 bands, specifically bands 2B and 3, from which the MICRU channels are extracted (cf. Table 1). Hence, we conclude that differences between MICRU channels at high CF values are dominated by instrumental effects and calibration deficiencies of the input data. We would like to note that we observed also the CF accuracy degrading near GOME-2 band edges when fine-tuning the MSC channel definitions (Table 1). The degradation depends only weakly on kernel width leading to the conclusion that this is a broadband effect. Furthermore, interferences with molecular absorption and atmospheric scattering above the clouds resulting in a wavelength dependent R may also cause a systematic slope bias. It needs to be noted, however, that the influence of the slope on the accuracy on small cloud fractions is minor.

Section 3.4.1. What is a conclusion of the comparison of MICRU and OCRA? Do the authors attribute the differences between the algorithms to the different treatment of surface BRDF?

The comparison between MICRU and OCRA is discussed in Section 4.2. The revised manuscript now states:

- The accuracy of singular OCRA measurements, however, is significantly and consistently lower compared to MICRU as revealed by the larger scatter of OCRA CF for very small MICRU CF than vice versa
- it can be concluded that OCRA's empirical correction algorithm is a bit too optimistic
- OCRA seems to properly account for this effect on PMD resolution
- This indicates that BRDF effects have a stronger influence on OCRA results for observation geometries opposing the sun and that the empirical VZA correction performed by OCRA is not sufficient.

P.31, L.6. "... different definition of the upper threshold." What do you mean? What is the upper threshold for OCRA?

This issue is now discussed in more detail following the comments of Referee #1. Changes to manuscript are detailed in our answers to his/her comments.

Section 3.4.2. Please formulate a purpose of comparing MICRU with three versions of FRESCO. Why do not select just the latest version of FRESCO?

All three FRESCO versions apply different strategies for supplying the background map leading to significantly different results as presented in the manuscript. The latest version actually not outperforms its predecessors in all aspects as already discussed in Section 3.4.2

We added the following motivation to Section 2.6 (Comparison data):

in order to study the particular differences with respect to background map generation and residual VZA dependence

---

## Author Comment (AC3) · 8 Dec 2020

**Authors' response to the comments of Eun-Su Yang on "MICRU background map and effective cloud fraction algorithms designed for UV/vis satellite instruments with large viewing angles" by Holger Sihler *et al.**

We would like to thank Eun-Su Yang for submitting a short comment (SC1) to our submission to AMTD.

For reference, the original comment below is typeset in black, our response in blue. Modifications of the original manuscript (green) are indicated in red.

The authors need to clarify definitions of relative azimuth angle (RAA) and scattering angle (SA). It seems as though the authors performed their CF calculations using correct RAA and SA in the paper; however, I noticed the definitions of RAA in Figure 1 and SA in Equation (7) do not agree with each other.

Suppose SZA = 30, VZA = 60, and RAA = 180 in Figure 1. Then, we can estimate SA = 90 from Figure 1.

On the other hand, plugging the above SZA, VZA, and RAA into Equation (7) gives:

SA = acos{-cos(VZA)*cos(SZA) + sin(SZA)*sin(VZA)*cos(RAA)}
= acos{-cos(60)*cos(30)+sin(30)*sin(60)*cos(180)} = 150.

Therefore, RAA in Figure 1 should be |VAA - (SAA - 180)| or |VAA - (SAA + 180)| instead of |VAA – SAA| where SAA is solar azimuth angle and VAA is viewing azimuth angle.

We thank Eun-Su Yang very much for pointing out this error. We double checked the SA definition in our calculations and they agree with Equation (7). Hence, we agree that the SAA is not correctly indicated Figure 1. We replaced Figure 1 with